# Reduced cyclin D3 expression in erythroid cells protects against malaria

Maria Giuseppina Marini[1,7], Maura Mingoia[1,7], Maristella Steri[1,7], Ioannis Tsamesidis[2,3], Maria Laura Idda[1,2], Alessia Manca[2], Cristina D'Avino[2], Francesca Virdis[2], Valeria Lodde[2], Antonella Mulas[1], Isadora Asunis[1], Xinyi Li[4], Margaret C. Steiner[4], Angela Loi[1], Cristian A. Caria[1], Maria Franca Marongiu[1], Matteo Floris[2], Michele Marongiu[1,2], Laura Manunza[1], Maristella Pitzalis[1], Valeria Orrù[1], Edoardo Fiorillo[1], Magdalena Zoledziewska[1], Paolo Moi[1], Francesco M. Turrini[5], Mauro Pala[1], Carlo Sidore[1], David Schlessinger[6], John Novembre[4], Antonella Pantaleo[2] & Francesco Cucca[1,2 ✉]

The severity of malaria varies substantially between individuals, but the mechanisms that underlie these differences remain unclear. Because erythrocytes have a key role in malaria biology, genetic variants associated with the development of these cells could inform the mechanisms that determine disease severity. Here we investigate the mechanistic basis of the association of the variant rs112233623-T with erythrocyte properties, and examine its role in modulating malaria severity. This variant is associated with increased levels of haemoglobin A2, increased erythrocyte size and reduced erythrocyte number[1,2]. It is found in an erythroid enhancer of *CCND3*, which encodes cyclin D3—a cell-division activator that enhances the pentose phosphate pathway and thereby helps to counteract reactive oxygen species (ROS)[3]. We show that rs112233623-T disrupts a binding site for the transcription factor SMAD3, weakens enhancer activity and, in erythrocyte precursors (erythroblasts), is associated with reduced *CCND3* expression and inhibition of the G1–S cell-cycle transition, concomitant with a reduction in the number of erythrocytes and an increase in their size. Using population genetic methods, we observe signatures of positive selection for rs112233623-T in the genetic history of Sardinia, a region in which malaria was once prevalent. Furthermore, we show that parasite growth is impaired in cultured *Plasmodium falciparum*-infected erythrocytes from rs112233623-T carriers, and that this impairment correlates with ROS levels. This mirrors our observations in erythrocytes from individuals who are deficient in the pentose-phosphate-pathway enzyme G6PD—a trait associated with protection against malaria in some settings— and highlights a common ROS-based mechanism of malaria resistance. Our results suggest that a reduction in *CCND3* in erythroblasts constitutes a mechanism of resistance to malaria, and could enable therapeutic interventions.

A long history of studies in human genetics has investigated how the powerful evolutionary pressure of endemic malaria has resulted in the selection of genetic variants that moderate the effect of the disease. In 1948, J. B. S. Haldane first suggested that heterozygous carriers of alleles that confer β-thalassaemia, a life-threatening anaemia at that time, could reduce mortality during malaria infection, and this 'heterozygote advantage' might explain the increased frequency of such alleles[4]. Since this initial proposal—commonly referred to as the malaria hypothesis— the increased frequencies, in areas where malaria was historically endemic, of not only β-thalassaemia, but also α-thalassaemia, sickle-cell anaemia (caused by the haemoglobin S (HbS) variant), haemoglobin C (HbC) disease, haemoglobin E (HbE) disease and ovalocytosis have all been hypothesized to be the result of selection[5]. Malaria resistance has also been proposed to account for the high frequency of alleles causing deficiency of the enzyme glucose-6-phosphate dehydrogenase (G6PD), and genetic studies have provided evidence for positive selection of the *G6PD*-deficiency A allele[6].

Broader genome-wide association studies (GWASs) have now identified additional DNA variants that are associated with the clinical severity of malaria[7–12], some of which—such as variation in the *ATP2B4* gene[13]—are also associated with features of red blood cells (RBCs), the cells that host key stages of the malaria parasite's life cycle in a human host. However, the mechanisms by which these variants and genes can both affect RBC parameters and ameliorate malaria remain mostly unknown.

[1]Institute for Genetic and Biomedical Research, National Research Council, Cagliari, Italy. [2]Department of Biomedical Sciences, University of Sassari, Sassari, Italy. [3]Department of Biomedical Sciences, International Hellenic University, Thessaloniki, Greece. [4]Department of Human Genetics, University of Chicago, Chicago, IL, USA. [5]Department of Oncology, University of Turin, Turin, Italy. [6]Laboratory of Genetics and Genomics, National Institute on Aging, NIH Baltimore, Baltimore, MD, USA. [7]These authors contributed equally: Maria Giuseppina Marini, Maura Mingoia, Maristella Steri. ✉e-mail: fcucca@uniss.it

Here we focus on explaining the molecular action of one allelic variant in the *CCND3* gene region, in terms of its effects on both blood-cell traits and resistance to malaria. Our first finding was an association signal with increased levels of haemoglobin A2 (HbA2), marked by the derived T allele of single-nucleotide polymorphism (SNP) rs112233623 close to *CCND3*, the gene encoding the cyclin D3 protein[1]. Notably, this allelic variant showed a much higher frequency (minor allele frequency (MAF) = 10.1%) in a population cohort from Sardinia[14], where the association was first detected, than in a sample set from northern Europe[1] (MAF = 1%).

Within the same gene region, the CHARGE consortium study (composed mainly of northern European individuals)[15] had previously found an association signal with reduced mean corpuscular volume (MCV) and increased RBC count, marked by the derived A allele of SNP rs9349205, located 160 bp upstream of rs112233623. Following up on this genetic finding, another group experimentally demonstrated the erythroid-cell enhancer activity of the core region surrounding SNP rs9349205, and a further increase in enhancer activity specifically for rs9349205-A, compared with the baseline ancestral G allele[16]. The researchers also showed that *Ccnd3*-knockout mice had a decreased RBC count and increased MCV, effects opposite to those of rs9349205-A in humans[16]. After further examination of human and mouse primary erythroid cells, the team proposed that cyclin D3 affects the number of cell divisions during the terminal differentiation of erythroid precursors, and thereby influences the size and number of erythrocytes, such that a lower level of *CCND3* leads to fewer but larger RBCs[16].

A more recent study in a large UK Biobank sample used statistical fine mapping in the *CCND3* gene region and stepwise conditional analysis to identify rs9349205-A as the primary association for decreased RBC count and rs112233623-T as a putative independent associated variant with opposite effects[2].

In addition to its effects on the cell cycle, the cyclin D3–CDK6 complex has been found in cancer cells to phosphorylate and thereby reduce the activity of the rate-limiting glycolytic pathway enzymes 6-phosphofructokinase (PFK1) and pyruvate kinase M2 (PKM2)[3]. Consequently, glucose-derived carbon is shunted into the pentose phosphate and serine pathways, leading to increased formation of NADPH, the source of reducing power to neutralize ROS[3]. To our knowledge, no investigation has so far reported whether these metabolic effects in cancer cells are recapitulated in RBCs and their precursors, and, if so, whether they are modified by rs112233623-T and rs9349205-A.

We therefore began this study with a series of related questions. How are the known actions of *CCND3* related to the known erythrocyte trait effects of variant rs112233623-T? Does this allelic variant affect the known *CCND3* erythroid enhancer, and, in particular, does it affect the binding of specific transcription factors? What effects are seen when rs112233623-T and rs9349205-A are inherited together on the same DNA molecule (haplotype)? Is the higher frequency of rs112233623-T in Sardinian individuals, compared with other European cohorts, more likely to be explained by random drift or positive selection during the evolutionary history of this population? And if selective mechanisms were involved, what was the selective pressure? In the present work, we address all of these queries, and we outline a role for reduced *CCND3* function in malaria resistance and identify its probable mechanism. Our results suggest that inhibiting *CCND3* could constitute a therapeutic modality for malaria.

## Genetic associations in the *CCND3* enhancer

Extending our previous work on the genetic architecture of haemoglobin levels in 6,305 individuals[1], we examined genetic associations in the *CCND3* gene region for a larger set of haematological traits in 6,824 individuals from the SardiNIA general population cohort (Methods). In contrast to findings in cohorts from northern Europe, where the association plots for the haematological traits RBC and MCV in this gene

**Table 1 | Association results for the rs112233623-T variant**

| Trait | Number of samples | Genotype count (CC/CT/TT) | MAF | Effect (β) | Standard error | *P* value |
|---|---|---|---|---|---|---|
| HbA2 (g dl⁻¹) | 6,762 | 5,480/1,219/63 | 0.099 | 0.306 | 0.032 | $4.03 \times 10^{-22}$ |
| MCH | 6,824 | 5,526/1,234/64 | 0.1 | 0.319 | 0.031 | $2.48 \times 10^{-24}$ |
| MCV | 6,824 | 5,526/1,234/64 | 0.1 | 0.356 | 0.032 | $3.77 \times 10^{-29}$ |
| RBC | 6,819 | 5,521/1,234/64 | 0.1 | −0.267 | 0.032 | $1.18 \times 10^{-16}$ |
| HbF (g dl⁻¹) | 6,710 | 5,436/1,211/63 | 0.099 | 0.091 | 0.032 | $4.32 \times 10^{-3}$ |
| HbA1 (g dl⁻¹) | 6,708 | 5,434/1,211/63 | 0.099 | 0.003 | 0.032 | 0.932 |

Tests for association of rs112233623-T with haematological phenotypes measured in the SardiNIA cohort. The table reports, from left to right, the trait tested; the number of samples (*n*) included in the analysis for each trait; the genotype count; the MAF; the effect size (*β*) expressed in standard deviation units, and its standard error; and the *P* value.

region are led by rs9349205-A (ref. 2), in Sardinia, rs112233623-T dominates the association profile, alongside another variant, rs113267280-G, that is in strong linkage disequilibrium (LD) with it ($r^2 = 0.96$). These two variants form the credible set for HbA2, RBC MCV and mean corpuscular haemoglobin (MCH) associations (Methods, Extended Data Fig. 1 and Supplementary Table 1), and it is not possible to determine from the genetic association data which variant is primarily associated with these traits and which is secondarily associated owing to high LD. Still, there is evidence—such as a higher index of predicted deleterious effect for rs112233623-T than for rs113267280-G, using the combined annotation-dependent depletion (CADD) score (https://cadd.gs.washington.edu/score) (16.21 versus 2.27, respectively)—suggesting that rs112233623-T is the primarily associated variant, as was originally proposed for the HbA2 association[1].

Focusing on this SNP and integrating previously reported associations[1,2], the derived T allele of rs112233623 is associated with increased HbA2 ($P = 4.03 \times 10^{-22}$, $\beta = +0.31$), MCV ($P = 3.77 \times 10^{-29}$, $\beta = +0.36$) and MCH ($P = 2.48 \times 10^{-24}$, $\beta = +0.32$), and decreased RBC count ($P = 1.18 \times 10^{-16}$, $\beta = -0.27$), as well as with increased fetal haemoglobin (HbF) ($P = 4.32 \times 10^{-03}$, $\beta = +0.09$) (Table 1 and Supplementary Information). The phenotypic effects on MCV and RBC count are similar to the haematological profile seen in *Ccnd3*-knockout mice[16], suggesting that the association signal marked by rs112233623-T results from reduced *CCND3* activity, probably owing to decreased expression.

Of note, in the SardiNIA cohort, the derived allele A of rs9349205 associates with increased MCV ($P = 8.8 \times 10^{-5}$, $\beta = +0.09$) and MCH ($P = 1.4 \times 10^{-4}$, $\beta = +0.09$), and decreased RBC count ($P = 1.6 \times 10^{-2}$, $\beta = -0.06$). This is seemingly the same direction of effects as rs112233623-T, and opposite to the decrease in MCV and increase in RBC count previously reported for rs9349205-A in other populations[2,15]. However, the apparent contradiction can be resolved with conditional association analysis. After controlling for the effect of rs112233623-T, rs9349205-A was indeed associated with a reduction in MCV ($P = 2.3 \times 10^{-2}$, $\beta = -0.05$) and MCH ($P = 3.3 \times 10^{-2}$, $\beta = -0.05$), and an increased RBC count ($P = 1.6 \times 10^{-2}$, $\beta = +0.06$), consistent with findings in other populations and with increased *CCND3* expression driven by rs9349205-A (refs. 16,17).

Similarly, in single-variant analyses, we find that rs9349205-A is associated with increased HbA2 ($P = 1.4 \times 10^{-9}$, $\beta = +0.14$), again with the same direction of effect as rs112233623-T. However, after conditional association analyses accounting for the effect of rs112233623-T, the association of rs9349205-A with HbA2 completely disappeared ($P = 0.64$), in contrast to the opposite effects of rs112233623-T on RBC count, MCV and MCH traits. This suggests that, unlike the putative decrease of *CCND3* expression due to rs112233623-T, which has a strong positive effect on HbA2 levels, the increase of *CCND3* expression due to rs9349205-A alone might not have any primary effect on HbA2 levels.

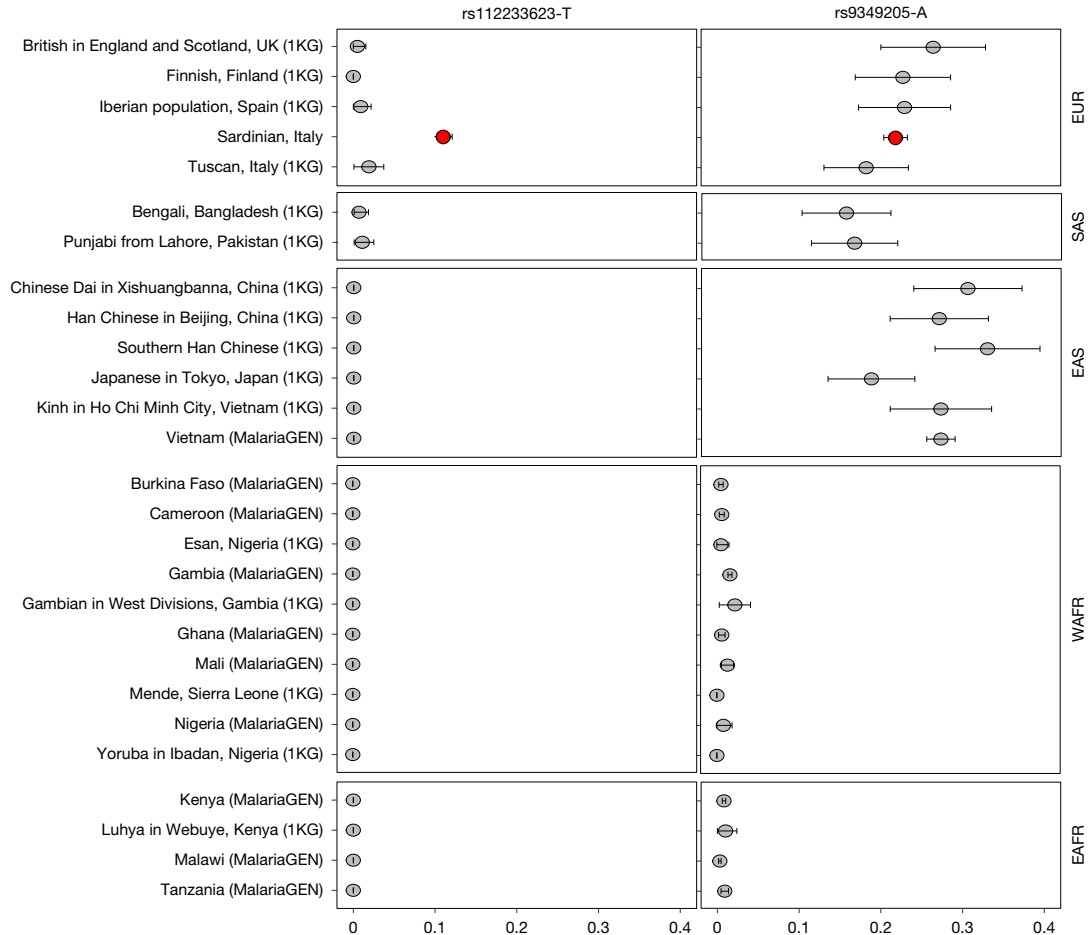

**Fig. 1 | Frequency of the rs112233623-T and rs93429205-A alleles in various human populations.** The frequencies of the rs112233623-T and rs93429205-A alleles in Sardinian and other populations were calculated in individuals who had undergone whole-genome sequencing, from the SardiNIA cohort (1,577 unrelated samples[51]; red circles) and from 1KG (2,504 unrelated samples[18]; grey circles). The allele frequencies for each variant, calculated under a binomial model, are shown as point estimates for each sampled population, with error bars indicating the 95% confidence interval. Any error bars that are not visible remain within the point symbol. Abbreviations on the right denote ancestry groups: EAFR, East African; EAS, East Asian; SAS, South Asian; WAFR, West African.

To better understand the contrasting association results for rs9349205-A and haematological traits in the SardiNIA cohort, compared with other European cohorts, and why conditional analysis adjusting for the effect of rs112233623-T renders the associations consistent, it is useful to examine the allele and haplotype frequencies of the two SNPs in various populations (for example, Sardinian individuals from the SardiNIA study cohort and European (EUR) individuals from the 1000 Genomes Project (1KG)[18]). The derived rs9349205-A allele is common both in SardiNIA and in 1KG EUR (frequency 21.8% and 24.4%, respectively), whereas the derived rs112233623-T allele is common in SardiNIA (frequency 10.1%) but much rarer in 1KG EUR, especially in the northern Europe subset (frequency 1%; Fig. 1). In Sardinian individuals, the two variants show considerable LD ($r^2 = 0.4$, $D' = 1$); the rs112233623-T allele is always found on an rs9349205-A background (Extended Data Fig. 2 and Supplementary Table 2), and roughly half of the individuals carrying rs9349205-A carry rs112233623-T.

In 1KG EUR, the rs112233623-T allele is also observed in an rs9349205-A background, but because it is much rarer, nearly all of the rs9349205-A-carrying haplotypes contain the alternative rs112233623-C allele (Extended Data Fig. 2 and Supplementary Table 2). A plausible scenario is that the derived rs112233623-T allele arose on a haplotype with the rs9349205-A-derived allele background, and subsequently the derived|derived allele haplotypic configuration rose in frequency locally in Sardinia. The difference in the discovery-association results between Sardinian and northern European cohorts (shown above) can thus be explained by the higher frequency of the rs9349205-A|rs112233623-T haplotype in Sardinian individuals, in which the expected effect of rs9349205-A could be reversed by a larger inferred opposite effect of rs112233623-T under a simple additive model (Supplementary Information).

This evidence together suggests that rs112233623-T is the primary variant underlying an association with several blood-cell traits that might be mediated by changes in *CCND3* expression. Next, therefore, we sought to assess the effects of rs112233623-T on *CCND3* expression and cell-cycle progression in erythroblast cultures.

## rs112233623-T and reduced *CCND3* expression

To assess the effects of rs112233623-T on *CCND3* erythroid expression, we used quantitative PCR with reverse transcription (RT–qPCR) and western blot analysis in in-vitro-differentiated primary erythroblasts derived from SardiNIA cohort volunteers who were homozygous for either the wild-type (WT) rs112233623-C allele (CC genotype) or the derived rs112233623-T allele (TT genotype). We focused on the basophilic stage of erythroblast differentiation, which corresponds to the peak of *CCND3* expression[19,20] (Extended Data Fig. 3). Compared with erythroblasts from individuals with the rs112233623-CC genotype, *CCND3* expression was reduced at both the mRNA and the protein level in erythroblasts from individuals with the rs112233623-TT genotype (Fig. 2a and Extended Data Fig. 4a).

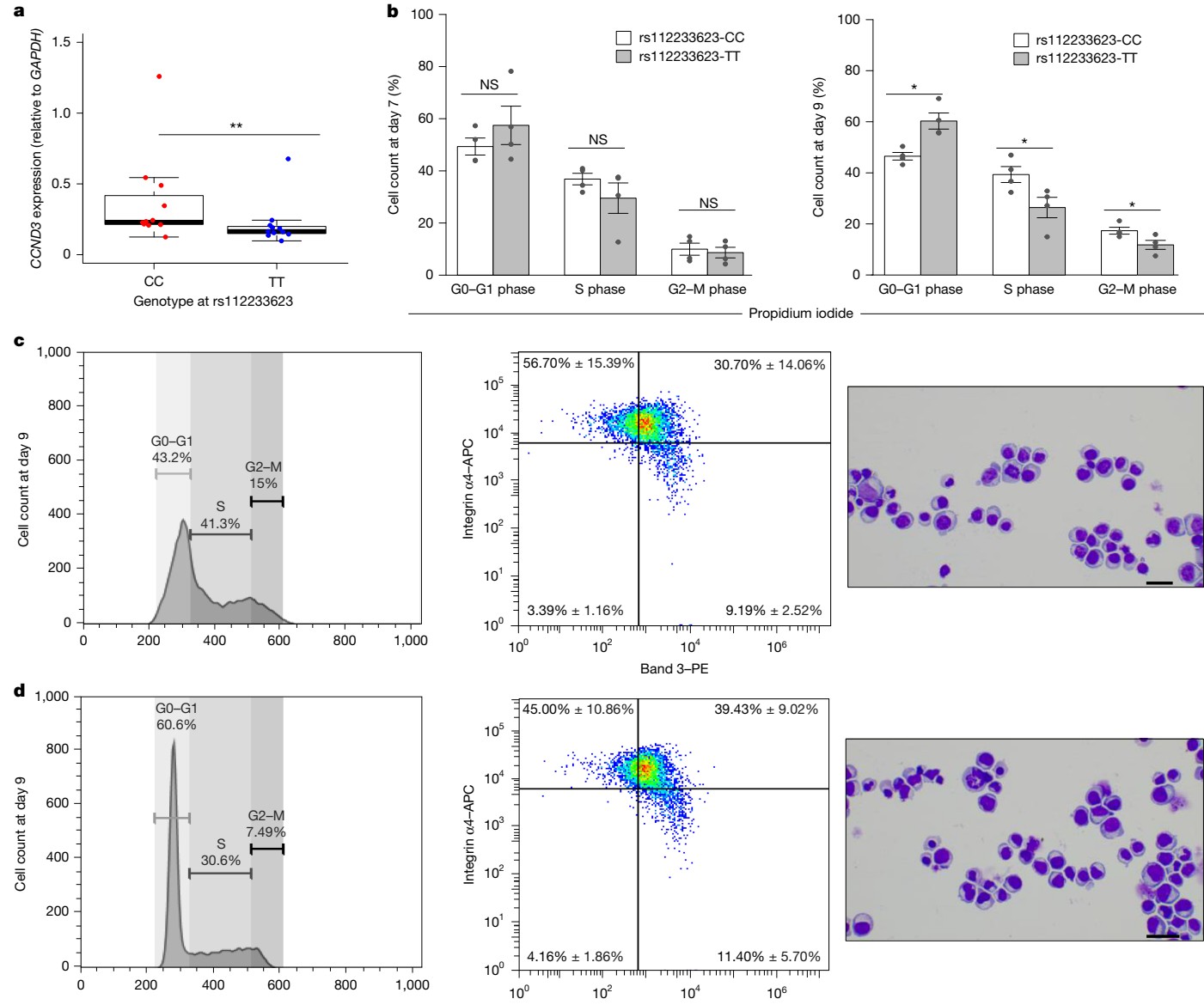

**Fig. 2 | Influence of the rs112233623-T allele on *CCND3* expression and the cell cycle in erythroid cultures. a**, Relative expression of *CCND3* mRNA in cultured erythrocyte progenitors (erythroblasts) from individuals who are homozygous for the WT allele rs112233623-C (CC), compared with those who are homozygous for the derived allele rs112233623-T (TT). The expression of *CCND3* was quantified by RT–qPCR relative to the expression of the endogenous control *GAPDH* at day 7 of the in-vitro-differentiated erythroid culture (for each genotype, *n* = 11 individuals). In the box plots, the line across the box indicates *CCND3* median expression, the box ranges from the first quartile to the third quartile of the distribution and the whiskers extend from each box to the most extreme data points. A Wilcoxon–Mann–Whitney test was used, with the level of significance indicated by asterisks (**$P$ < 0.01; one-sided $P$ = 9.62 × 10$^{-3}$).
**b**, Histograms showing the distribution of cell-cycle stages, reported as G0–G1, S and G2–M, in erythroblasts derived from individuals homozygous for the

derived rs112233623-T allele versus the WT rs112233623-C allele at two time points of erythroid differentiation: day 7 (left) and day 9 (right). Data are mean ± s.e.m.; *n* = 4 per genotype. A two-sided Student's *t*-test was used, with the level of significance indicated by asterisks (*$P$ < 0.05; NS, not significant). The statistical results for all comparisons are provided in Supplementary Table 3.
**c,d**, Analysis of erythroblasts derived from individuals homozygous for the WT rs112233623-C allele (**c**) or the derived rs112233623-T allele (**d**) at day 9 of erythroid differentiation. In each panel, from left to right: representative flow-cytometry plots with propidium iodide staining, showing the distribution of cell-cycle phases (G0–G1, S and G2–M); representative flow-cytometry dot plots of GPA$^+$-gated erythroblasts, showing the expression of Band 3 and integrin α4 (data are mean ± s.e.m.; *n* = 4 per genotype); and cytospin preparations stained with May–Grünwald–Giemsa (scale bars, 20 μm; 50× magnification).

Using erythroblasts derived from volunteers in the same cohort, we further tested whether the reduction in *CCND3* expression associated with the rs112233623-T allele inhibits the transition from G1 to S phase, as previously described for the loss of function of *Ccnd3* in knockout mice[16]. We measured cell-cycle levels at two time points of differentiation—day 7 and day 9—corresponding mainly to the basophilic and to the polychromatophilic stages of erythroblast differentiation, respectively; at the latter polychromatophilic stage,

*CCND3* expression declines (Extended Data Fig. 3), according to published RNA-sequencing (RNA-seq) data[19,20]. In erythroblasts with the rs112233623 TT genotype, a higher percentage of cells in G0–G1 and a lower percentage of cells in S and G2–M phases were observed at day 7, and these differences became significant at day 9 of differentiation (Fig. 2b–d and Extended Data Fig. 4c,d). These results were confirmed by the differential growth curves observed for the two genotypes (Extended Data Fig. 4b).

These results support a model in which the rs112233623-T allele is associated with reduced *CCND3* expression, which, in turn, leads to reduced cell-cycle progression during erythropoiesis, a reduction in the number of mitoses during erythropoiesis, a higher MCV and a lower RBC count. We next sought to understand the mechanism by which of rs112233623-T affects *CCND3* expression.

## An inhibitory site in the *CCND3* enhancer

We next investigated how the activity of the *CCND3* enhancer region, where rs112233623 and rs9349205 are located, might be altered or disarmed by rs112233623-T to account for the reduced expression of *CCND3* and resulting haematological associations. We cloned a 372-bp DNA fragment of the enhancer surrounding rs9349205 and rs112233623, carrying either the WT configuration (the G|C allelic combination across both SNPs, respectively, in the haplotype) or the combined derived alleles (the A|T allelic combination) of these SNPs, into a luciferase expression vector. The constructs were then tested for their ability to activate expression of luciferase in human umbilical cord-derived erythroid progenitor 2 (HUDEP-2) cells, a model that can recapitulate the development of erythroid cells[21]. In these transfection experiments, the derived A|T allele combination showed a ten-fold reduction in enhancer activity, compared with the WT G|C allele combination (Extended Data Fig. 5a), despite experimental findings showing that the A-derived allele of rs9349205 alone increases activity[16]. These functional results are thus consistent with our genetic evidence that the action of rs112233623-T overrides the opposite effect of rs9349205-A. They are also in agreement with the reduced expression of *CCND3* mRNA in erythroblasts derived from individuals who are homozygous for the rs112233623-T allele (shown above), and with the results of another study, which found that shorter constructs that retained the T allele had reduced activity in luciferase transactivation experiments in K562 erythroleukaemia cells[2].

The luciferase reporter results suggested that the rs112233623-T allele modifies transcription-factor binding. We investigated this possibility further by in silico inference of putative transcription-factor-binding sites in the sequence surrounding the rs112233623 variant (https://bio.tools/alggen). We identified a consensus binding sequence for SMAD3 (CAGACA) in the WT sequence, the fifth nucleotide of which corresponds to rs112233623-C. SMAD3 is a member of the SMAD family of transcription factors, which are involved in late erythroid maturation[22]. Notably, the change rs112233623C>T in the binding sequence (CAGATA) would be expected to abolish SMAD3 binding while creating a putative binding site (AGATA[A/G]) for GATA1 (Extended Data Fig. 5b). GATA1 is a key member of the GATA family of transcription factors, which are involved in haematopoietic development[23], and acts as a repressor in this context (see below).

The predictions of transcription-factor binding from the in silico analysis were experimentally confirmed in vitro using labelled oligonucleotide probes containing either the WT (C) or the derived (T) allele of rs112233623 in electrophoretic mobility shift assays (EMSAs) in cell extracts expressing either SMAD3 or GATA1 protein (Methods). By supershift analysis with antibodies and competition analysis, we found that the C>T transition in rs112233623 indeed reduced the binding of SMAD3 (Extended Data Fig. 5c) and increased the binding of GATA1 (Extended Data Fig. 5d).

The differential binding of SMAD3 to the region surrounding rs112233623 was confirmed by chromatin immunoprecipitation (ChIP) assays in erythroblasts from volunteers homozygous for either WT rs112233623-C or the derived rs112233623-T allele (Extended Data Fig. 5e). However, in the ChIP assays, GATA1 binding remained relatively unchanged, presumably compensated by a nearby GATA motif (Extended Data Fig. 5f).

We next confirmed the proposed effects of the variants on the erythroid-specific enhancer of *CCND3* in the presence of SMAD3. Luciferase expression constructs containing the *CCND3* enhancer bearing the rs112233623-C allele were expressed in HUDEP-2 cells at higher levels after transactivation with SMAD3, especially when the coactivator p300 was also provided[24]. Conversely, the level of expression of constructs bearing the rs112233623-T allele showed no response to transactivation (Extended Data Fig. 5a).

By contrast, the same *CCND3* luciferase constructs expressed in HUDEP-2 cells but cotransfected with GATA1 in combination with its coactivator FOG1[25] showed an even greater reduction in luciferase activity of the *CCND3* enhancer bearing the rs112233623-T derived allele. The activity of the *CCND3* enhancer was also reduced, albeit to a lesser extent, for luciferase constructs containing the WT rs112233623-C allele (Extended Data Fig. 5a). A plausible explanation for this intrinsically slight repressive effect of GATA1 on the WT enhancer is the presence of an additional canonical GATA1 site in the WT enhancer fragment, which was shown to be occupied by the GATA1 protein in previous ChIP–seq assays[16].

Together, these data confirm that the rs112233623-T derived allele of the *CCND3* erythroid enhancer decreases the expression and impairs the activity of *CCND3* through reduced binding of the activator SMAD3 and increased binding of the repressor GATA1.

## Blocking TGFβ–SMAD reduces the levels of cyclin D3

Because SMAD3 and the highly related SMAD2 are intermediate effectors of TGFβ signalling[26], we thought that the TGFβ–SMAD3 pathway would be likely to modulate the expression of *CCND3* in erythroid cells. We verified this hypothesis by analysing the expression of *SMAD3* and *CCND3* in HUDEP-2 cells at various time points after treatment with TGFβ. qPCR analysis showed that induction by TGFβ led to a rapid increase in *SMAD3* expression, followed by a substantial increase in the expression of *CCND3* (Extended Data Fig. 6a). This result is consistent with a previous study, in which TGFβ increased the expression of *CCND3* in MCF-7 cells[27]. Moreover, treating HUDEP-2 cells simultaneously with both TGFβ and SB-505124 — a specific inhibitor of the phosphorylation of the TGFβR1 receptors ALK4 and ALK5 — prevents SMAD3 phosphorylation and thereby precludes any subsequent TGFβ-mediated increase in the expression of *CCND3*.

To further validate the stimulation of *CCND3* by SMAD3 in HUDEP-2 cells, while accounting for potential compensatory actions of SMAD2[28], we inactivated both SMAD3 and SMAD2 by CRISPR–Cas9 targeted editing. This resulted in an overall reduction of 60% in the expression of the target *CCND3* gene, with an even greater decrease of 86% at the protein level (cyclin D3/β-actin ratio; Extended Data Fig. 6b, bottom). These findings are in agreement with previous observations of a significant reduction in *Ccnd3* expression in mouse stromal cells when both *Smad2* and *Smad3* were knocked down[29].

## Selection for low erythroid levels of cyclin D3

Given its phenotypic effects, the evolutionary history of the *CCND3* rs112233623-T allele is of particular interest. The rs112233623-T allele is common in the SardiNIA cohort (MAF of around 10%), less frequent in cohorts from mainland Italy (MAF of around 2%), rarer still in cohorts from northern Europe and South Asia (MAF < 1%) and thus far undetected in cohorts from Africa and East Asia (Fig. 1). We used evidence from allele frequency differentiation and haplotype variation (Methods) to assess whether the higher frequency of rs112233623-T in the SardiNIA cohort is likely to be the result of positive selection that favoured this specific variant in the genetic history of Sardinia, or whether patterns of variation at the locus are best explained by random genetic drift. Consistent with positive selection on this variant, the observed differentiation in allele frequency between SardiNIA and 1KG EUR samples is greater than that seen for the vast majority of 1,075 genomic variants matched for allele frequency in Sardinia and other pertinent

features ($F_{ST}$ = 0.094, 99.9th 'genomic percentile' of differentiation; Extended Data Fig. 7a). Using a more specific metric for positive selection based on the length of the core haplotype around the variant of interest (integrated haplotype score; iHS[30]), the haplotype carrying rs112233623-T shows a level of extended homozygosity in SardiNIA that is in the tail of values observed in matched variants across the genome (iHS = 2.16, 99.1th genomic percentile; Extended Data Fig. 7b,d). Likewise, the haplotype diversity is reduced in individuals from the SardiNIA cohort, compared with the 1KG EUR cohort (cross-population extended haplotype homozygosity (xp-EHH) = 1.99, 99.4th genomic percentile; Extended Data Fig. 7c). Reconstruction of allele frequency trajectories over time using a local ancestral recombination graph surrounding rs112233623 (using CLUES[31]; Methods) showed a rise in the frequency of rs112233623-T towards the present time, with a paired selection coefficient point estimate for this allelic variant that was among the highest values obtained for comparable genomic variants ($s$ = 0.026, 99.5th genomic percentile; Extended Data Fig. 8 and Supplementary Information). Together, these observations support the inference that the haplotype carrying the rs112233623-T decrease of expression (DoE) allele had a beneficial effect on fitness in the genetic history of Sardinia.

### CCND3 DoE impairs *P. falciparum* through ROS

Given the already described phenotypic effects on blood-cell traits, and the plausible evidence for positive selection on the *CCND3* DoE rs112233623-T variant, we wondered whether the variant is associated with malaria outcomes—especially because malaria was more prevalent in Sardinia than it was anywhere else in Europe until its eradication there around the 1950s[32,33]. To investigate this possibility, we assessed the effect of this variant on the *P. falciparum* life cycle in vitro, using RBCs derived from study participants. We also tested the *CCND3* increase of expression (IoE) rs9349205-A variant, given its opposite biological—and thus possibly opposite evolutionary—effects, as well as the combination of both alleles. To do this, we used the SardiNIA general population cohort, recalling individuals who were homozygous for the haplotypes carrying the variants of interest (Fig. 3a). The individuals included 17 homozygotes for the *CCND3* IoE rs9349205-A and the DoE rs112233623-T derived alleles (IoE/IoE|DoE/DoE); 14 homozygotes for the IoE allele rs9349205-A and for the WT allele rs112233623-C (IoE/IoE|WT/WT); and, as a control group, 7 homozygotes for the WT basal expression alleles rs112233623-C and rs9349205-G (WT/WT|WT/WT). To reduce background variation and avoid confounding, the individuals in these categories were selected from among those who do not carry other reported protective malaria variants that are common in Sardinia (for example, β- or α-globin or the *G6PD*-deficient Mediterranean (*G6PD*-Med) variants; Methods). We also included an additional group of five male individuals hemizygous for the *G6PD*-Med allele (rs5030868-A) and carrying the WT genotype at rs9349205 and rs112233623 (WT/WT|WT/WT, *G6PD*-Med); this permitted us to compare the *CCND3* variants of interest with a variant that has been previously proposed to confer protection against malaria derived from *P. falciparum* and *Plasmodium vivax*[34–37].

To assess genotypic effects on malaria resistance, RBCs purified from recalled SardiNIA donors were infected with the Uganda Palo Alto strain of *P. falciparum* (FUP) and maintained in culture for three blood-stage cycles, as described in the Methods.

Infected RBCs from individuals with the IoE/IoE|DoE/DoE genotype (that is, homozygous for the haplotype carrying the *CCND3* IoE and the DoE alleles together) exhibited progressively impaired *P. falciparum* growth, as indicated by reduced parasitaemia levels when compared with the WT/WT|WT/WT control group (Fig. 3b and Supplementary Table 6). Notably, the inhibition of parasite growth in IoE/IoE|DoE/DoE RBCs, which was already evident during the first cycle and suggestive of reduced invasion efficiency, became even more evident from the second cycle onwards, with parasites being retained in the schizont stage, leading to delayed merozoite egress and reinvasion of uninfected RBCs (Fig. 3c). Furthermore, the growth restriction became more pronounced in the third cycle, manifesting as a marked reduction in viable parasites, compared with the control group (WT/WT|WT/WT; Fig. 3c). A significant inhibition of parasite growth was observed in IoE/IoE|DoE/DoE RBCs (Fig. 3d), with comparable effects seen in hemizygous *G6PD*-Med erythrocytes (WT/WT|WT/WT *G6PD*-Med) (Fig. 3b–d).

By contrast, RBCs from individuals carrying the IoE/IoE|WT/WT genotype (that is, homozygous for the haplotype split carrying the IoE allele but not the DoE allele) showed the highest level of parasitaemia, faster parasite development and a lack of inhibition of parasite growth, compared with the other genotypes (Fig. 3b–d).

The differences in parasite development between the various genotypes are also evident in photomicrographs of blood smears (Extended Data Fig. 9) and in the videos available at https://figshare.com/s/d89f8f9f90d3a76eccb9 (ref. 38).

Consistent results with regard to parasitaemia levels were also obtained by repeating the same experiments in RBCs from a smaller group of recalled individuals carrying the same genotypes of interest as those studied in the previous experiments, but infected in parallel with the *P. falciparum* FUP strain and two other *P. falciparum* isolates (3D7 and DD2). This experiment suggests that the observed associations, including those between DoE rs112233623-T and *P. falciparum* growth, are unlikely to be influenced by genetic differences in *P. falciparum* (Extended Data Fig. 10).

Together, these in vitro results show that parasite growth is substantially reduced in RBCs of DoE rs112233623-T carriers. Furthermore, the results suggest a modest—if any—detrimental effect of the IoE rs9349205-A on parasitaemia during malaria infection when it is not on the same haplotype as rs112233623-T.

Given the reported effects of inhibition of cyclin D3 on oxidative stress in various cancer-cell lineages[3], we hypothesized that rs112233623-T-mediated reduction of *CCND3* expression might result in an increase in ROS in infected RBCs, and that this could eventually contribute to parasite clearance. This would thus be analogous to the effect seen when G6PD, the first enzyme in the pentose phosphate pathway, is lacking. Opposite effects in the production of ROS would then be expected in infected RBCs from carriers of rs9349205-A who do not also carry rs112233623-T. To test these hypotheses, we used liquid chromatography and mass spectrometry (LC–MS) to measure the levels of ROS (hydrogen peroxide; $H_2O_2$) in RBCs from donors with different *CCND3* and *G6PD* genotypes (see above). RBCs purified from individuals homozygous for the DoE rs112233623-T allele indeed showed a significant increase in the concentration of ROS, compared with RBCs purified from individuals homozygous for the WT rs112233623-C allele—although not as extreme as that seen in individuals hemizygous for G6PD deficiency (*G6PD*-Med; Fig. 3e). By contrast, a reduction in ROS levels was observed in individuals homozygous for the IoE rs9349205-A allele, compared with those who were homozygous for the WT rs9349205-G allele (and thus also homozygous for WT rs112233623-C). When we correlated ROS levels with *P. falciparum* growth data, we observed a significant inverse correlation between ROS and parasite development during the third cycle (144 hours post-infection; hpi), with individuals from different *CCND3* genotype categories clustering coherently (Fig. 3f).

These findings suggest that the reduction of *P. falciparum* growth in RBCs and the increased levels of ROS observed in the context of reduced *CCND3* expression and *G6PD*-Med deficiency are likely to reflect a shared mechanism of resistance based on increased oxidative stress.

## Discussion

Here we provide direct experimental evidence showing that the genetic variant rs112233623-T has a causal role in haematological trait variation and confers protection against malaria. We show that rs112233623-T abolishes a DNA-binding site for the activating transcription factor

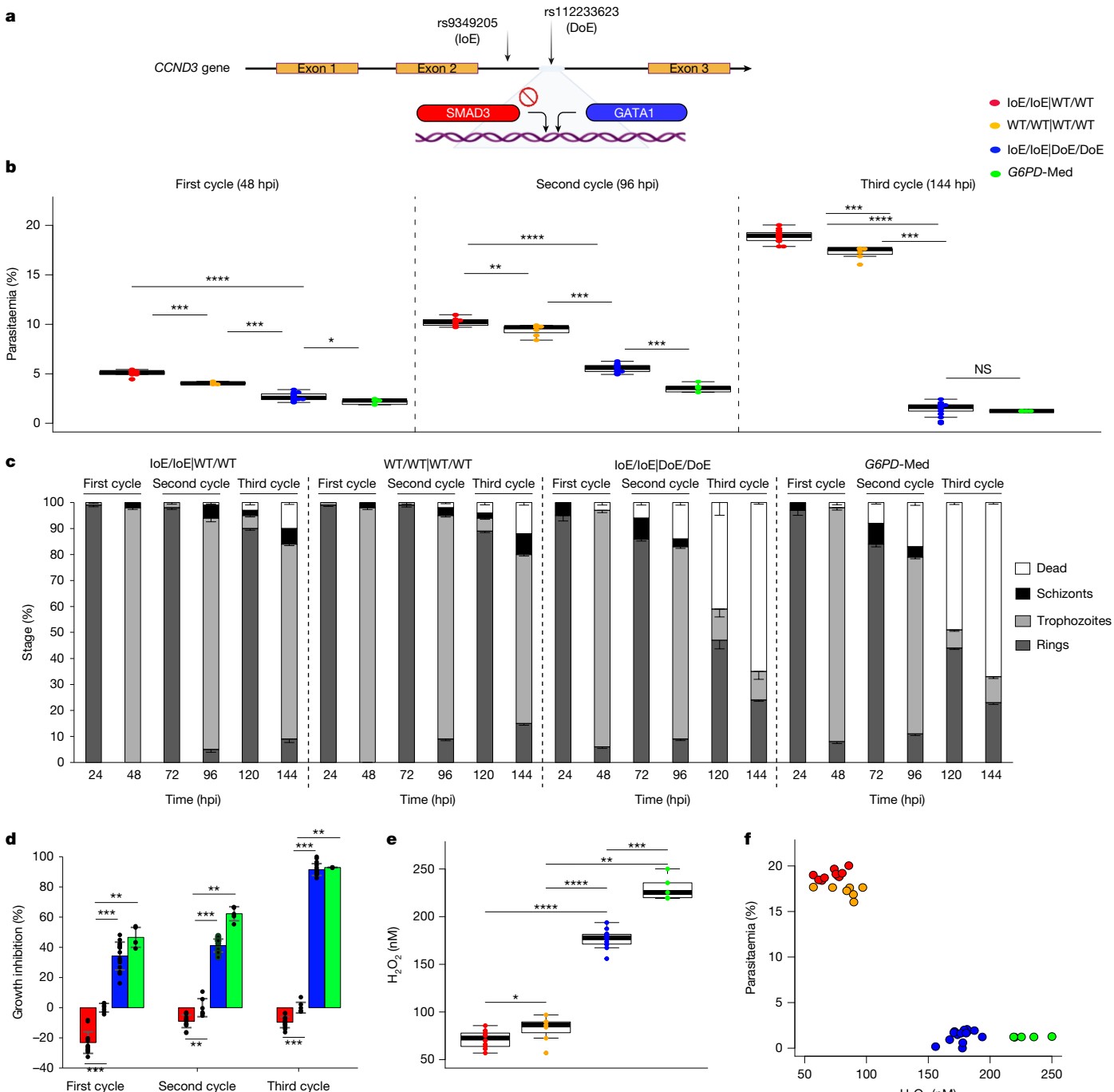

**Fig. 3 | Effects of genotype variation at rs112233623 and rs9349205 in RBCs infected with *P. falciparum*. a**, Schematic representation of the *CCND3* erythroid enhancer, showing the relative positions of the SNPs based on the *CCND3* full-length coding sequence (chromosome 6, NC_000006.12, GRCh38. p14). **b**, Level of *P. falciparum* parasitaemia at each of three life cycles (48, 96 and 144 hpi) in RBC samples from donors with the following genotypes: homozygotes for the IoE rs9349205-A allele and the WT basal expression rs112233623-C allele (IoE/IoE|WT/WT; *n* = 14) in red; homozygotes for the rs9349205-G and rs12233623-C WT alleles (WT/WT|WT/WT; *n* = 7) in orange; homozygotes for the rs9349205-A IoE and rs112233623-T DoE alleles (IoE/IoE|DoE/DoE; *n* = 17) in blue; and homozygotes for the rs9349205-G and rs112233623-C WT alleles, and hemizygotes for the *G6PD* Mediterranean allele (*G6PD*-Med; *n* = 5), in green. In the box plots, the horizontal line indicates the median level of parasitaemia, the box denotes the interquartile range (IQR; 25th–75th percentile) and the whiskers extend to the most extreme data points within 1.5 × IQR of the box, with points beyond this range plotted individually as outliers. **c**, Percentage

distribution of different parasite stages (rings, trophozoites, schizonts and dead parasites) over three life cycles in RBCs. Samples are as in **b**, with genotypes at the top of the graphs. The first cycle is defined as 24–48 hpi, the second cycle as 72–96 hpi and the third cycle as 120–144 hpi. Data are mean ± s.d. for each stage. **d**, Percentage of parasite growth inhibition, normalized to the control group (WT/WT|WT/WT), at each of three life cycles (48, 96 and 144 hpi, respectively) in the same samples as in **b**. Data are mean ± s.d.; colours correspond to the genotypes in **b**. **e**, Box plots of ROS ($H_2O_2$) concentration in RBCs from donors with the indicated genotypes, coloured as in **b** (*n* = 10, 7, 14 and 5 biologically independent samples, respectively). **f**, Correlation between *P. falciparum* parasitaemia and ROS ($H_2O_2$) levels in infected RBCs at 144 hpi. Values for each donor are indicated by dots, with colours corresponding to genotypes as in **b**. All *P* values in **b**,**d**,**e** are from two-sided non-parametric Wilcoxon–Mann–Whitney tests, with significance indicated by asterisks (**P* < 0.05, ***P* < 0.01, ****P* < 0.001 and *****P* < 0.0001). Statistics for all comparisons are provided in Supplementary Table 3.

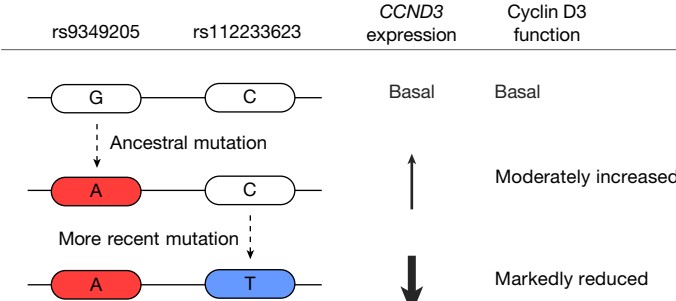

| rs9349205 | rs112233623 | *CCND3* expression | Cyclin D3 function |
|---|---|---|---|
| G | C | Basal | Basal |
| A | C | Moderately increased | |
| A | T | Markedly reduced | |

**Fig. 4 | Influence of different rs9349205–rs112233623 haplotypes on the expression and function of *CCND3*.** The white oblongs represent the WT G and C alleles at the rs9349205 and rs112233623 loci, respectively. The derived alleles at these loci are indicated by red and blue oblongs, respectively. The top dotted arrow depicts an earlier mutation (occurring before modern humans migrated out of Africa) from rs9349205-G to rs9349205-A. The bottom dotted arrow shows a more recent mutation (after modern humans left Africa) from rs112233623-C to rs112233623-T. The corresponding effects of these mutations and derived haplotypes on gene expression and function are also illustrated schematically.

SMAD3 and creates a binding site for the repressive transcription factor GATA1. The consequences are reduced expression of *CCND3* and, as owing to the lower level of cyclin D3 protein, inhibition of the G1–S transition in RBC precursors. The steady-state number of RBCs therefore decreases, whereas the volume of each cell increases. At the same time, HbA2—and, to a lesser extent, HbF—increases, highlighting a specific window during terminal erythropoiesis in which the shortening of the cell cycle reduces (albeit to varying degrees) the typical increase in the δ-globin/β-globin and γ-globin/β-globin synthesis ratios as erythroblasts complete their maturation[39,40].

We infer that rs112233623-T is likely to have arisen on a background haplotype that also included another variant, rs9349205-A, which, when not co-inherited with rs112233623-T, has an opposite though weaker effect on *CCND3* expression and haematological phenotypes. Capitalizing on the unique genetic architecture of the *CCND3* gene region in the Sardinian population, we show that when the two variants are co-inherited, the smaller effect of rs9349205-A is overwhelmed by the stronger effect of rs112233623-T, which reduces *CCND3* expression and has a knockout-like phenotypic effect on RBC traits (Fig. 4). The findings at the population level are also supported at the molecular level, and specifically in transfection experiments of erythroid cell lines using cloned DNA fragments with the two allelic variants together, in which we observed a marked reduction in enhancer activity, compared with the increase in activity reported for the derived A allele of rs9349205 alone. There is a probable mechanistic basis for the observed effects. Previous results have shown that rs9349205-A enhances *CCND3* expression during erythroid differentiation by fostering an increase of chromatin accessibility at the enhancer[17]. On the basis of our finding that SMAD3 can no longer bind to the binding site after it has been altered by rs112233623-T, any effect of rs9349205-A on chromatin accessibility would be irrelevant on an rs112233623-T background.

The rise in the frequency of rs112233623-T in the Sardinian population seems to have occurred rapidly—sufficiently rapidly to suggest that positive selection of rs112233623-T occurred in populations with Sardinian ancestry (Extended Data Figs. 7 and 8). Consistent with the crucial function of cyclin D3 in RBC precursors, we considered whether past endemic malaria in Sardinia was the primary driver of this positive selection. Using cohort-based association studies to test for a possible protective effect of rs112233623-T against malaria infection is not possible, because the variant is absent in areas of the world where malaria is still endemic, and malaria was eradicated in Sardinia, where the allele is common, in the early 1950s. Nonetheless, we were able to identify an antimalarial effect in vitro by infecting cultured erythrocytes from Sardinian individuals with varying rs112233623 genotypes, as well as other variants of interest, with *P. falciparum*. This was made feasible by the high frequency of the T allele in Sardinia and the availability of a bioresource of thousands of genetically profiled individuals. We found that in RBCs of individuals homozygous for the *CCND3* DoE rs112233623-T variant, there was a substantial inhibition of *Plasmodium* development, ultimately leading to parasite death. This was correlated with a substantial increase in the levels of ROS in these cells, comparable with that seen in individuals hemizygous for G6PD deficiency (Fig. 3).

The strong inhibitory effect of the rs112233623-T variant on parasitaemia in vitro, together with evidence for its positive selection in a population with a long history of endemic malaria, supports the hypothesis that this variant is protective against severe malaria, which is inherently associated with high levels of parasitaemia. This predicted defence against severe malaria by increased ROS production taps into the same pathway as G6PD deficiency, and is one of the main proposed mechanisms of action of commonly used antimalarial drugs such as quinolines, atovaquone and artemisinin and its derivatives[41,42]. All of these drugs have been shown to increase oxidative stress to levels that compromise malarial parasites, resulting in damage to cellular components, including lipids, proteins and DNA, and ultimately to cell death via apoptosis or necrosis. In fact, documented increases of ROS in individuals with the sickle-cell trait or β-thalassaemia[43,44] raise the possibility that oxidative stress, which has been regarded as a feature of pathophysiology of other RBC disorders, is also involved in their mediation of resistance to malaria.

In addition to the ROS effects, we propose that the antimalarial action of reduced *CCND3* expression is also mediated by its effects on the cell cycle of erythroblasts in the bone marrow, the preferential niche for the commitment and development of *P. falciparum* gametocytes. Previous research has shown that the maturation time of gametocytes in erythroblasts (around 10 days) is longer than the time that it normally takes for erythroblasts to develop from orthochromatophilic stages to reticulocytes in vitro (about 3–4 days)[45]. A prolonged period of erythroblast development is promoted by *Plasmodium* to ensure its gametocytogenesis through the release of factors that affect both infected and uninfected erythroblasts[45], contributing to the ineffective erythropoiesis and anaemia observed in malaria. The reported lower number of erythroblast divisions when cyclin D3 expression is reduced[16] might thus counteract the delay in the differentiation of erythroblasts that is normally imposed and needed by *P. falciparum*.

A genetically inherited reduction in *CCND3* expression could thus prevent severe anaemia—a major cause of death—in two ways: first, by alleviating peripheral haemolysis through increased ROS levels; and second, by ameliorating ineffective erythropoiesis by reducing cell divisions during terminal erythropoiesis in the bone marrow. This latter effect could also prevent the sexual maturation of the parasite, thereby hindering disease transmission.

The decrease of *CCND3* expression might have a similar protective role in *P. vivax* infections in the bone marrow[46], and a comparable protective effect of increased ROS production in peripheral blood during *P. vivax* infections is consistent with the reported protection against severe *P. vivax* infection conferred by the *G6PD*-Med variant[35] (rs5030868-A).

In terms of the therapeutic relevance of our findings, because *CCND3* is expressed during erythroblast maturation but not in mature erythrocytes, the antimalarial effects of *CCND3* inhibition are likely to be related mainly to its action in the bone marrow. These include immediate cell-cycle effects—moderating ineffective erythropoiesis—and delayed metabolic effects against peripheral haemolysis through increased ROS in erythrocytes, which will depend on the daily replacement rate of erythrocytes from the bone marrow. However, compared with the steady state, erythrocyte turnover is also expected to be significantly increased by inhibition of *CCND3* in the bone marrow and by the

erythropoietic stress caused by haemolysis during malaria infection. Our findings thus provide a rationale for inhibiting cyclin D3 as early as possible during parasite infection. Cyclin D3 inhibition, including in combination therapies with artemisinin, could thus be used to help treat malaria and reduce its transmission and disease burden, which remain a major health problem in many regions of the world[47].

In addition, anti-cyclin-D3 therapies could also help to control Burkitt's lymphoma—an aggressive non-Hodgkin B cell lymphoma that has been clearly linked to malaria in highly malaria-endemic areas of Africa[48]—both indirectly, by controlling the spread of malaria, and more directly, by counteracting the effect of known somatic mutations increasing the activity of *CCND3*, which have been shown to promote its pathogenesis[49].

The requirement for cyclin D3 in the development of immune cells[50] suggests that caution is needed in interventions with antagonizing therapies, but the possibility that cyclin D3 can tolerate some degree of inhibition for a defined period of time is supported by data showing that even *Ccnd3*-knockout mice are vital and develop into adulthood[16].

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

# Methods

## Cell lines and primary cell cultures

All cells were cultured under 5% $CO_2$ at 37 °C. HEK-293T and COS7 cells (ATCC) were cultured in Dulbecco's modified Eagle's medium (DMEM) with 10% heat-inactivated fetal bovine serum (FBS) (Gibco), 1% L-glutamine and 1% penicillin–streptomycin (Sigma). HUDEP-2 cells (provided by Y. Nakamura and R. Kurita) were cultured as described[21].

CD34+ haematopoietic stem and progenitor cells were isolated from peripheral blood from donors and sorted using CD34+ MACS (Miltenyi) according to the manufacturer's protocol. The cells were differentiated using a three-phase erythroid in vitro differentiation protocol as previously described[52,53]. In brief, the cells were expanded in StemSpan medium (Stem Cell Technologies) supplemented with 100 ng ml$^{-1}$ stem cell factor (SCF) (PeproTech), 100 ng ml$^{-1}$ Flt-3 (PeproTech), 100 ng ml$^{-1}$ thrombopoietin (TPO) (PeproTech) and 20 ng ml$^{-1}$ IL-3 (PeproTech) for a total of six days.

After six days, the cells were reseeded in phase I differentiation medium (days 1–7) consisting of Iscove's modified Dulbecco's medium (IMDM) (Gibco) supplemented with 2% human plasma (Stem Cell Technologies), 3% human serum (Sigma), 3 units ml$^{-1}$ heparin (Sigma), 10 μg ml$^{-1}$ insulin (Sigma), 3 units ml$^{-1}$ erythropoietin (Janssen-Cilag), 10 ng ml$^{-1}$ SCF (PeproTech), 1 ng ml$^{-1}$ IL-3 (PeproTech) and 200 μg ml$^{-1}$ holo-transferrin (Sigma), at a density of around $0.5 \times 10^6$–$1 \times 10^6$ cells per ml for a total of seven days. Phase II medium (days 8–12) included the same cytokines except for IL-3. During phase III (days 13 and beyond) holo-transferrin was increased to 1 mg ml$^{-1}$ and SCF was withdrawn. Erythroid differentiation was monitored by fluorescence-activated cell sorting (FACS) analysis and May–Grünwald–Giemsa (RAL Diagnostics) staining of cytospin slides. Images were captured from cytospin slides using a PrimoStar 3 microscope (Zeiss) coupled with an Axiocam 208 camera (Zeiss). For growth-curve analysis, cultured erythrocyte progenitor cells were seeded in the same amounts and then counted at the indicated days both by haemocytometer (Tali, Invitrogen) and by microscope with trypan blue staining (Invitrogen) to exclude dead cells. To test for TGFβ stimulation, HUDEP-2 cells were stimulated with 5 ng ml$^{-1}$ TGFβ (Sigma) and sampled for *SMAD3* and *CCND3* mRNA expression profiling at various time points after the treatment (0 min, 60 min, 120 min, 240 min and 300 min). The inhibitory effects on this pathway were evaluated with 20 μM SB-505124 (Sigma) in 0.2% DMSO, as previously described[54]. All of the cells used tested negative for mycoplasma contamination.

## Human blood samples

Whole-blood samples were collected from volunteers enrolled in the SardiNIA general population cohort[14], at the project centre in Lanusei (Ogliastra, Sardinia). All participants gave informed consent for the study protocols, which were approved by the Sardinian Regional Ethics Committee (protocol 2171/CE). These participants have agreed in principle to be recalled for functional follow-up experiments based on their genotypes at loci of interest, such as those reported in this study. The samples collected for the experiments reported in this study were specifically selected to ensure representation of homozygous genotypes at the loci of interest, rs112233623 and rs9349205 close to the *CCND3* gene. For the malaria experiments, individuals who were carriers of the most frequent allelic variants in Sardinia that have previously been associated with protection against malaria were excluded from the collection. These variants include the β⁰39 thalassaemia mutation (rs11549407-A) in the *HBB* gene (frequency = 0.051), the 3.7-kb α-thalassaemia deletion (NG_000006.1:g.34164_37967del3804; frequency = 0.14) and the Mediterranean deficiency variant (rs5030868-A) in the *G6PD* gene (frequency = 0.083), with the exception of a small group of samples that were selected to be hemizygous for the *G6PD* Mediterranean deficiency variant. We also tried to ensure a balance in terms of sex in the various genotype groups being compared (Supplementary Information).

Whole blood was collected using heparin vacutainer tubes (BD Biosciences) for gene-expression and cell-cycle experiments based on the generation of erythrocyte progenitor cells, and acid–citrate–dextrose (ACD) vacutainer tubes (BD Biosciences) for *P. falciparum* infection experiments. The blood was kept at ambient temperature in vitro before processing for the specific experiments, all of which began on the same day. Further details about sample collection are provided in the Supplementary Information, and the overlap of donors across the functional and *P. falciparum* growth experiments is detailed in Supplementary Table 5.

## Flow-cytometry staining

All staining steps were performed at 4 °C, with efforts made to minimize sample exposure to ambient light. For flow-cytometry analysis, $3 \times 10^5$ in-vitro-differentiated erythroid progenitor cells were washed twice in PBS and resuspended in staining buffer (1% FBS in PBS). Cells were surface-stained for 20 min with the following antibodies: 1:50 PC7-conjugated CD235a (glycophorin A, Beckman Coulter), 1:100 PE-conjugated anti-Band3 (IBGRL) and 1:20 APC-conjugated anti-CD49d (integrin α4 chain, Beckman Coulter). After washing, cells were resuspended in 500 μl FACSFlow (BD Biosciences). For the single-cell population, CD235a–PC7 versus FSC-A was used to gate on CD235a–PC7-positive cells. Erythroblast maturation was evaluated by flow cytometry through the simultaneous detection of surface markers integrin α4, glycophorin A (GPA) and Band 3, using anti-CD49d–APC, anti-CD235a–PC7 and anti-Band3–PE monoclonal antibodies, respectively. Data were collected using BD FACS Accuri C6 Plus (BD Biosciences). For cell-cycle analysis at each time point, cultured erythroid progenitor cells were collected, washed in PBS, resuspended in permeabilizing solution and incubated at 37 °C for 30 min. Propidium iodide (BD Biosciences) was then added to a final concentration of 10 μg ml$^{-1}$, and cells were incubated for an additional 10 min at room temperature before analysis. Cells were acquired on a FACSCanto (BD Biosciences) coupled with FACSDiva software (v.6.1.3).

The data analysis was performed using FlowJo software (v.10.10.0).

## Luciferase reporter analysis

Firefly luciferase reporter constructs (pGL4.23, Promega) were generated by cloning the variant of interest centred in 372 nucleotides of genomic context upstream of the minimal promoter. Expression plasmids were prepared by subcloning human cDNA clones for each of *SMAD3*, *p300* (clone provided by D. Barilà) and *GATA1* (clone provided by F. Grosveld) into the pEF5-HA vector (vector provided by N. Bottini). The expression plasmid pCMV6-Entry FOG1 was purchased from Origene. To avoid erroneous normalization of transfection efficiency, we used a Renilla mutant control plasmid (pRLSV40-mut-GATA) in which the GATA response elements had been mutagenized, thus rendering pRL-SV40 unresponsive to GATA stimulation. Primer sequences are listed in Supplementary Table 4.

Firefly constructs (2,000 ng) were cotransfected with the pRLSV40-mut-GATA construct (300 ng) alone or with the expression plasmid (1,000 ng) into 625,000 HUDEP-2 cells using the Neon electroporation system (Invitrogen, 1200 V, 20 msec, two pulses). After 48 h, luciferase activity was measured using a Synergy 2 plate reader (Biotek) with a Dual-Glo luciferase assay system (Promega) according to the manufacturer's instructions. Firefly luciferase levels were normalized to the pRLSV40-mut-GATA Renilla luciferase activity for each sample, and all values were then plotted relative to the empty pGL4 vector.

## EMSAs

EMSAs were performed as previously described[55]. Radiolabelled double-stranded oligonucleotides were prepared by annealing oligonucleotides containing the rs112233623 WT (C) or the derived (T) allelic variants. Small-scale nuclear extracts were prepared as previously

described[56] from COS7 cells that had been treated with TGFβ (5 ng ml⁻¹ for 16 h) as previously described[57] and then transfected with the eukaryotic expression vectors pEF5HA-SMAD3 or pEF5HA-GATA1, or empty pEF5HA vector as a control. The presence and correct size of the proteins of interest in all extracts were verified in western blot assays. For supershift experiments, EMSA reactions were preincubated for 30 min on ice with specific antibody (anti-HA.1, BioLegend; anti-GATA1, Santa Cruz Biotechnology). Competitor cold oligonucleotides were added in 25×, 50× and 100× molar excess. The protein–DNA interaction products were resolved on a 5% non-denaturing acrylamide gel (37.5:1 acrylamide:bis-acrylamide, 0.25× TBE). Representative full-scan blots are provided in the Supplementary Information and in the source data. Primer sequences are listed in Supplementary Table 4.

## ChIP assay

ChIP was performed by a SimpleChIP Plus Enzymatic Chromatin IP Kit (Magnetic Beads, Cell Signaling Technology, 9005S). In brief, cells ($4 \times 10^6$ per immunoprecipitation) were fixed with 1% formaldehyde for 10 min at room temperature, and the reaction was quenched with glycine at a final concentration of 125 mM. Nuclei were prepared, and chromatin was incubated with micrococcal nuclease at 37 °C for 15 min, which was followed by an appropriate amount of sonication. Immunoprecipitations were performed using 10 μg of an antibody against SMAD3 or GATA1, or 1 μg with nonspecific rabbit IgG as negative control (2729S Cell Signaling Technology), according to the manufacturer's instructions, at 4 °C for 12–16 h. The next day, the immunocomplexes were rotationally incubated with 30 μl of ChIP-Grade Protein G Magnetic Beads for two hours at 4 °C and then washed three times using low-salt wash buffer and once with high-salt wash buffer at 4 °C for 5 min per wash. Chromatin was eluted by ChIP elution buffer for 30 min at 65 °C with gentle vortex mixing (1,200 rpm) and cross-links were reversed by treatment with 5 M NaCl and proteinase K for two hours at 65 °C. ChIP DNA was purified and subsequently quantified by qPCR using primers designed to amplify a region surrounding rs112233623 in the *CCND3* enhancer and SimpleChIP Human α Satellite Repeat Primers (4486, Cell Signaling Technology). Data were finally presented as percentages of the input DNA. Primer sequences are listed in Supplementary Table 4.

## Genome editing by CRISPR–Cas9

Single guide RNAs (sgRNAs) targeting *SMAD3* and *SMAD2* were developed using a CRISPR design tool kindly provided by the Zhang laboratory (Broad Institute, MIT) (see Supplementary Table 4 for sgRNA sequences). The annealed oligos were ligated into the BsmbI-digested LentiViral (LV) construct LentiCRISPRV2 (Addgene). Lentiviral vectors were obtained by transfecting HEK-293T cells with 10 μg LentiCRISPRV2 (expressing both sgRNA and Cas9 or Cas9 only), 7.5 μg psPAX2, 2.5 μg pMD2G-VSVG and 1.25 μM pRSV-REV, using polyethylenimine (PEI; ref. 58) (Polysciences). After 72 h, viral particles in the supernatants were collected by centrifugation at 20,000 rpm for two hours at 4 °C. HUDEP-2 cells were then transduced with CRISPR–Cas9 by centrifugation at 2,800 rpm for 90 min at 30 °C with virus at a multiplicity of infection of 40. Forty-eight hours after transduction, cells were positively selected with the addition of 1 μg ml⁻¹ of puromycin to the medium. To validate biallelic CRISPR indel detection at the genomic target site, DNA was extracted from puromycin-selected HUDEP-2 cells with the QIAamp DNA Mini Kit (QIAGEN), PCR-amplified (primer sequences are listed in Supplementary Table 4) and sequenced by the Sanger method in a 3130xl Genetic Analyzer (Applied Biosystems). The percentage of indels was calculated using TIDE v.3.3.0 (https://tide.nki.nl/)[59].

## RT–qPCR analysis

RNA was extracted from cells using the RNeasy extraction kit (QIAGEN) according to the manufacturer's instructions. First-strand cDNA synthesis was performed with the SuperScript III kit (Thermo Fisher Scientific) for equivalent amounts of starting RNA from all samples. The cDNA was analysed on an ABI 7700 machine using SYBR green master mix (Applied Biosystems), and normalized to *GAPDH* or *B2M* (β2-microglobulin) as an internal control. All samples were done in triplicate. PCR cycle conditions were 95 °C for 5 min, and 40 cycles of 95 °C for 10 s, 54 °C for 10 s, 72 °C for 15 s. Analyses of Ct values were performed as previously described[60]. PCR primer pairs are listed in Supplementary Table 4.

## Analysis of RNA-seq data from public datasets

Raw RNA-seq data (FASTQ files) of An et al.[19] and Ludwig et al.[20] were downloaded from the NCBI Sequence Read Archive (SRA) (https://www.ncbi.nlm.nih.gov/sra). RNA-seq reads were aligned using STAR software (v.2.7.10b)[61] against a transcriptome reference generated by RSEM software (v.1.3.1)[62]. The transcriptome reference was built from the hs37d5 human genome assembly (https://ftp.1000genomes.ebi.ac.uk/vol1/ftp/technical/reference/phase2_reference_assembly_sequence/) and GENCODE v.14 gene annotation (https://www.gencodegenes.org) using the rsem-prepare-reference function of RSEM. Normalized gene-expression levels (fragments per kilobase of transcript per million mapped reads; FPKM) were computed with the rsem-calculate-expression function of RSEM. Box plots were generated with R (v.3.5.3).

## Western blotting

Western blotting was done following standard protocols. Proteins were collected in RIPA buffer (Millipore) supplemented with 2 mM PMSF (Sigma), and cOmplete Mini Protease Inhibitor Cocktail (Roche). Protein yield was determined with the Bradford protein assay (Bio-Rad), and equal amounts of total protein were separated by SDS–PAGE (Thermo Fisher Scientific). Proteins were transferred onto a polyvinylidene difluoride (PVDF; Amersham Biosciences) membrane in a wet transfer system (Bio-Rad). Membranes were then incubated with HRP-conjugated secondary antibodies (Santa Cruz Biotechnology), developed with an ECL system (Amersham Biosciences) according to the manufacturer's instructions and visualized using photographic film. The ratio between cyclin D3 and β-actin was calculated using Image J1.52a (https://imagej.net/ij/). Representative full-scan blots are provided in the Supplementary Information and in the source data.

## Reagents and antibodies

Recombinant TGFβ and TGFβ type I receptor inhibitor SB431542 were purchased from Sigma-Aldrich (T7039 and S4696). The following antibodies were used: PC7-conjugated anti-human CD235a (GPA–PC7, 1:50) supplied by Beckman Coulter, clone 11E4B-7-6, A71564; PE-conjugated anti-Band3 (Band 3–PE, 1:100) supplied by IBGRL, clone BRIC 6, 9439PE; APC-conjugated anti-CD49d (integrin α4–APC, 1:20) supplied by Beckman Coulter, clone HP2/1, B01682; anti-human GATA1 (GATA1, 1:10) supplied by Santa Cruz Biotechnology, clone N1, sc-266X; anti-influenza haemagglutinin epitope tag HA.11 (SMAD3–HA, 1:10) supplied by BioLegend, clone 16B12, MMS-101P; anti-human cyclin D3 (cyclin D3, 1:250) supplied by Santa Cruz Biotechnology, clone D-7, sc-6283; anti-human SMAD2 (SMAD2, 1:500) supplied by Santa Cruz Biotechnology, clone S-20, sc-6200; anti-β-actin (β-actin, 1:1,000) supplied by Santa Cruz Biotechnology, clone C4, sc-47778; anti-cyclin D3 (cyclin D3, 1:2,000) supplied by Cell Signaling Technology, clone DCS22, 2936S; rabbit anti-mouse IgG–HRP (1:20,000) supplied by Santa Cruz Biotechnology, sc2005; goat anti-rabbit IgG–HRP (1:20,000) supplied by Santa Cruz Biotechnology, sc2004; rabbit anti-goat IgG–HRP (1:20,000) supplied by Santa Cruz Biotechnology, sc-2768; anti-human SMAD3 (SMAD3, 10 μg in 100 μl for immunoprecipitation and 1:1,000 for western blot) supplied by Cell Signaling Technology, clone C67H9, 9523; anti-human GATA1 (GATA1, 10 μg in 100 μl) supplied by Abcam, ab11852; and normal rabbit IgG, (IgG, 1 μg in 100 μl) supplied by Cell Signaling Technology, 2729S.

### Plasmodium falciparum culture

In vitro studies were done using the genetically stable reference FUP strain (mycoplasma-free) according to previously reported protocols[63]. The FUP strain of *P. falciparum* selected for our in vitro study (provided by E. Schwarzer) has been used extensively in both in vitro and in vivo studies to investigate various aspects of malaria infection, including RBC invasion, growth and development in the host. Compared with other strains, the FUP strain tends to maintain its genetic integrity during in vitro culture, making it a reliable choice for long-term studies[64]. To evaluate the reproducibility of our findings across genetically distinct parasite backgrounds, we performed additional experiments using the 3D7 (African) and DD2 (Southeast Asian) isolates.

### Infection of RBCs from genotyped individuals

Fresh genotyped venous blood (about 25 ml) was collected from genotyped donors into ACD vacutainer tubes (BD Biosciences) and immediately processed. RBCs were separated from plasma and leukocytes by three washings in RPMI 1640 containing 25 mM HEPES (Gibco) and supplemented with 2 mM glutamine, 20 mM glucose, 27 µg ml$^{-1}$ hypoxanthine and 32 µg ml$^{-1}$ gentamicin (Sigma) (pH 7.2). The washed genotyped RBCs were used to maintain the parasite in vitro cultures in the same medium. Parasite cultures were synchronized as described previously[65]. Cultures were incubated at 37 °C under a 5% $O_2$, 5% $CO_2$, 90% $N_2$ (v/v) atmosphere as previously described[66].

To synchronize the genotyped cultures by parasite stage, schizonts (mature forms of *Plasmodium*) were collected from the 40–60% interface after passing a mixed-stage culture through a discontinuous Percoll mannitol density gradient by centrifugation at 5,000$g$ for 30 min. The ring-stage parasites (early parasite stage) from the bottom fraction were also kept for further synchronous culturing. To start plasmodium cultures, density-isolated schizonts were added to erythrocytes in growth medium to a density of 2.5% parasites per erythrocyte.

All experiments involving in vitro malaria infections were performed in parallel with two initial haematocrit settings: 2% and 1%. This parallel experimental design provided the option to select between the two settings for subsequent short- and long-term parasite culture analysis, circumventing the complex and unpredictable process of repeating experiments from a small pool of donors with the relevant multilocus genotype combination, who are difficult to resample.

Samples with an initial haematocrit of 2% used 240 µl of RBCs with 2.5% schizont parasitaemia. We added 240 µl of packed erythrocytes every 48 h, coupled with replacement of the growth medium. This led to a gradual increase in haematocrit up to approximately 6% by the third cycle.

For 1% initial haematocrit samples, we used 2.5% schizont parasitaemia to infect 120 µl of packed erythrocytes and the same amount of RBCs was added to the cultures every 48 h, alongside the replacement of the growth medium. In this case, the final haematocrit level reached approximately 3% by the third cycle.

These two settings were used for all replicates at each time point (24, 48, 72, 96, 120 and 144 hpi). Because analysis of long-term cultures was performed to allow a wider range of observations, experiments were selected that began with an initial haematocrit level of 1%.

This is because the haematocrit level affects the glucose level significantly: as the haematocrit increases, so does glucose consumption, resulting in glucose depletion; in turn, the glucose level affects parasitaemia significantly, because the parasites rely heavily on glycolysis for energy and survival. Especially in the late cycles of long-term cultures, when the haematocrit level increases proportionally with the progressive addition of erythrocytes, we expect that parasitaemia will be less (negatively) affected by glucose consumption by both uninfected and infected erythrocytes at the selected lower haematocrit setting.

For each condition, two thin blood smears were independently counted by three trained microscopists in a blinded manner according to the latest guidelines of the World Health Organization (Malaria Microscopy Quality Assurance Manual). The resulting six counts were averaged to obtain a single arithmetic mean per condition.

Total parasitaemia, stages of morphology, parasite viability and inhibition rate were determined using Diff-Quick (RAL Diagnostics)-stained thin blood smears and light microscopy (Carl Zeiss Microscope Primo Star, objective lenses: 20×/0.50 PlanF and 100×/1.25 Oil Plan-ACHROMAT). The percentage of total parasitaemia was determined as follows: (number of parasitized erythrocytes/total number of erythrocytes) × 100. For each sample, at least 5,000 cells were counted. Damaged and pyknotic forms were counted separately and excluded from the total parasitaemia count.

The percentage of ring parasitaemia was determined as follows: (number of rings (single and multiple)/total number of infected RBCs) × 100.

The percentage of trophozoite parasitaemia was determined as follows: (number of trophozoites (single and multiple)/total number of infected RBCs) × 100.

The percentage of schizont parasitaemia was determined as follows: (number of schizonts (most mature form of *P. falciparum* asexual stages)/total number of infected RBCs) × 100.

The percentage of dead parasites was determined as follows: (number of pyknotic forms (blue dots inside RBCs)/total number of infected RBCs) × 100.

The percentage of parasite growth inhibition was calculated as follows: (1 − parasitaemia value)/(parasitaemia WT/WT|WT/WT average value) × 100.

The experiments were done in five batches to ensure, as far as possible, that individuals recruited for each experimental batch had a similar proportion of the different genotypes, especially IoE and DoE, to control for potential batch effects (Supplementary Information). A table listing the batch assignment for each donor used in the infection experiments is provided in the source data. We also ensured that all assays were performed under strictly controlled, standardized and homogeneous conditions to maintain consistency and comparability between batches, using exactly the same experimental protocols, including identical reagent batches, instrument settings and environmental parameters. This approach ensured the validity, accuracy and reproducibility of the results and minimized the variability associated with sample processing over time.

### LC–MS measurements for ROS detection

Plasma was separated from whole blood by centrifugation at 200$g$ for 5 min at 4 °C. White blood cells and pellets were thereby removed and the collected RBCs were transferred into a fresh microcentrifuge tube. The RBCs were then diluted 100-fold in PBS and probed with 20 µM coumarin boronic acid (Cayman Chemical). Before LC–MS analysis, 100 µl of methanol was added to the samples, which were then centrifuged at 5,000$g$ for 30 min at 4 °C. LC–MS analysis was performed on an UltiMate 3000 UPLC system (Thermo Fisher Scientific) as previously described[67]. The limit of quantitation was calculated from calibration curves. For hydrogen peroxide species, the limit of detection was set to 11 nM, and the limit of quantification was set to 37 nM.

### Statistical analyses

**Targeted genetic association.** Association signals for HbA2 and HbF levels (g dl$^{-1}$), as well as MCV, RBC and MCH, were detected for the *CCND3* gene region in a cohort of up to 6,824 samples from the SardiNIA cohort, as previously reported[1] in a subset of 6,305 individuals from the same cohort in an analysis of haemoglobin levels.

Analyses used a genetic map based on 7,928 genotyped samples, with the OmniExpress Illumina array and iScan system using the Infinium HTS protocol, and imputed on a genome-wide scale using a Sardinian sequence-based reference panel of 3,514 individuals, as previously described[51]. The target region of ±0.5 Mb around the gene was analysed

using a linear mixed model that accounts for cryptic relatedness and population stratification, implemented in the q.emmax function of the EPACTS program (http://genome.sph.umich.edu/wiki/EPACTS). Before the analysis, phenotype levels were normalized with inverse-normal transformation and adjusted for covariates of sex, age and age$^2$ and for the presence of at least one of the 3 $\beta^0$ mutations ($\beta^0$39 (rs11549407), *HBB*:c.20delA (rs63749819) and *HBB*:c.315+1G>A (rs33945777)), all directly genotyped or imputed. Moreover, HbF outliers with values higher than 5% were not considered in the association analysis, as previously reported[1]. An association signal was considered significant at a *P*-value threshold of $6.9 \times 10^{-9}$ (ref. 51). Conditional analyses were performed for each trait by adding the most associated SNP as a covariate to the model adjusted for age, age$^2$, sex and $\beta^0$: no independent variant was found at the locus. To detect a causal variant, or causal variants, in the associated region, fine mapping was done by applying credible set analyses to the traits showing a significant association signal in the region (HbA2, MCV, MCH and RBC) using FINEMAP v.1.4.2 software; the --cond option and settings for one causal variant (--n-causal-snps 1) and for a maximum of ten causal variants (--n-causal-snps 10) for each association profile were used, as previously described[68]. Fine-mapping analyses converged for all tested traits after one causal SNP was detected, finding a single credible set at the 95% summed posterior probability. Variants in the credible set were then annotated for likely deleterious effect using the CADD score[69] (using Ensembl GRCh37 release 111, January 2024).

**Population genetic analyses.** To investigate the hypothesis that the high allele frequency of rs112233623 in the Sardinian population results from positive selection rather than from genetic drift, standard metrics based on allele frequency ($F_{ST}$) and haplotype differentiation patterns (iHS and xp-EHH) were calculated as previously described[70]. In particular, a subset of 1,577 unrelated samples belonging to the SardiNIA whole-genome sequence reference panel[51] was studied. This subset was defined by computing the genome-wide proportion of pairwise identity by descent (IBD) ($\pi$) on a random set of one million SNPs with a MAF > 0.05 in the 1KG population sample; for each pair of individuals with $\pi$ > 0.05, the offspring was preferentially removed if in a trio; otherwise, the individual with the largest summed value of $\pi$ across all other relationships with $\pi$ > 0.05 was removed. Analyses on SardiNIA were then compared with those on 500 EUR samples included in 1KG phase 3 (ref. 18). To estimate the $F_{ST}$ statistic, the Weir–Cockerham formula implemented in VCFtools v.0.1.12b (https://vcftools.github.io/index.html) was used to compare SardiNIA with 1KG EUR samples. For the haplotype-based analyses, variants with a MAF < 0.01 and a Hardy–Weinberg equilibrium (HWE) test $P < 10^{-4}$ in Sardinian samples were first removed; this conservative filter facilitated better reconstruction of haplotypes by avoiding genotyping errors. The extended haplotype homozygosity (EHH) and the unstandardized iHS and xp-EHH values were then calculated using selscan[71]. Standardized statistics were obtained using the norm tool included in selscan. Finally, to establish a statistical benchmark for each test statistic, we constructed an empirical background distribution based on randomly selected variants from sequenced SardiNIA genomes that matched our target variant rs112233623 for three features: MAF ± 2% in SardiNIA; a measure of background selection (B score (ref. 72) ± 50 units); and local recombination rate in a 50-kb region around the variant (±0.5 cM per Mb; ref. 70) for a total of 13,134 variants. To exclude background variants in LD, variants with a correlation ($r^2$) ≥ 0.1 were then filtered using the pruning procedure implemented in plink (https://www.cog-genomics.org/plink/1.9/), to obtain 1,075 variants that were used for the empirical background distribution used to calculate the genomic percentile values. Percentiles are calculated for standardized values for $F_{ST}$ and xp-EHH tests, and for absolute standardized values for iHS. A high percentile value (such as 99%) suggests that for the statistic calculated, the variant shows more extreme signatures of positive selection than do 99% of the genomic background variants.

**Coalescent-based analyses of recent positive selection. Description of dataset and preprocessing steps.** For the purposes of coalescent-based analyses, we used the whole-genome sequencing dataset described previously[73,74] (*n* = 3,514). This dataset includes individuals from the SardiNIA Project[51], as well as from two case–control studies conducted in Sardinia. Related individuals were pruned from this dataset on the basis of the proportion of pairwise shared IBD, as described[73]. Given the computational demands, in a first step we sampled 100 unrelated individuals at random, followed by a second step in which we randomly selected a larger sample of 300 unrelated individuals for coalescent-based analyses.

**Construction of genealogies using Relate.** Genealogies were constructed using the Relate method (v.1.1.9), as described previously[75]. In brief, this method approximates the ancestral recombination graph by inferring a series of marginal trees along the genome. For all analyses using Relate, we followed the pipeline described in the online documentation (https://myersgroup.github.io/relate/). First, the ConvertFromVcf module was used to convert files from VCF format to the haps/sample format used by Relate, and the provided script (PrepareInputFiles.sh) was used to prepare the data. Then Relate was run with the mutation rate set to $1.25 \times 10^{-8}$ and the haploid effective population size set to 30,000 to generate anc/mut files. The script EstimatePopulationSize.sh was then used to estimate coalescence rates, which were then used to re-estimate branch lengths in the anc/mut files with the script ReEstimateBranchLengths.sh. Marginal trees were visualized using the provided script TreeViewMutation.sh.

**Genealogies for the sampled Sardinian individuals.** We first constructed a genealogy for the 100 sampled individuals described in the 'Description of dataset and preprocessing steps' subsection above, using data from chromosome 6. As a precursor to using CLUES to infer selection coefficients across the genome, we repeated the Relate analysis using data from each autosome (1–22). We then used the script SampleBranchLengths.sh to sample branch lengths using Markov chain Monte Carlo (MCMC) for each variant in the SNP set of the 1,075 selected matching variants. We used the coalescence rates inferred by Relate and these sampled branch lengths as input and set the mutation rate to $1.25 \times 10^{-8}$ per bp per generation. This step was repeated for each variant for which we estimated a selection coefficient (see below).

**Multi-population genealogy.** We also constructed a genealogy using Relate for a merged dataset including the 100 individuals described in the 'Description of dataset and preprocessing steps' subsection above, 107 Tuscan individuals from Italy[18] and 108 Yoruba individuals from Ibadan, Nigeria[18]. We followed the same procedure described in the 'Construction of genealogies using Relate' subsection above to construct and visualize this genealogy for variants on chromosome 6, including rs112233623 and rs9349205.

**Estimation of allele frequency trajectory and selection coefficient for rs112233623 using CLUES.** We used the CLUES method[31] to jointly infer the allele frequency trajectory and selection coefficient for the DoE variant (rs112233623). To do so, we followed the steps described above to run Relate tree inference for the sampled individuals described in the 'Description of dataset and preprocessing steps' subsection above and sample branch lengths from the trees. We then used the CLUES script inference.py to run CLUES with the sampled branch lengths (--times) and coalescence rates (--coal) outputted by Relate as input to the script. The CLUES-provided script plot_traj.py was then used to visualize the allele frequency trajectory.

**Empirical distribution of selection coefficient estimates.** To contextualize the results of the selection inference procedure, we repeated the CLUES analysis for the 1,075 SNPs selected to match by MAF, B score (background selection) and recombination rate with that of the DoE variant rs112233623. The HLA region was excluded from this SNP set. Of the 1,075 matched SNPs, 1,035 and 1,054 mapped onto the marginal tree inferred by Relate using 100 and 300 samples, respectively, and the remaining variants were excluded from further analysis.

**Statistical analyses in experiments.** On the basis of data distributions and sample sizes, parametric *t*-tests or non-parametric Wilcoxon–Mann–Whitney tests were applied to compare means or medians, respectively, and to establish statistical significance. To test a specific hypothesis, the one-tailed alternative-hypothesis test was used. Statistical analyses were performed using R v.3.5.3 (2019-03-11).

Correlations were calculated using the Pearson's product moment correlation coefficient and *P* values were computed under the two-side hypothesis (cor.test function in R).

### Reporting summary

Further information on research design is available in the Nature Portfolio Reporting Summary linked to this article.

### Data availability

Data are provided as source data or Supplementary Information, or are available at https://figshare.com/s/d89f8f9f90d3a76eccb9 (ref. 38). Population genetics analyses in non-Sardinian populations have been done using the 1KG phase 3 data, available at https://ftp.1000genomes.ebi.ac.uk/vol1/ftp/release/20130502/. Public GWAS summary statistics are available at https://www.ebi.ac.uk/gwas/home. RNA-seq datasets analysed in this study were obtained from publicly available repositories. Raw sequencing data from Ludwig et al.[20] are available in the NCBI SRA under accession SRP150349, and RNA-seq data from An et al.[19] are accessible under accession SRP035312. The human genome assembly hs37d5, used to generate the transcriptome reference for expression quantification, is available at https://ftp.1000genomes.ebi.ac.uk/vol1/ftp/technical/reference/phase2_reference_assembly_sequence/. The gene annotation (GENCODE v.14) used to generate the transcriptome reference is available at https://www.gencodegenes.org. Source data are provided with this paper.

### Code availability

The code generated in this study is available at https://figshare.com/s/d89f8f9f90d3a76eccb9 (ref. 38).

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

**Acknowledgements** We thank all of the volunteers who participated in this study; M. Devoto for a critical reading of the paper; M. Masala for help and advice; R. Kurita and Y. Nakamura for providing the HUDEP-2 cell line; S. Murino for help with collecting samples; V. Sogos for help with acquiring microscope images; and the CeSAR (Centro Servizi d'Ateneo per la Ricerca) of University of Cagliari and R. Pillai for FACS technical assistance. We acknowledge support by grants 75N95021C00012 (SardiNIA5) and HHSN271201600005C (SardiNIA4) to F.C. from the Intramural Research Program of National Institute of Aging, National Institutes of Health (NIH), for the set-up and maintenance of the Sardinian cohort study.

**Author contributions** F.C. conceived and supervised the overall study. M.G.M. and M. Mingoia designed the functional experiments to clarify the biological bases of the observed genetic associations and the pathways involved. A.L. and I.A. prepared luciferase plasmids and performed transfections, EMSA and western blot experiments. M. Mingoia and L.M. cultured erythroid progenitors. M. Mingoia designed the CRISPR–Cas9 genome editing strategy and M. Mingoia, M.G.M. and A.L. analysed the editing efficiency. C.A.C., F.V. and M.F.M. performed FACS analysis. A.P. designed and I.T., M.L.I., A. Manca and V.L. performed *P. falciparum* infection experiments. I.T. and C.D. performed ROS experiments and analysis. M.G.M. and A.P. supervised the malaria and ROS experiments and data analyses. A. Mulas, M. Pitzalis, V.O. and E.F. performed recruitment and genotyping of the samples. M.S., M.F., M. Pala, M. Marongiu and C.S. performed bioinformatics and statistical analysis. M.S., M.F., X.L. and M.C.S. performed population genetics analysis. M.G.M., M. Mingoia and F.C. drafted the manuscript. M.S., C.S., J.N. and A.P. contributed to the writing of specific sections. M.G.M., M. Mingoia, M.S., M.L.I., M.F., M.Z., P.M., F.M.T., C.S., D.S. and J.N. revised the manuscript. F.C. oversaw the various contributions made to writing the manuscript. F.C. provided funds and reagents. All authors read the paper and contributed to its final form.

**Funding** Open access funding provided by Consiglio Nazionale Delle Ricerche (CNR) within the CRUI-CARE Agreement.

**Competing interests** C.S. is currently an employee of Regeneron Pharmaceuticals and beneficiary of stock options and grants. The remaining authors declare no competing interests.

**Additional information**
**Correspondence and requests for materials** should be addressed to Francesco Cucca.

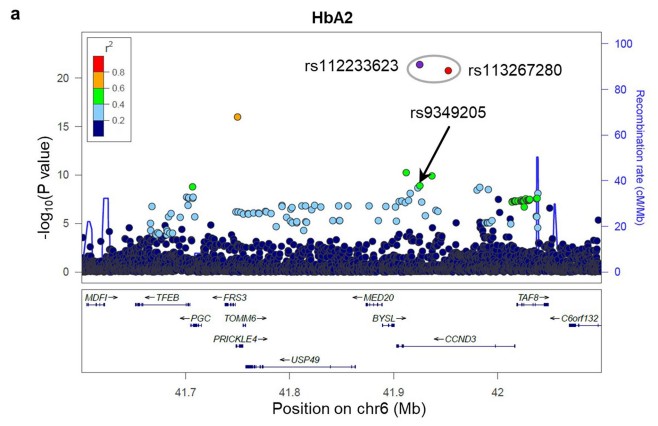

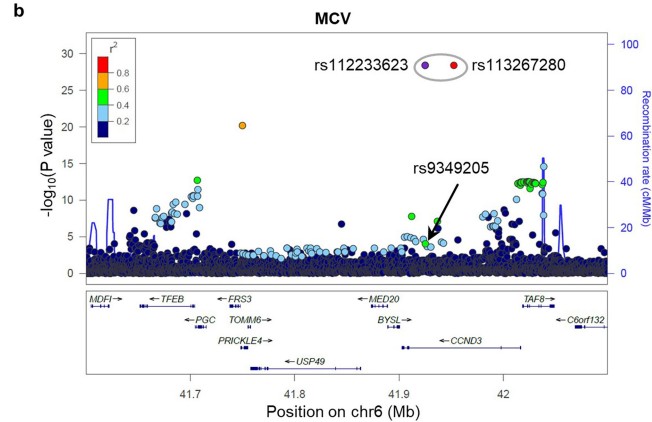

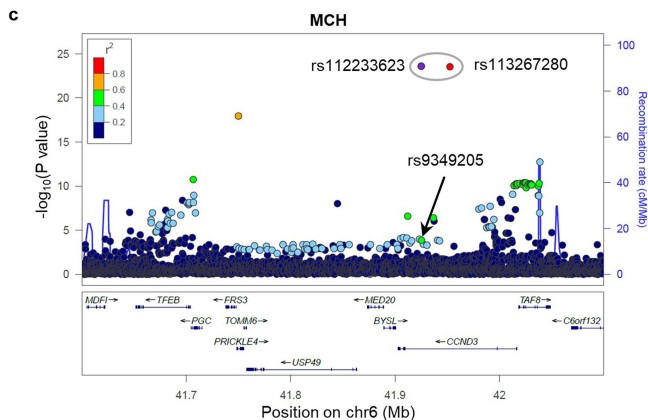

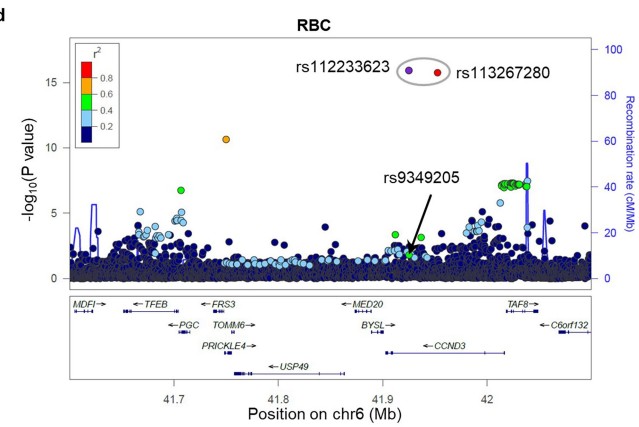

**Extended Data Fig. 1 | Regional association plots in the *CCND3* region.**
**a**–**d**, Association profiles for (**a**) HbA2 (g/dl); (**b**) MCV; (**c**) MCH; and (**d**) RBC, are shown. The significance of the association (-log10 (P-value), left y axis) for each trait is plotted relative to the genomic positions on the hg19/GRCh37 genomic build (x axis). Symbols reflect genomic functional annotations. SNPs are coloured to reflect their LD with rs112233623 in Sardinians (indicated with a purple dot). Two other variants, rs113267280 and rs9349205, are explicitly marked. Variants circled in grey indicate the 95% summed posterior probability credible set. Association P values were computed in n = 6,824 Sardinian samples using the EPACTS q.emmax() test, a linear mixed model that accounts for sample structure and relatedness; all P values are two-sided. The genome-wide significance threshold is $P < 6.9 \times 10^{-9}$. The plots were drawn using the standalone version of LocusZoom (http://locuszoom.org/).

**a**

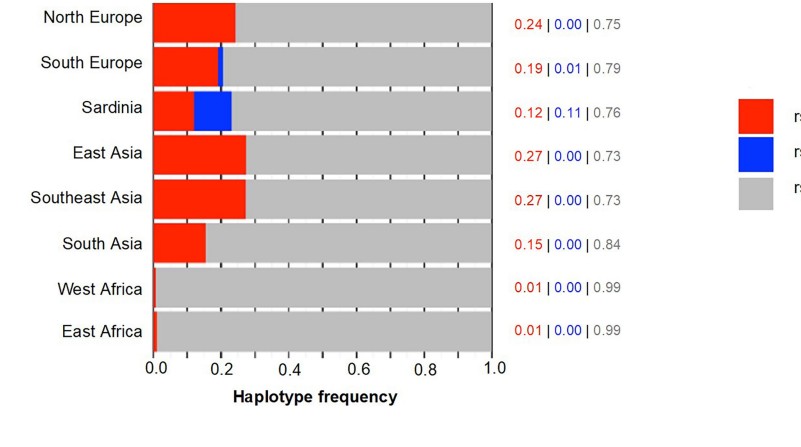

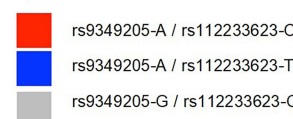

rs9349205-A / rs112233623-C

rs9349205-A / rs112233623-T

rs9349205-G / rs112233623-C

**b**

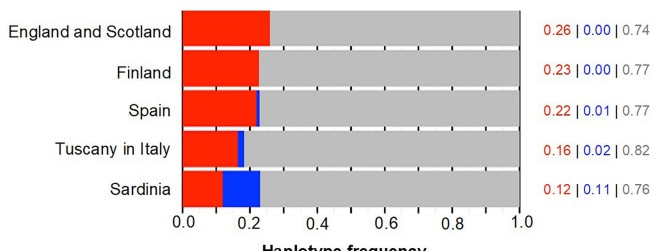

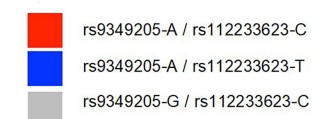

rs9349205-A / rs112233623-C

rs9349205-A / rs112233623-T

rs9349205-G / rs112233623-C

**Extended Data Fig. 2 | Population-specific haplotype frequencies of rs9349205–rs112233623 haplotypes. a,b,** The haplotype frequencies of rs9349205–rs112233623 haplotypes in 1KG superpopulations (**a**) and in European populations (**b**) were calculated with LDLink interface (https://ldlink.nih.gov/?tab=ldhap) in 1KG[18] populations. Legend: **a,** Northern Europe: British in England and Scotland, UK + Finns, Finland; Southern Europe: Iberian population, Spain + Tuscans, Italy; South Asia: Bengalis, Bangladesh + Punjabis of Lahore, Pakistan; East Asia: Dai Chinese in Xishuangbanna, China + Southern Han Chinese in Beijing, China; South East Asia: Japanese in Tokyo, Japan + Kinh in Ho Chi Minh City, Vietnam; West Africa: Esan, Nigeria + Gambians in West Divisions, Gambia + Mende, Sierra Leone + Yoruba in Ibadan, Nigeria; East Africa: Luhya in Webuye, Kenya. **b,** Finns, Finland; British in England and Scotland, UK; Iberian population, Spain; Tuscans, Italy and Sardinia.

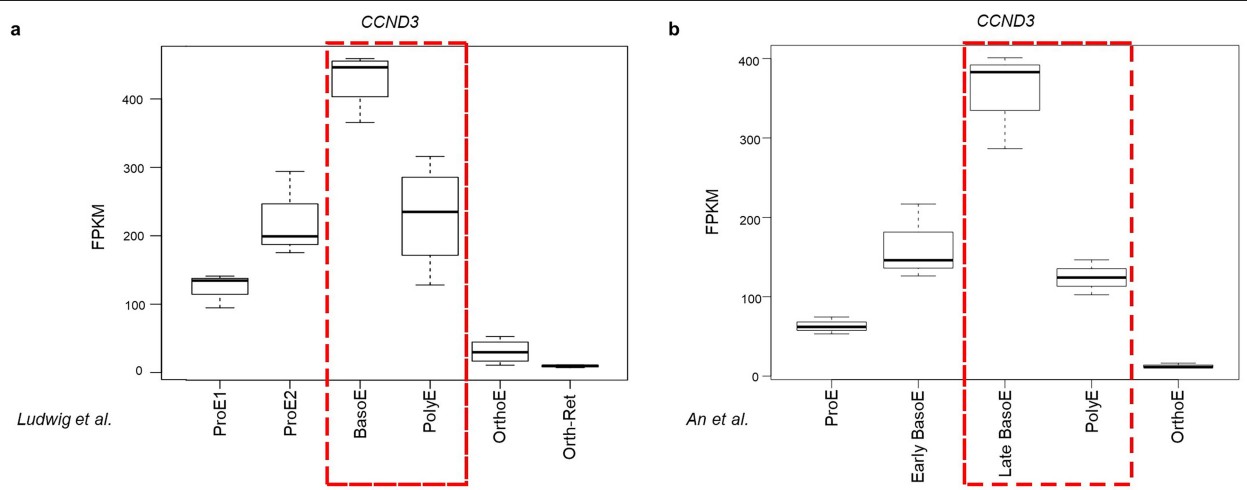

**a** *CCND3*

FPKM

*Ludwig et al.*

ProE1 ProE2 BasoE PolyE OrthoE Orth-Ret

**b** *CCND3*

FPKM

*An et al.*

ProE Early BasoE Late BasoE PolyE OrthoE

**c**

day -6    day 0    day 7    day 9    day18

Expansion    Erythroid differentiation

CD34⁺ isolated
from peripheral blood

α4-integrin-APC

Band3-PE

**d**

day 7    day 9

**e**

*CCND3* expression
(relative to *β2M*)

Day 7    Day 9

**Extended Data Fig. 3** | See next page for caption.

**Extended Data Fig. 3 | Expression of *CCND3* in erythroid cultures.**
**a**,**b**, Normalized expression levels (fragments per kilobase of transcript per million mapped reads; FPKM) of *CCND3* in proerythroblast (ProE1, n = 3; ProE2, n = 3), basophilic erythroblast (BasoE, n = 4), polychromatophilic erythroblast (PolyE, n = 4), orthochromatophilic erythroblast (OrthoE, n = 4), and orthochromatophilic / reticulocyte (OrthoE-Ret n = 4) RNA-seq data from Ludwig et al.[20] (**a**); and in proerythroblast (ProE, n = 3), early basophilic erythroblast (Early BasoE, n = 3), late basophilic erythroblast (Late BasoE, n = 3), polychromatophilic erythroblast (PolyE, n = 3), and orthochromatophilic erythroblast (OrthoE, n = 3) RNA-seq data from An et al.[19] (**b**), reprocessed according to the Methods. Box-and-whiskers plots indicate the median (by a line across the box); the box ranges from the first quartile to the third quartile of the distribution; the "whiskers" on box plots extend from the box to the most extreme data points. Red dotted boxes indicate basophilic erythroblasts, characterized by the peak of *CCND3* expression, and polychromatophilic erythroblasts, marked by the decline in *CCND3* expression. **c**, Representative flow-cytometry dot plots of GPA⁺ gated erythroblasts showing expression of Band3 and integrin α4 at two successive points day 7 (left) and day 9 (right) of differentiation. **d**, Morphological appearance of cells collected at the same time points as above (day 7 and day 9) showing the prevalence of BasoE and PolyE stages. Representative cytospin slides were stained with May–Grünwald–Giemsa (n = 3 biologically independent experiments). Bars represent 10 µm. **e**, Relative expression of *CCND3* mRNA quantified by RT–qPCR in different samples (n = 3 biologically independent experiments; 3 donors) for two time points, day 7 and day 9, of erythroid differentiation relative to a β2-microglobulin (β2M) endogenous control. Data are shown as box-and-whisker plots, with boxes indicating the interquartile range (IQR; 25th–75th percentile), the horizontal line the median, and whiskers extending to 1.5× IQR, and individual points beyond the whiskers shown as outliers. The two-sided P-value (P = 2.88⁻⁰⁴) assessing the differential expression at the time points considered was obtained using the Wilcoxon–Mann–Whitney non-parametric test. The statistical results for all comparisons are provided in Supplementary Table 3.

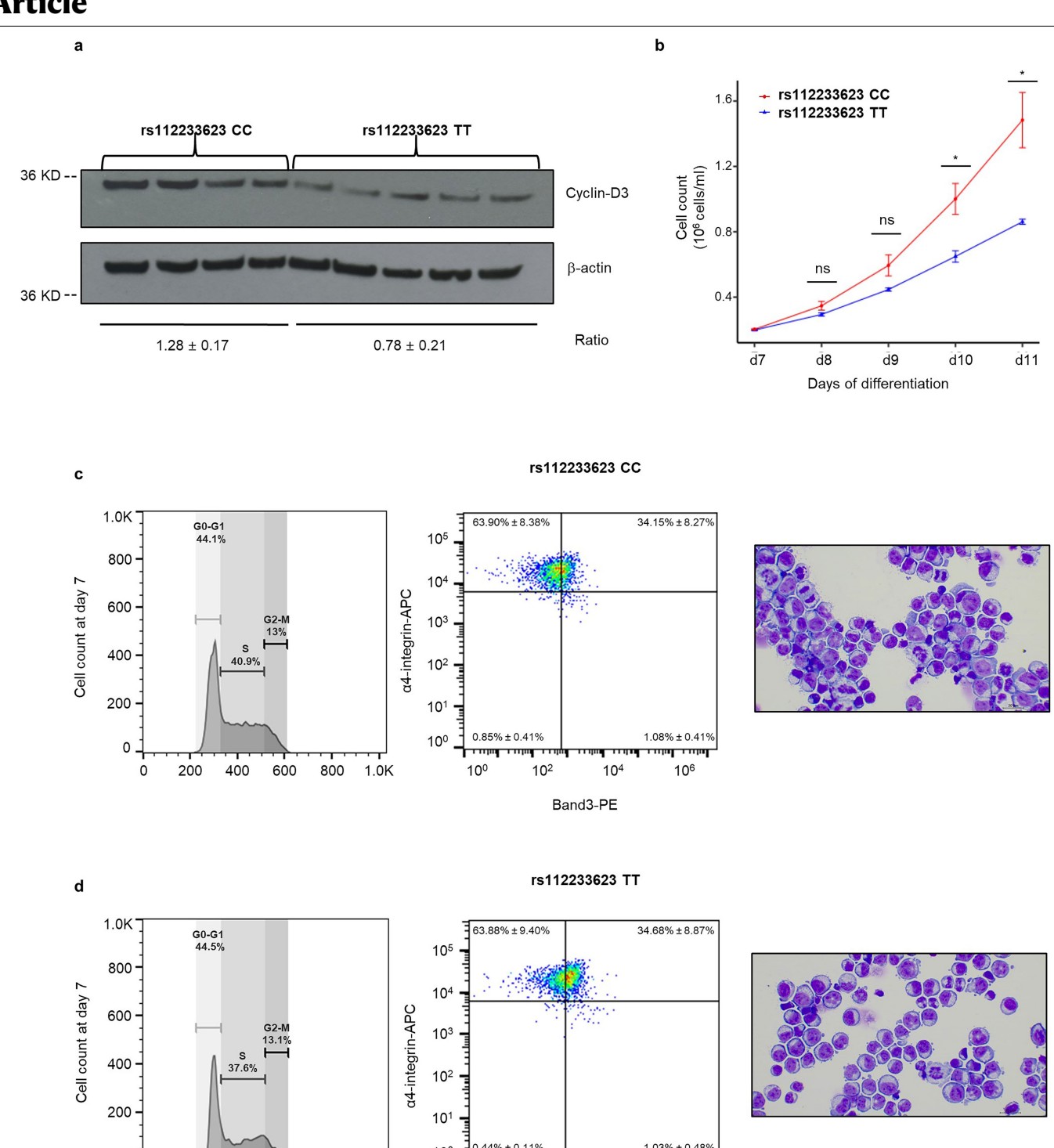

**Extended Data Fig. 4 | Influence of the rs112233623-T allele on cyclin D3 expression, growth and cell cycle in erythroid cultures. a**, Western blot assay showing cyclin D3 expression in erythroblasts derived from individuals homozygous for the WT rs112233623-C allele (CC, n = 4) versus the derived rs112233623-T allele (TT, n = 5); human β-actin was run on the same gel as loading control. Band intensities in the immunoblots were quantified using ImageJ software using the same detection time point for both signals (1 min) and the differences were expressed as ratio of Cyclin D3 to β-actin means for each group. For gel source data, see Supplementary Fig. 1. **b**, Growth curves of cultured erythroid progenitor cells derived from individuals homozygous for the WT rs112233623-C allele (CC, red line) versus derived rs112233623-T allele (TT, blue line). The mean ± s.e.m is shown, (for each genotype, n = 3 biologically independent experiments). A two-sided Student's *t*-test was used (*P < 0.05; ns, not significant). The statistical results for all comparisons are provided in Supplementary Table 3. **c**,**d**, Analysis of erythroblasts derived from individuals homozygous for the WT rs112233623-C allele (**c**) or the derived rs112233623-T allele (**d**) at day 7 of erythroid differentiation. In each panel, from left to right: representative flow-cytometry plots with propidium iodide staining, showing the distribution of cell-cycle phases (G0–G1, S, G2–M); dot plots of GPA+ gated erythroblasts showing expression of Band3 and integrin α4 (data are mean ± s.e.m., n = 4 per genotype); and cytospin preparations stained with May–Grünwald–Giemsa (scale bars, 20 μm; 50× magnification).

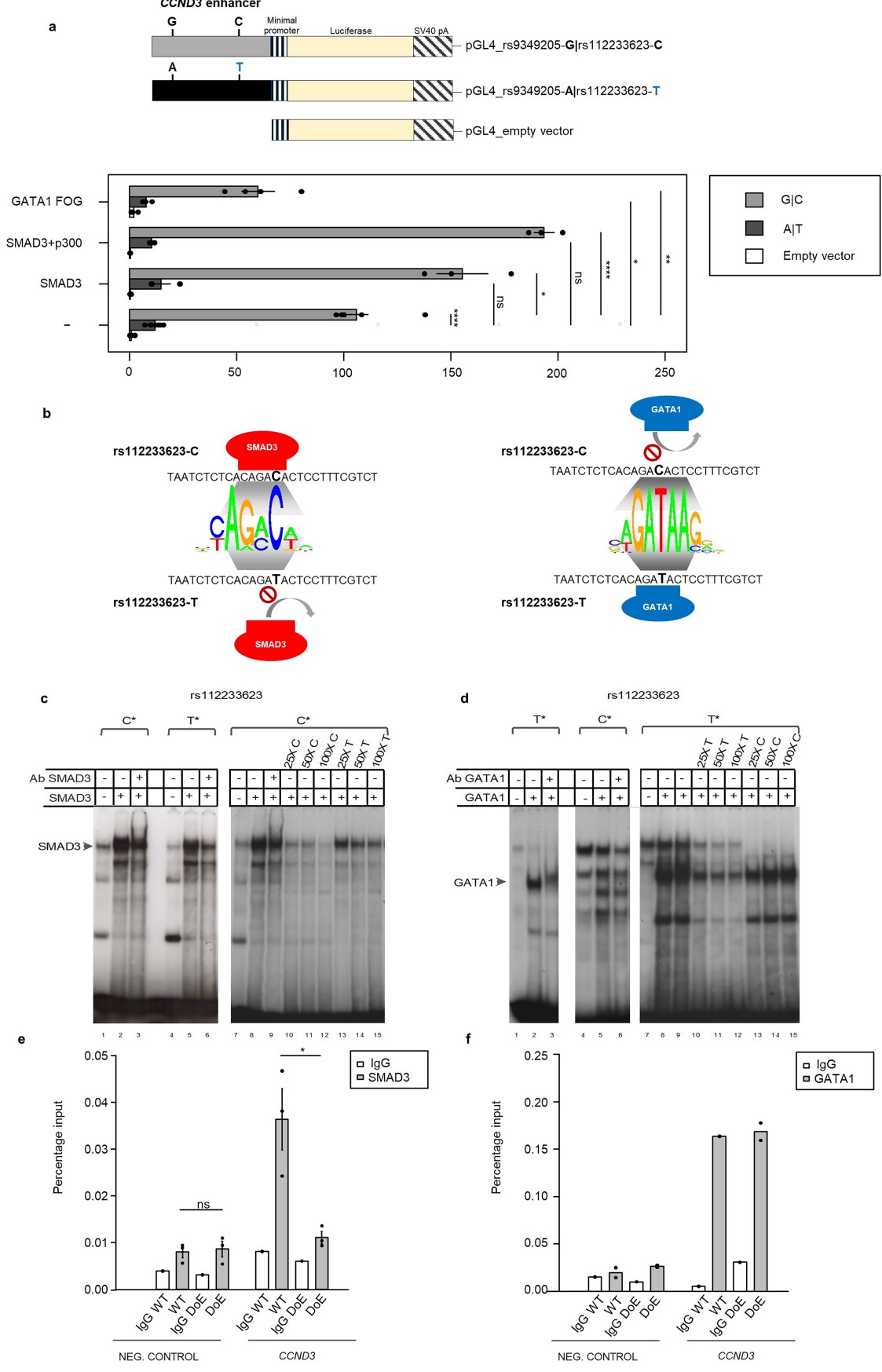

**Extended Data Fig. 5** | See next page for caption.

**Extended Data Fig. 5 | Influence of the rs112233623-T allele on the binding of transcription factors SMAD3 and GATA1. a**, Top, vector content and expression of luciferase reporter genes driven by the extreme allele combinations: on the one hand the WT rs9349205-G|rs112233623-C (G|C) alleles on the other hand the derived rs9349205-A| rs112233623-T (A|T) alleles of the *CCND3* enhancer in the erythroid cell line HUDEP-2, with empty vector as a control construct. Below, histograms show averages of the relative luminescence activity of the combinations of the two extreme alleles at rs9349205 and rs112233623 described above in absence (-) or presence of GATA1 and FOG-expressing plasmids; SMAD3 and p300 expressing plasmids; SMAD3 expressing plasmid alone; or empty expression vector. All values are plotted relative to the WT (G|C) construct. Note that the activity of the empty pGL4 vector is barely detectable. The mean ± s.e.m is shown (n = 7 (vectors); n = 3 (vectors + SMAD3), n = 3 (vectors+SMAD3 + p300), n = 4 (vectors + GATA1 + FOG) biologically independent experiments). A two-sided Student's *t*-test was used, with level of significance indicated by asterisks (*P < .05;**P < .01; ****P < .0001, ns, not significant). The statistical results for all comparisons are provided in Supplementary Table 3. **b**, In silico prediction of binding to rs112233623 allele variants, showing derived allele T hindering binding of SMAD3 while favouring GATA1 binding. **c,d**, Representative electrophoretic mobility shift assays (EMSA) showing binding of SMAD3 and GATA1 proteins with labelled oligonucleotide probes (*) containing the WT (C) or derived (T) allele of rs112233623 (n = 3 biologically independent experiments). Competitor unlabelled oligonucleotides were used at the indicated fold excess to demonstrate specificity of binding. **c**, SMAD3 binds to the WT rs112233623-C allele (lanes 2 and 8); is weakly supershifted by anti-SMAD3 antibody (lanes 3 and 9); and is competed away by an excess of unlabelled DNA oligonucleotides containing the WT rs112233623-C allele (lanes 10-12, at 25X, 50X and 100X respectively), but not by an excess of oligonucleotide containing the derived rs112233623-T allele (lanes 13-15, at 25X, 50X and 100X respectively). **d**, GATA1 binds to the rs112233623-T derived allele (lanes 2, 8 and 9), is supershifted by anti-GATA1 antibody (lane 3); and is competed away by an excess of unlabelled DNA oligonucleotides containing the derived rs112233623-T allele (lanes 10-12, at 25X, 50X and 100X respectively) compared to unlabelled DNA oligonucleotides containing the WT rs112233623-C allele (lanes 13-15, at 25X, 50X and 100X respectively). GATA1 does not appear to bind to the WT rs112233623-C allele (lane 5). The samples derive from the same experiment and, the gels were processed in parallel. For gel source data, see Supplementary Fig. 1. **e,f**, ChIP–qPCR for SMAD3 or GATA1 binding to the *CCND3* enhancer region surrounding rs112233623 (*CCND3*) and to a SimpleChIP Human α Satellite as negative control in erythroblasts derived from individuals homozygous for the rs112233623-T decrease of expression (DoE) allele versus the WT rs112233623-C allele (WT). **e**, ChIP was performed using an antibody against SMAD3, results are represented as percentage of input and nonspecific IgG used as negative control. The mean ± s.e.m is shown (n = 3 biologically independent experiments). A two-sided two-Sample t-test was used; significant differences are indicated (*P < .05; ns, not significant). The statistical results for the comparisons are provided in Supplementary Table 3. **f**, ChIP assays were conducted with an antibody against GATA1, results are represented as percentage of input and nonspecific IgG used as negative control (n = 2 biologically independent experiments).

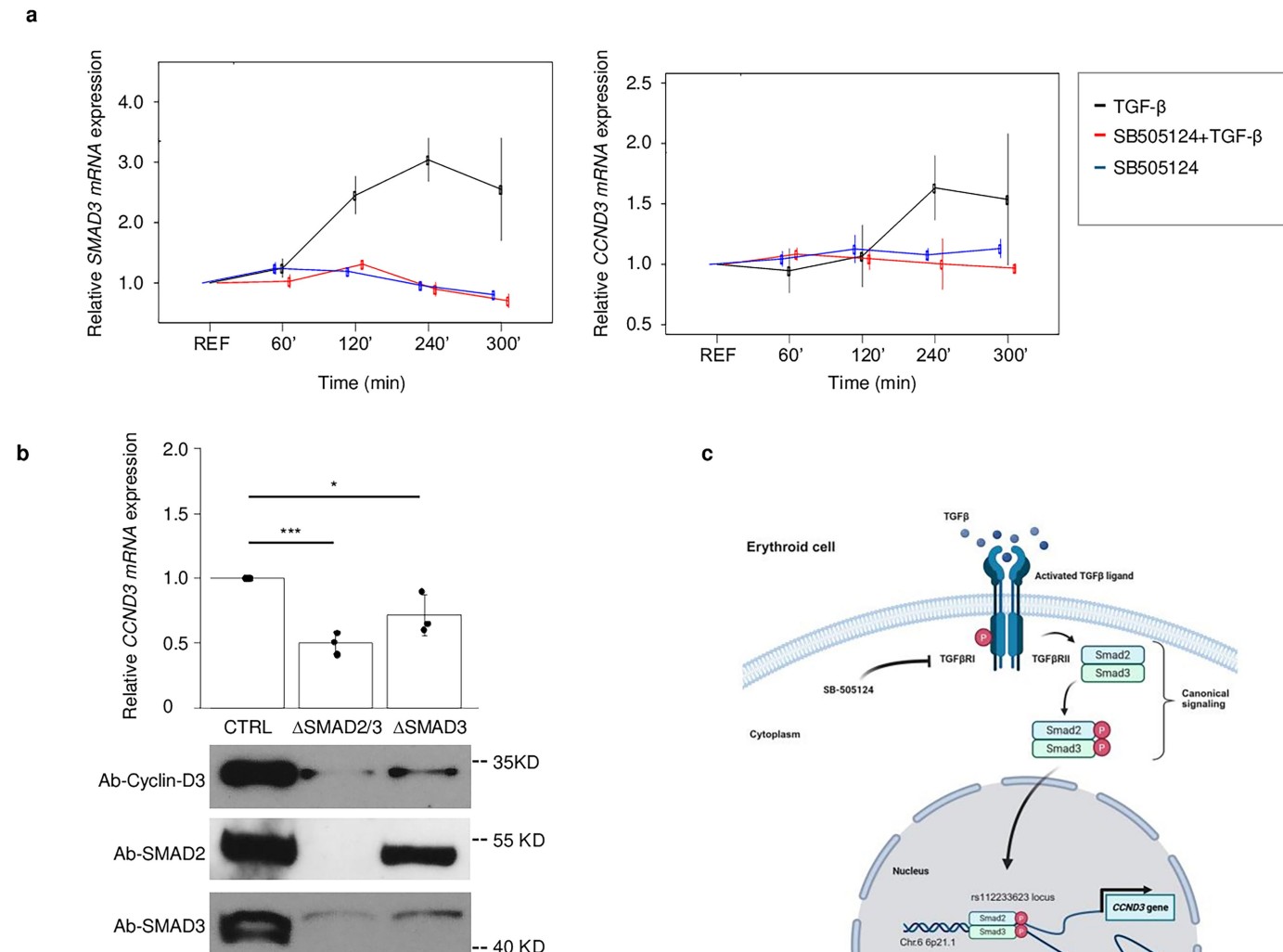

**Extended Data Fig. 6 | The TGFβ–SMAD pathway in the regulation of *CCND3*.**
**a**, Stimulation of *SMAD3* (left panel) and *CCND3* mRNA expression (right panel) by TGFβ in HUDEP-2 cells (black line) at 60', 120', 240', and 300' time points, quantified by RT–qPCR analysis using Glyceraldehyde 3-phosphate dehydrogenase (*GAPDH*) as an endogenous control. *SMAD3* and *CCND3* stimulation is abrogated by addition of the TGFβ specific inhibitor, SB-505124 (red line SB-505124 + TGFβ, blue line SB-505124 alone). The the mean ± s.e.m is shown (n = 3 biologically independent experiments at 60' and 300'; n = 4 biologically independent experiments at 120' and 240' time points). **b**, Top, RT–qPCR analysis of *CCND3* mRNA level in HUDEP-2 cells transduced with single guide gRNAs (sgRNAs) targeting the coding region of *SMAD3* and *SMAD2* (Δ*SMAD2/3*) or *SMAD3* alone (Δ*SMAD3*) compared to cells transduced with Cas9 only, without sgRNA (CTRL). The *CCND3* mRNA expression was normalized to *GAPDH* mRNA. The mean ± s.e.m is shown (n = 3; 3 biologically independent replicates). Statistical significance, calculated by two-sided t-test, is indicated by asterisks (*P < .05, ***P < .001). The statistical results for all comparisons are

provided in Supplementary Table 3. Bottom, western blot assay showing Cyclin D3 downregulation in HUDEP-2 cells after gene-editing of the *SMAD3/SMAD2* (Δ*SMAD2/3*) or *SMAD3* (Δ*SMAD3*) coding region; human β-actin was run on the same gel as loading control. Bands in immunoblots were quantified using ImageJ software (https://imagej.nih.gov/ij/download.html) and the ratio between Cyclin D3 and β-actin in each sample was normalized relative to control sample of lane 1 (CTRL). For gel source data, see Supplementary Fig. 1. **c**, Schematic diagram of the TGFβ signalling pathway involved in the regulation of *CCND3* expression. TGFβ ligand binds to its receptor TGFβRII, leading to phosphorylation of TGFβRI, which in turn phosphorylates SMAD2/3, enabling their translocation to the nucleus. Phosphorylated SMAD2/3 complexes (pSMAD2/3) bind to the *CCND3* erythroid-specific enhancer around rs112233623, thereby activating *CCND3* gene expression (considering NM_001760.5). Red star in the locus 6p21.1 indicates rs112233623. The statistical results for all comparisons are provided in Supplementary Table 3.

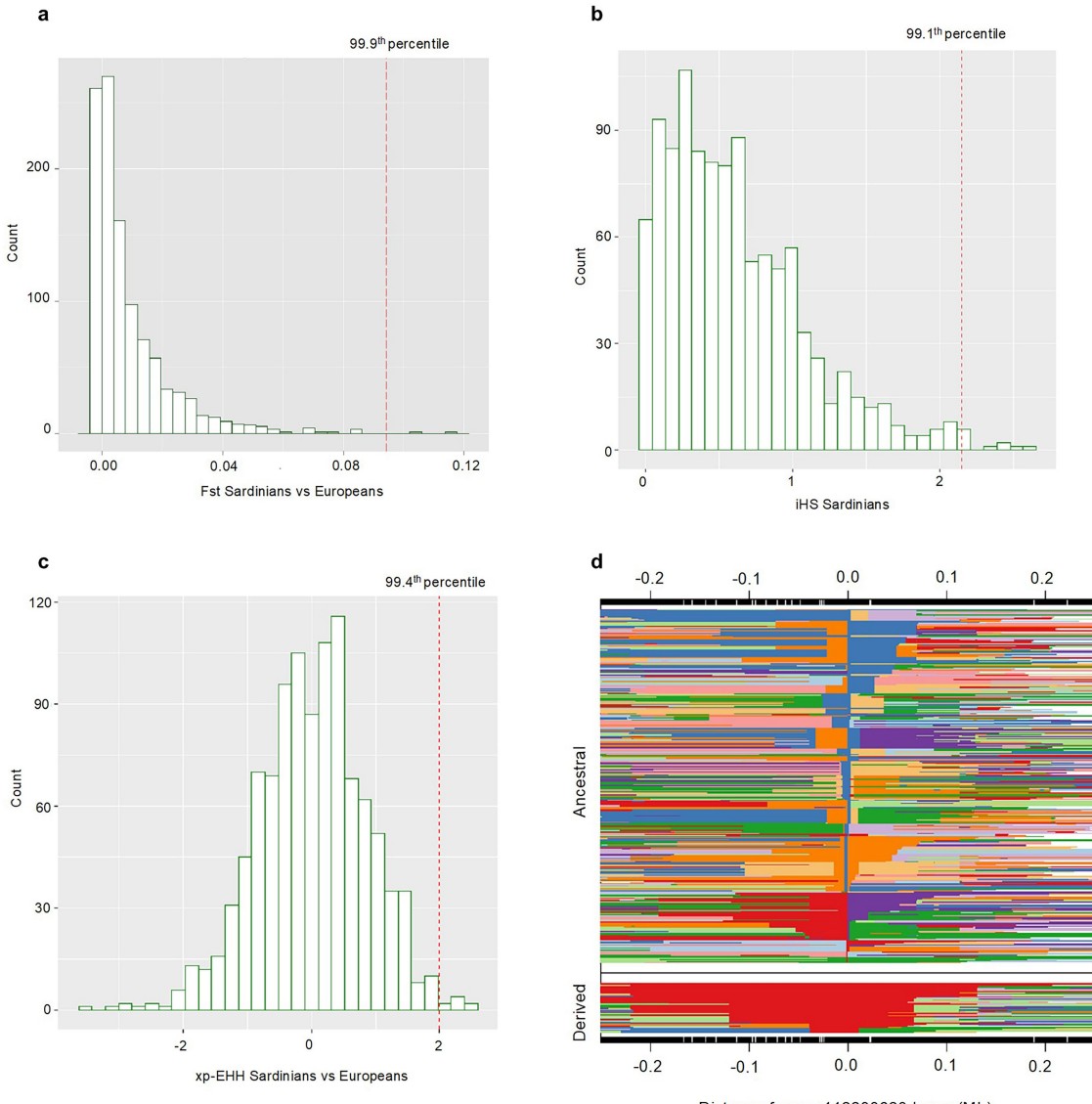

**Extended Data Fig. 7 | Allele-frequency differentiation and haplotype-based tests for positive selection on rs112233623. a–c**, Genomic distribution of $F_{ST}$ values of Sardinians compared to the EUR supergroup from 1KG (**a**), absolute standardized iHS values in SardiNIA (**b**) and xp-EHH standardized values in Sardinians compared to the EUR from the 1KG (**c**). In **a**–**c**, the distribution for 1,075 variants matched with rs112233623 by allele frequency in SardiNIA, local recombination rate, and B score is shown. The vertical red lines represent rs112233623 $F_{ST}$, iHS and xp-EHH values. **d**, Extended haplotype homozygosity for rs112233623-derived (T) and -ancestral (C) alleles.

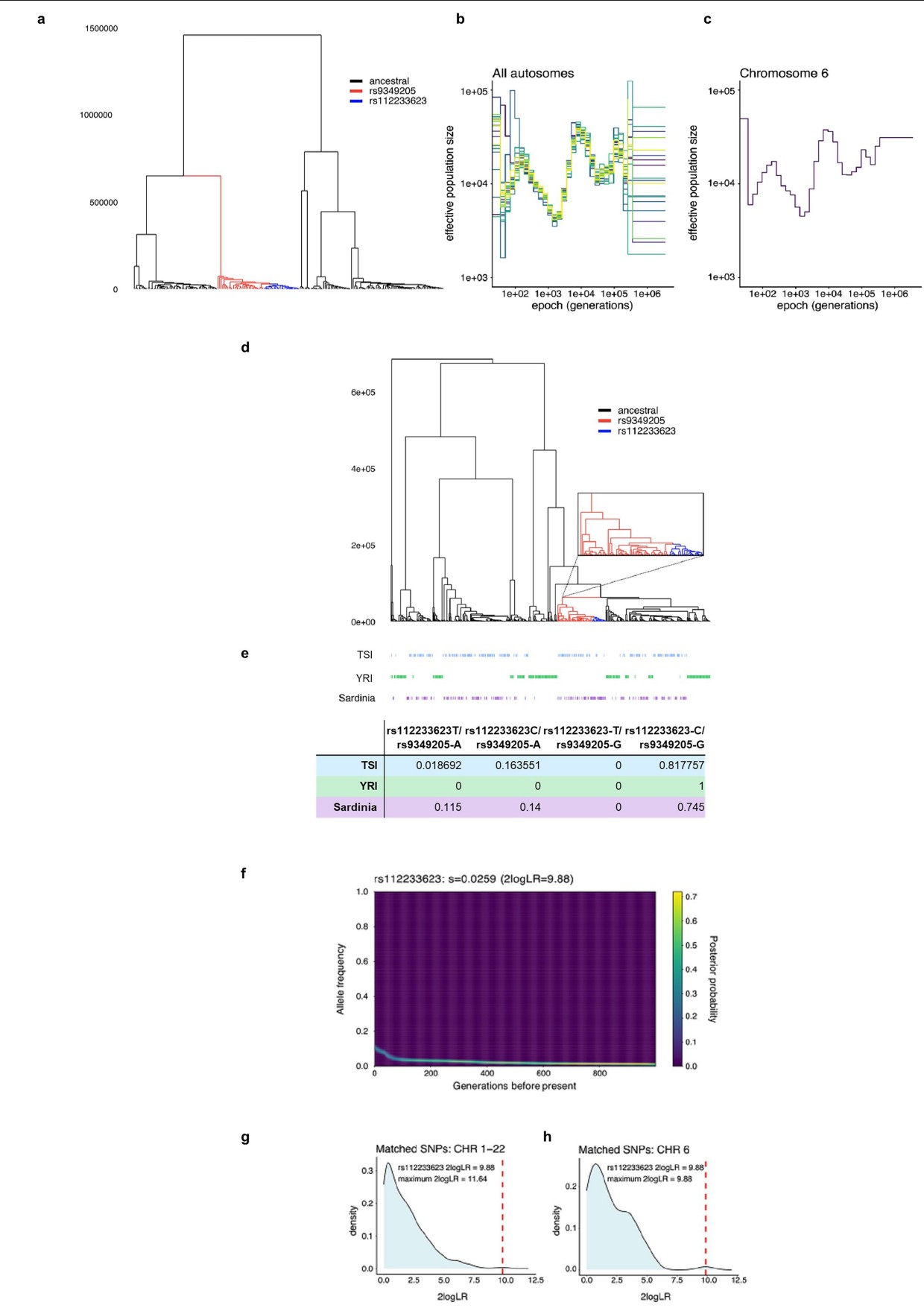

**Extended Data Fig. 8** | See next page for caption.

**Extended Data Fig. 8 | Estimation of genealogies, allele frequency trajectory and selection coefficient for rs112233623. a**, Plot of the single marginal tree that contains the variants of interest (rs112233623 and rs9349205) in the ancestral recombination graph inferred from the described subset of 100 Sardinian individuals. The lineages carrying the derived allele at the increase of expression (IoE; rs9349205) and decrease of expression (DoE; rs112233623) sites are highlighted in red and blue, respectively. **b,c**, Effective population size estimates inferred from the described subset of Sardinian individuals using the Relate program (Methods): inferred across each autosome (**b**) and inferred from chromosome 6 (**c**). **d**, Plot of the single marginal tree that contains the variants of interest (rs112233623 and rs9349205) in the ancestral recombination graph inferred from a merged dataset including 100 Sardinian individuals[73],

107 Tuscan individuals from Italy and 108 Yoruba individuals from Ibadan, Nigeria[18]. Population labels corresponding to each haplotype are depicted by the coloured lines below the tree. **e**, Haplotype frequencies for rs112233623–rs9349205 across each population are reported using the same label colours as in **a**. **f**, Allele frequency trajectory for rs112233623 in Sardinian individuals, as inferred by CLUES. **g**, Distribution of logLR outputted by CLUES for 1,035 SNPs matched by minor allele frequency, B score (background selection), and recombination rate with that of rs112233623, excluding the HLA region. The value of 2logLR (in the 99.5th percentile of logLR values for matched variants, excluding the HLA region) is marked by the red dashed line. **h**, Similar to **g**, but including only matched variants from chromosome 6 (74 total SNPs, excluding the HLA region).

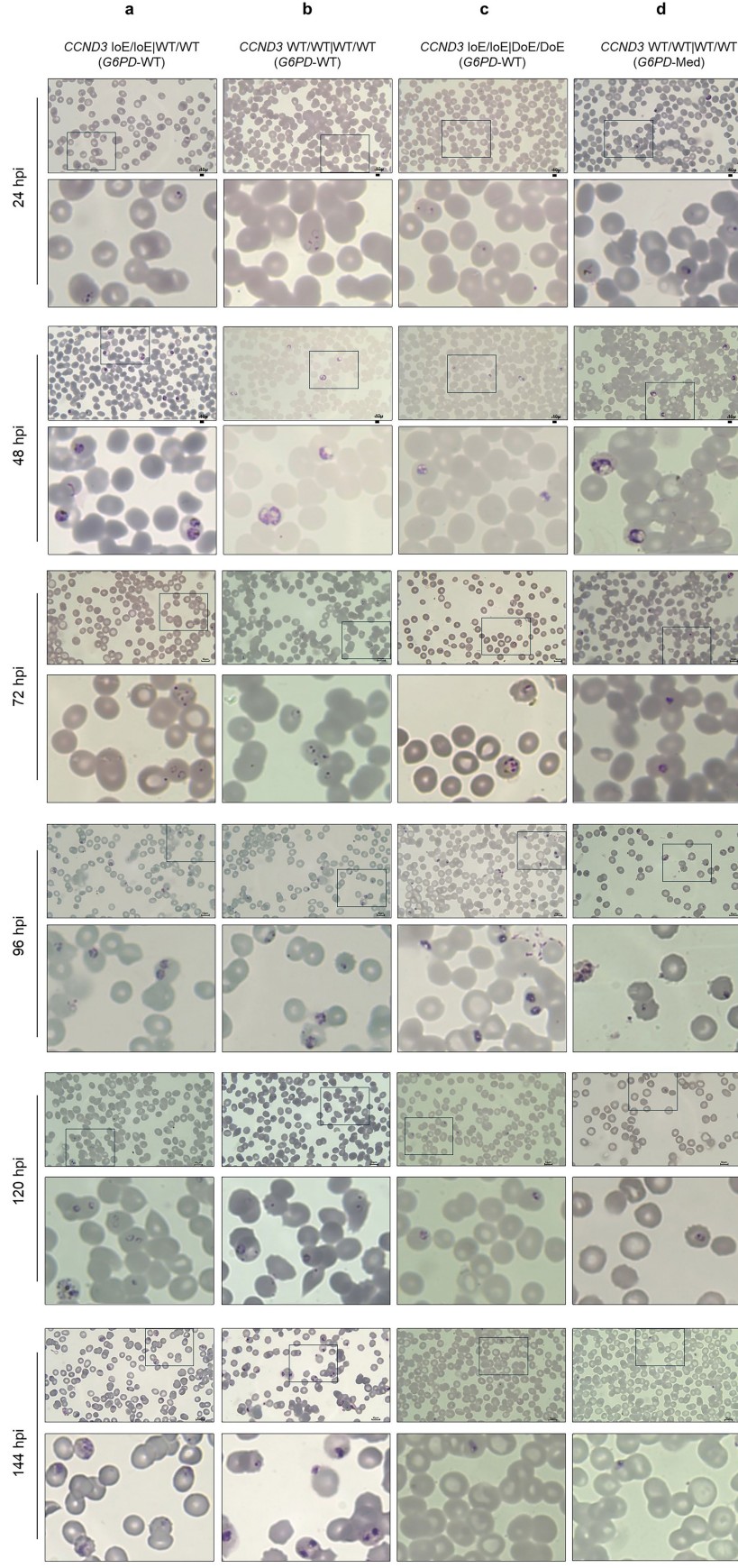

**Extended Data Fig. 9** | See next page for caption.

**Extended Data Fig. 9 | Differences in parasite development between genotypes. a–d**, Representative photomicrographs of blood smears of parasite cultures measured in RBCs from individuals bearing the alleles indicated at the top of the corresponding panels (top: large field, bottom: zoom) at different time points throughout the assay: 24, 48, 72, 96, 120 and 144 hpi homozygotes for the IoE allele rs9349205-A, for the WT allele rs112233623-C, and WT for *G6PD*-Med (IoE/IoE|WT/WT;*G6PD*-WT) (**a**); homozygotes for the WT alleles rs9349205-G, rs112233623-C and WT for *G6PD* (WT/WT|WT/WT;*G6PD*-WT) (**b**); homozygotes for the derived IoE allele rs9349205-A, for the derived DoE allele rs112233623-T and WT for *G6PD* (IoE/IoE|DoE/DoE;*G6PD*-WT) (**c**); and homozygotes for the WT alleles rs9349205-G, rs112233623-C, and hemizygous for the *G6PD* Mediterranean deficiency allele (WT/WT|WT/WT; *G6PD*-Med) (**d**) (n = 14, 7, 17 and 5 biological independent samples derived from selected donors, respectively). The scale bars are 10 μm. In **a**,**b**, the photomicrographs show a similar pattern, with *Plasmodium* cultures remaining synchronous up to 144 h post-invasion. By contrast, in **c**, the photomicrographs show that parasite growth is significantly slowed in the *CCND3* IoE/IoE|DoE/DoE genotype, similar to the impaired development observed in *G6PD*-Med deficient individuals (**d**). This phenomenon is evident by a decrease in parasitaemia, becoming more pronounced from 72 h onwards, by an increase in the number of dead parasites, and by an abundance of pycnotic nuclei of *P. falciparum* (**c**,**d**), compared to IoE/IoE |WT/WT;*G6PD*-WT (**a**) and WT/WT|WT/WT *G6PD*-WT control group (**b**).

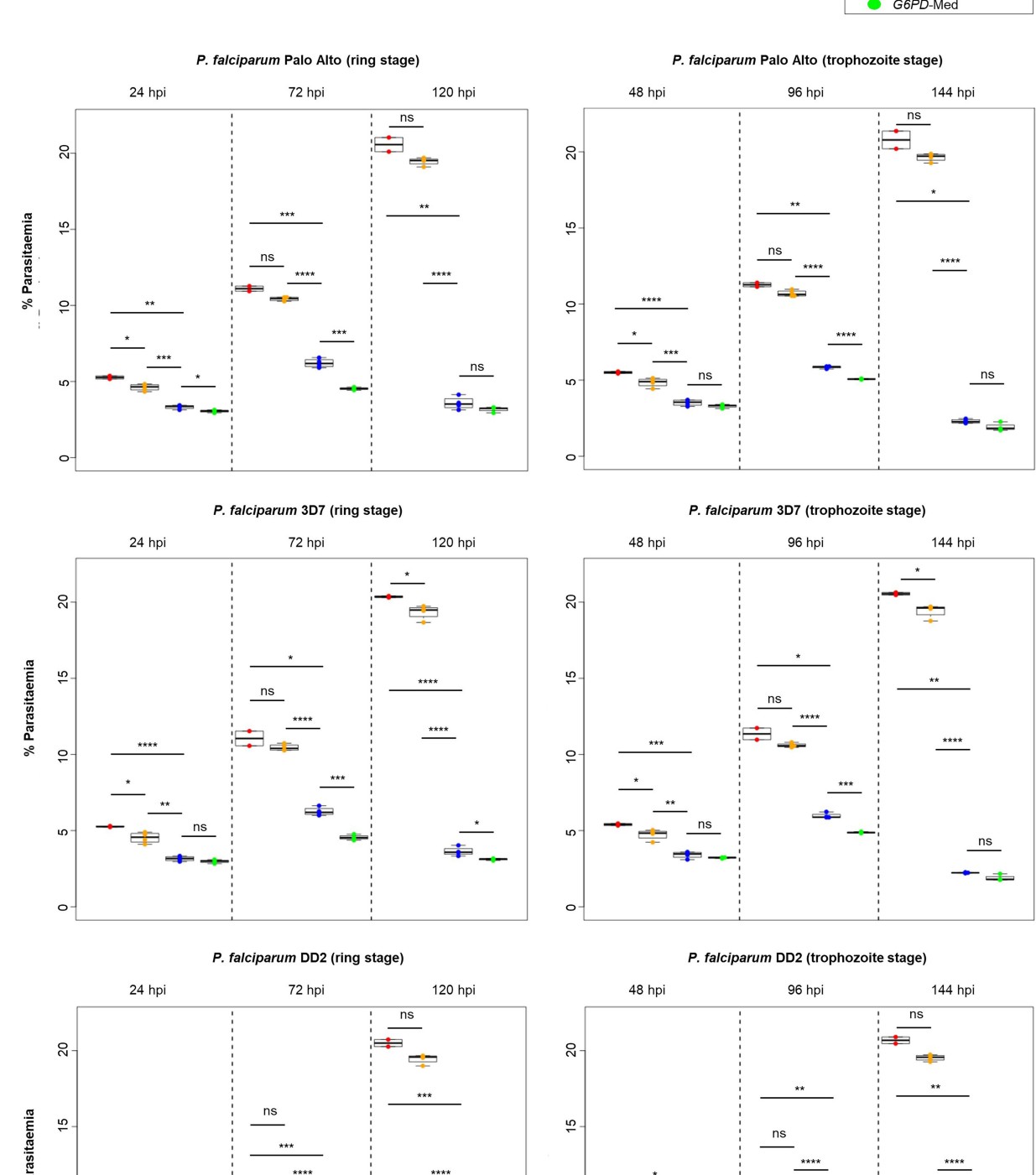

**Extended Data Fig. 10** | See next page for caption.

**Extended Data Fig. 10 | Differences in parasite development across genetically diverse strains of *P. falciparum*.** Level of *P. falciparum* parasitaemia across different isolates (Palo Alto; above, 3D7; middle, DD2; below) in RBCs derived from donors homozygotes for the IoE allele rs9349205-A, for the WT allele rs112233623-C, and for *G6PD*-WT (IoE/IoE|WT/WT;*G6PD*-WT) in red; homozygotes for the WT alleles rs9349205-G, rs112233623-C and *G6PD*-WT (WT/WT|WT/WT;*G6PD*-WT) in orange; homozygotes for the derived IoE allele rs9349205-A, the derived DoE allele rs112233623-T and for *G6PD*-WT (IoE/IoE|DoE/DoE;*G6PD*-WT) in blue; and homozygotes for the WT alleles rs9349205-G, rs112233623-C, and hemizygous for *G6PD* Mediterranean allele (WT/WT|WT/WT; *G6PD*-Med) in green (n = 2, 4, 4 and 3 biological independent samples derived from selected donors, respectively). Measurements were taken at 24, 72, and 120 hpi (left side) and at 48, 96, and 144 hpi (right side). Data are shown as box-and-whiskers plots, where the box denotes the interquartile range (IQR; 25th–75th percentile) and the horizontal line in each box indicating the median level of parasitaemia. The whiskers extend to the most extreme data points within 1.5× IQR of the box. Two-sided P values are based on the non-parametric Wilcoxon–Mann–Whitney test (*P < .05; **P < .01; ***P < .001; ****P < .0001; ns, not significant). The statistical results for all comparisons are provided in Supplementary Table 3.

# Reporting Summary

## Statistics

For all statistical analyses, confirm that the following items are present in the figure legend, table legend, main text, or Methods section.

| n/a | Confirmed | |
|---|---|---|
| ☐ | ☒ | The exact sample size (*n*) for each experimental group/condition, given as a discrete number and unit of measurement |
| ☐ | ☒ | A statement on whether measurements were taken from distinct samples or whether the same sample was measured repeatedly |
| ☐ | ☒ | The statistical test(s) used AND whether they are one- or two-sided<br>*Only common tests should be described solely by name; describe more complex techniques in the Methods section.* |
| ☐ | ☒ | A description of all covariates tested |
| ☐ | ☒ | A description of any assumptions or corrections, such as tests of normality and adjustment for multiple comparisons |
| ☐ | ☒ | A full description of the statistical parameters including central tendency (e.g. means) or other basic estimates (e.g. regression coefficient) AND variation (e.g. standard deviation) or associated estimates of uncertainty (e.g. confidence intervals) |
| ☐ | ☒ | For null hypothesis testing, the test statistic (e.g. *F*, *t*, *r*) with confidence intervals, effect sizes, degrees of freedom and *P* value noted<br>*Give P values as exact values whenever suitable.* |
| ☐ | ☒ | For Bayesian analysis, information on the choice of priors and Markov chain Monte Carlo settings |
| ☒ | ☐ | For hierarchical and complex designs, identification of the appropriate level for tests and full reporting of outcomes |
| ☐ | ☒ | Estimates of effect sizes (e.g. Cohen's *d*, Pearson's *r*), indicating how they were calculated |

*Our web collection on statistics for biologists contains articles on many of the points above.*

## Software and code

Policy information about availability of computer code

| | |
|---|---|
| Data collection | FACSCANTO (BD Bioscience); BD Accuri™ C6 Plus (BD Biosciences); Sinergy2 (Biotek); ABI 7700 machine (Applied biosystems); 3130xl Genetic Analyzer (Applied biosystems); Microscope Primo Star (Carl Zeiss) Objective lenses: 20x/0.50 PlanF and 100x/1.25 Oil Plan-ACHROMAT; Tali® Image-Based Cytometer (Invitrogen); Ultimate 3000 UPLC system (Thermo Scientific); iSCAN system (Illumina); PrimoStar 3 microscope and Axiocam 208 color camera (Carl Zeiss) Objective lense: 50X/1.0 Oil iPlan-ACHROMAT. |
| Data analysis | The software used in data analysis are described in the Methods section and are the following:<br>association analyses with EPACTS-3.2.6 (http://genome.sph.umich.edu/wiki/EPACTS), linkage disequilibrium-based statistics calculated using PLINK v1.9 (https://www.cog-genomics.org/plink/1.9/), fine-mapping with FINEMAP v1.4.2 (http://www.christianbenner.com), tests used in the biological experiments, descriptive statistics and plots performed using R version 3.5.3 (https://cran.r-project.org/). The population genetics analyses included the estimation of: Fst statistic using vcftools v.0.1.12b (http://vcftools.sourceforge.net/); EHH, iHS and xp-EHH statistics using selscan v2.0.0 (https://github.com/szpiech/selscan); genealogies using Relate v1.1.9 (Speidel et al, Nat Genet 2019); allele frequency trajectories and selection coefficients using CLUES (Stern, A. et al, PLOS Genet 2019). CADD scores are available at https://cadd.gs.washington.edu/score.<br>F RNA seq reads were aligned with STAR software (version 2.5.3a) to the human genome (hs37d5). rsem-calculate-expression function of RSEM software (version v1.3.1) and GENCODE gene annotation (version 14) (downloaded from https://www.gencodegenes.org) were used to generate (Fragment Per Kilobase of transcript per Million mapped reads)FPKM.<br>Flow cytometry data recording through FACSDiva software (version 6.1.3); Accuri C6 Plus and BD Csampler Plus software (version 1.0.34) and data plots by using FlowJo software (V10.10.0).<br>SgRNA oligonucleotides designed with CRISPR design tool (crispr.mit.edu). CRISPR/Cas9 InDels percentage calculation through TIDE V.3.3.0 (https://tide.nki.nl/).<br>Western blot quantification performed using ImageJ 1.52a (downloaded from http://imagej.nih.gov/ij). |

Figure 3 panel a was created in BioRender. Fiorillo, E. (2025) https://BioRender.com/guualid (Agreement number LU294VQC18).
Extended Figure 6 panel c was created in BioRender. Fiorillo, E. (2025) https://BioRender.com/guualid (Agreement number  YV294W1S4K).

For manuscripts utilizing custom algorithms or software that are central to the research but not yet described in published literature, software must be made available to editors and reviewers. We strongly encourage code deposition in a community repository (e.g. GitHub). See the Nature Portfolio guidelines for submitting code & software for further information.

## Data

Policy information about availability of data

All manuscripts must include a data availability statement. This statement should provide the following information, where applicable:

- Accession codes, unique identifiers, or web links for publicly available datasets
- A description of any restrictions on data availability
- For clinical datasets or third party data, please ensure that the statement adheres to our policy

Data are provided as Source Data or Supplementary Information.  The data generated in this study are available at https://figshare.com/s/d89f8f9f90d3a76eccb9. Population genetics analyses in non-Sardinian populations have been conducted using the1000 Genomes Project-phase 3 data, available at https://ftp.1000genomes.ebi.ac.uk/vol1/ftp/release/20130502/.  Public GWAS summary statistics are available at https://www.ebi.ac.uk/gwas/home. The human genome assembly hs37d5 (GRCh37 with decoy sequences), used to generate the transcriptome reference for expression quantification, is available at https://ftp.1000genomes.ebi.ac.uk/vol1/ftp/technical/reference/phase2_reference_assembly_sequence/. Gene annotation (GENCODE version 14) used to generate the transcriptome reference is available at https://www.gencodegenes.org.

## Research involving human participants, their data, or biological material

Policy information about studies with human participants or human data. See also policy information about sex, gender (identity/presentation), and sexual orientation and race, ethnicity and racism.

| Reporting on sex and gender | In the SardiNIA project, we collected the information about the sex of the participants and we used it as a covariate in the association models. No information about gender has been collected. |
|---|---|
| Reporting on race, ethnicity, or other socially relevant groupings | No information about socially relevant grouping has been collected and used. |
| Population characteristics | The individuals analyzed here belong to the SardiNIA general population cohort, a longitudinal study of ageing conditions in ~8,000 individuals from 4 nearby villages in the Lanusei Valley on the island of Sardinia, Italy. The SardiNIA study also allows GWAS of thousands of cross-sectional quantitative traits generated from tests/samples collected during the periodic visits (every 3-4 years) of the volunteers, and it is also a bioresource of samples that can be collected for functional analyses from individuals recalled based on their genetic profile. The 6,824 individuals analyzed here were randomly selected from the SardiNIA participants, aged between 18 and 102 years, of whom about 43% are male.<br>Given the variability with age and the differences between men and women in the haematological measurements included in this work, all genetic analyses, especially the association analyses, were adjusted for the confounding effect of sex and age. In addition, to take into account the high degree of ageing of the SardiNIA cohort, its effect on the characteristics was taken into account and adjusted by including the term "age2" in the statistical models. Furthermore, given the strong effect of β0 mutations on haematological variables and their high frequency in Sardinia, we also adjusted the analyses for the presence of at least one of the three β0 mutations (β039 (rs11549407), HBB:c.20delA (rs63749819) and HBB:c.315+1G>A (rs33945777)). |
| Recruitment | Samples for the GWAS of quantitative haematological traits reported in the study were randomly recruited from the cohort participants, none were selected or grouped by health status or other criteria.<br>As non-random sampling of related individuals is a potential source of bias, we diversified the appointment schedule of related individuals.<br>Sample recruitment for specific biological experiments was performed on individuals carrying specific rs112233623 genotypes, in particular i) homozygotes for the Increase of Expression (IoE) allele rs9349205-A and for the Decrease of Expressionn (DoE) allele rs112233623-T (IoE/IoE|DoE/DoE); ii) homozygotes for the IoE allele rs9349205-A and for the WT allele rs112233623-C (IoE/IoE|WT/WT); and as a control group, iii) homozygotes for the major alleles rs9349205-G and rs112233623-C (WT/WT|WT/WT). An additional group of iv) male individuals hemizygous for the G6PD deficient allele (rs5030868-A, Mediterranean variant) and carrying the wild type genotype at rs9349205 and rs112233623 (WT/WT|WT/WT, G6PD-Med) was included. |
| Ethics oversight | Sardinian Regional Ethics Committee (prot. n. 2171/CE). |

Note that full information on the approval of the study protocol must also be provided in the manuscript.

# Field-specific reporting

Please select the one below that is the best fit for your research. If you are not sure, read the appropriate sections before making your selection.

☒ Life sciences          ☐ Behavioural & social sciences          ☐ Ecological, evolutionary & environmental sciences

For a reference copy of the document with all sections, see nature.com/documents/nr-reporting-summary-flat.pdf

# Life sciences study design

All studies must disclose on these points even when the disclosure is negative.

| Sample size | The sample size used for the GWAS and selection analyses corresponded to the vast majority (N=6,824) of participants in the SardiNIA cohort and was not determined a priori, although previous analyses indicated that it provided adequate statistical power for the effects tested. No statistical methods were used to predetermine sample sizes for biological experiments.

The sample size in experiments derived from the SardiNIA cohort depended on the number of volunteers who were homozygous for the CCND3 IoE and DoE genotypes and negative for beta- and alpha-thalassaemia, as well as the G6PD Mediterranean deficiency mutation. Participants were also selected based on their willingness to undergo blood collection, including repeated sampling over time. As participation and sampling were voluntary, the number of samples available for each experimental setup could not be predetermined or controlled by the investigators. For other experiment we performed three or more independent biological replicates and three or more technical replicates. We choose a sample size of at least three replicates because it represent the minimal threshold where the standard error of the mean stabilizes sufficiently for basic inferential statistics.

More in detail: sample sizes for the biological experiments performed to assess P. falciparum invasion and maturation on volunteer RBC cultures were based on the number of volunteers actually carrying the genotypes of interest in the cohort and their availability for resampling for the proposed experiments (for the IoE/IoE|WT/WT genotype the sample size is 14; for the WT/WT|WT/WT genotype the sample size is 7; for the IoE/IoE|DoE/DoE genotype the sample size is 17; for the WT/WT|WT/WT|G6PD-Med genotype the sample size is 5).
For Reactive Oxigen Species (ROS) measurement  the number of samples derived from volunteers carrying the genotypes of interest in the cohort was N=10 for the IoE/IoE|WT/WT genotype, N=7 for the WT/WT|WT/WT genotype, N=14 for the IoE/IoE|DoE/DoE genotype and N=5 for the WT/WT|WT/WT|G6PD-Med genotype.
Sample sizes for the biological experiments performed with in vitro-differentiated primary erythrocytes precursors (erythroblasts) derived from volunteers carrying the genotypes of interest to measure the CCND3 mRNA (N=11 for each group) and the effects on cell cycle and erythroid differentiation by flow cytometry is N=4 for each group (rs112233623-C homozygous and rs112233623-T homozygous).
The expression of CCND3 mRNA in primary erythroid cultures at day 7 and day 9 of erythroid in vitro differentiation, assessed by flow cytometry and RTqPCR, was performed using rs112233623-C homozigous genotype.
The functional studies in HUDEP-2 cell line have been performed using at least the minimum number of replicates (N=3) to perform statistics:
- Luciferase assay to valuate the influence of the rs112233623-T allele on the binding of transcription factors SMAD3 and GATA1 was performed in at least three independent experiments.
- ChIP-assay to demonstrate the binding of SMAD3 in presence of the rs112233623-C allele was valuated in three independent biological replicates.
- TGF-β induced stimulation of SMAD3 expression and consequentely of CCND3 mRNAs synthesis performed was  estimated in at least three independent biological replicates.
- CRISPR/Cas9-induced depletion of SMAD3 and SMAD2 genes and the consequentely expression of Cyclin-D3 mRNA and protein was measured in three independent experiments. |

| Data exclusions | To avoid possible confounding factors, individuals carrying variants reported to be protective against malaria in Sardinia at loci other than CCND3 (including beta or alpha globin and G6PD deficiency) were excluded from enrolment in this study a priori. No data were excluded from the analysis of any experiment. |

| Replication | The biological independence of the experiments in this study was achieved through:  the independent splitting of each cell type; the use of primary cells from different donors; different batches of experiments.  All association results for haematological traits are replicated in published, independent population studies (Ulirsch et al, Nat Genet 2019; Vuckovic et al, Cell 2020).The association results on haemoglobin levels represent an extension of those reported in Danjou et al, Nat Genet 2015. |

| Randomization | Samples were collected from participants in the SardiNIA cohort based on their selected genotype.
Each data point obtained from Plasmodium falciparum in vitro blood cultures was compared with the corresponding data from the wild-type control genotype for the variants considered in the study. For experiments involving cell lines, the negative control consisted of wild-type cells that had not undergone any genomic modification or treatment. All experimental data were normalised to the negative control values, except for the data derived from volunteers, which are reported solely according to their distinct genotypes. |

| Blinding | No group allocation has been performed and all genetic analyses were blinded to any information on the enrolled participants.
Blinding was performed when counting the parasitaemia of P. falciparum in in vitro blood cultures. Other analyses were objective measures, free from bias. |

# Reporting for specific materials, systems and methods

We require information from authors about some types of materials, experimental systems and methods used in many studies. Here, indicate whether each material, system or method listed is relevant to your study. If you are not sure if a list item applies to your research, read the appropriate section before selecting a response.

## Materials & experimental systems

| n/a | Involved in the study |
|-----|----------------------|
| ☐ | ☒ Antibodies |
| ☐ | ☒ Eukaryotic cell lines |
| ☒ | ☐ Palaeontology and archaeology |
| ☒ | ☐ Animals and other organisms |
| ☒ | ☐ Clinical data |
| ☒ | ☐ Dual use research of concern |
| ☒ | ☐ Plants |

## Methods

| n/a | Involved in the study |
|-----|----------------------|
| ☒ | ☐ ChIP-seq |
| ☐ | ☒ Flow cytometry |
| ☒ | ☐ MRI-based neuroimaging |

## Antibodies

| | |
|---|---|
| Antibodies used | PC7-conjugated anti-human CD235a (GPA-PC7 1:50) supplied by Beckman Coulter, clone # 11E4B-7-6, catalog #A71564; PE-conjugated anti-Band3 (Band3-PE 1:100) supplied by IBGRL, clone # BRIC 6, catalog # 9439CSPE; APC-conjugated anti-CD49d (α-4 integrin-APC 1:20) supplied by Beckman Coulter, clone # HP2/1, catalog #B01682; Anti-human GATA1 (GATA1 1:10) supplied by Santa Cruz Biotechnology, clone #N1 catalog #sc-266X; Anti-influenza hemagglutinin epitope tag HA.11(SMAD3-HA 1:10), supplied by Biolegend, clone #16B12, catalog #MMS-101P; Anti-human CyclinD3 (CyclinD3 1:250) supplied by Santa Cruz Biotechnology, clone #D-7, catalog #sc-6283; Anti-human Smad2 (Smad2 1:500) supplied by Santa Cruz Biotechnology, clone #S-20, catalog #sc-6200; Anti-beta-actin (beta-actin 1:1000) supplied by Santa Cruz Biotechnology, clone #C4, catalog #sc-47778; Rabbit anti-mouse IgG-HRP (1:20000), supplied by Santa Cruz Biotechnology, catalog #sc2005; Goat anti-rabbit IgG-HRP(1:20000), supplied by Santa Cruz Biotechnology, catalog #sc2004; Rabbit anti-goat IgG-HRP (1:20000), supplied by Santa Cruz Biotechnology, catalog #sc-2768; Anti-human SMAD3 (SMAD3 10µg in 100 µL for IP and 1:1000 for WB) supplied by Cell Signaling Technology clone #C67H9, catalog #9523; Anti-human GATA1 (GATA1 10µg in 100 µL) supplied by Abcam catalog #ab11852; Normal Rabbit IgG, (IgG, 1µg in 100 µL), supplied by Cell Signaling Technology, catalog #2729S. |
| Validation | CD235a-PC7: https://www.beckman.it/search#q=A71564&t=coveo-tab-techdocs Band3-PE:https://nhsbtdbe.blob.core.windows.net/umbraco-assets-corp/6596/bric-6.pdf (Van den Akker E et al., Haematologica 2010; Lamarque et al., CDD 2022) α-4 integrin-APC:https://www.beckman.it/search#q=B01682&t=coveo-tab-techdocs Anti-GATA1 Ab: https://www.scbt.com/p/gata-1-antibody-n1 (Daniels et al. Nat Commun 2023; Arlet et al. Nature 2014) Anti-HA tag Ab: https://www.biolegend.com/fr-fr/products/purified-anti-ha-11-epitope-tag-antibody-11374?GroupID=GROUP26 (Paik I, et al. Nat Commun. 2019; Wan J, et al. Nat Commun. 2019) Anti-CyclinD3 Ab: https://www.scbt.com/p/cyclin-d3-antibody-d-7 (Simoneschi et al. Nature 2021; Wu et al. Nat Cancer 2021) Anti-Smad2 Ab: https://www.scbt.com/p/smad2-antibody-s-20 (He et al. Oncogene 2001; Papoutsidakis et al. Int. J. Oncol2020) Anti-beta-actin Ab: https://www.scbt.com/it/p/beta-actin-antibody-c4 (Fu et al. Nature 2025; Anerillas et al. Nat Aging 2023). Anti-mouse: https://www.scbt.com/it/p/goat-anti-mouse-igg-hrp (Liubet al. Nat Commun 2020; Chang et al. Nat Commun 2015) Anti-rabbit: https://www.scbt.com/p/goat-anti-rabbit-igg-hrp (Sun et al. Nat Commun 2017; Chimge et al. Nat Commun 2016) Anti-goat: https://www.scbt.com/p/rabbit-anti-goat-igg-hrp (Morimoto et al. Nat Commun 2021; .Totsuka et al. Nat Commun 2014) Anti-SMAD3 Ab: https://media.cellsignal.com/coa/9523/7/9523-lot-7-coa.pdf Anti-GATA1Ab: https://www.abcam.com/en-us/products/primary-antibodies/gata1-antibody-ab11852?srsltid=AfmBOopQf9m1uxEEUEx40e5t1geCaLYQ-vMUmr5SlhFU4mdB85C5lk_U#tab=datasheet (Deorfler et al. Nat Genet. 2021; King et al Nat Commun. 2021) IgG: https://media.cellsignal.com/coa/2729/11/2729-lot-11-coa.pdf |

## Eukaryotic cell lines

Policy information about cell lines and Sex and Gender in Research

| | |
|---|---|
| Cell line source(s) | The Human Umbilical cord-Derived Erythroid Progenitors HUDEP-2 cell line was obtained through a collaboration with Prof. Yukio Nakamura and Rio Kurita (RIKEN RBC Cell Bank, Japan). The human embryonic kidney HEK-293T cell line and the African green monkey kidney fibroblast-like cell line COS7 were obtained commercially (ATCC). |
| Authentication | HUDEP-2, HEK293T and COS7 cell lines were not authenticated. |
| Mycoplasma contamination | All cell lines tested negative for mycoplasma contamination. |
| Commonly misidentified lines (See ICLAC register) | The cell lines used are not listed as commonly misidentified. |

## Plants

Seed stocks

n/a

Novel plant genotypes

n/a

Authentication

n/a

## Flow Cytometry

### Plots

Confirm that:

☒ The axis labels state the marker and fluorochrome used (e.g. CD4-FITC).

☒ The axis scales are clearly visible. Include numbers along axes only for bottom left plot of group (a 'group' is an analysis of identical markers).

☒ All plots are contour plots with outliers or pseudocolor plots.

☒ A numerical value for number of cells or percentage (with statistics) is provided.

### Methodology

Sample preparation

To minimize time-dependent artifacts, erythroid-specific markers were analyzed within 2 hours after cell collection.

For flow cytometry phenotypic analysis of erythroid cultured cells, 3x10^5 erythroid cells were washed twice in FACS Flow (BD Biosciences), resuspended in 50 µL, and stained with the following anti-human antibodies: anti-CD49d-APC (Beckman Coulter), anti-CD235a-PC7 (Beckman Coulter), and anti-Band3-PE (IBGRL). The erythroid cultured cells were then washed with FACS Flow, and 20,000 events were recorded to examine two time points, day 7 and day 9 stages of erythroid differentiation. All staining steps were performed at room temperature, with efforts made to minimize exposure of the samples to ambient light.

Erythroblasts were analyzed using Propidium Iodide (PI; BD Biosciences) for cell cycle. At two specific time points cells were harvested, washed and fixed for PI staining. PI was added 10 minutes prior to Flow Cytometer acquisition.

Instrument

BD Accuri™ C6 Plus (BD Biosciences) and FACSCanto (BDbioscience).

Software

FlowJo V10.0.0 (BD Biosciences)

Cell population abundance

In vitro differentiated human erythroblasts were not sorted using flow cytometry, but evaluated for differentiation (CD49d vs Band3) on total CD235a-PC7 positive cells and for cell cycle for the abundance of PI signal (G0-G1<S<G2-M).

Gating strategy

Linear FSC-A versus linear SSC-A was used to gate the erythroid cultured cells. Doublet discrimination was performed based on light scatter (FSC-H versus FSC-A, SSC-H versus SSC-A). For the single-cell population, CD235a-PC7 versus FSC-A was used to gate on CD235a-PC7 positive cells. To identify basophilic cells (primarily CD49d-APC positive) and polychromatophilic cells (double-positive for Band3-PE and CD49d-APC), we plotted Band3-PE versus CD49d-APC on CD235a-PC7 positive cells.

FSC-A versus SSC-A were used to separate erythroblasts from debris. FSC-A versus FSC-H was used to gate single cells. Single cells were visualized on a histogram plot and linear Propidium Iodide-PerCP-CY5.5 was used to gate cell cycle phases (G0-G1, S and G2-M).

☒ Tick this box to confirm that a figure exemplifying the gating strategy is provided in the Supplementary Information.

