## [Peer Review File · Nature]

Evidence for malaria protection by reduced Cyclin-D3 erythroid expression

Corresponding Author: Professor Francesco Cucca

Version 1:

Reviewer comments:

Referee #1

(Remarks to the Author)

In this study, Maria Marini et al., show that a polymorphism in the cyclin D3 gene (CCND3) enhancer affects susceptibility of individuals to malaria. The authors demonstrate that this polymorphism decreases binding of SMAD3 while increasing the repressive interaction of GATA1 with the CCND3 enhancer. Consequently, decreased expression of cyclin D3 inhibits expansion of erythroblasts resulting in larger erythrocytes containing more hemoglobin. Moreover, inhibition of cyclin D3 increases the levels of reactive oxygen species (ROS), which also contributes to the malaria resistance.

This is a very elegant and important study, which will be of broad interest. Showing that a component of the core cell cycle machinery influences the susceptibility to malaria is very novel, very interesting and unexpected.

From the cell cycle standpoint, the results presented here are convincing and make perfect sense. However, I have a few suggestions for improvement:

1. In Fig. 1, the authors show that rs112233623-T variant decreases cyclin D3 mRNA expression. It would be critical to show that the levels of cyclin D3 protein are also reduced.
2. Can the authors demonstrate, using chromatin immunoprecipitation, that binding of SMAD3 and GATA1 to rs112233623-T enhancer is decreased and increased, respectively? So far, this important evidence is only shown using get supershifts.
3. If I am not mistaken, Figure 2 shows that the "wild-type" G/C enhancer is activated by SMAD3 and repressed by GATA1. In contrast, the A/T enhancer shows drastically decreased basal activity, and it is no longer affected by SMAD3 or GATA1. This is different from authors' interpretation: "By contrast, the same CCND3 luciferase constructs expressed in HUDEP-2 cells but co-transfected with GATA1 in combination with its coactivator FOG1 27 showed almost complete suppression of luciferase activity of the CCND3 enhancer bearing the rs112233623-T derived allele. The activity of the CCND3 enhancer was also reduced, albeit to a lesser extent, for luciferase constructs carrying the wild-type rs112233623-C allele (Figure 2b)".
4. CDK4 was shown to directly phosphorylate and inhibit SMAD3 (PMID: 15241418; PMID: 20736297). While erythroid cells likely express CDK6, not CDK4, as the major cyclin D-partner, it may be worth to discuss the findings of Marini et al in the context of these past publications.
5. Minor point. The authors wrote: "in addition to affecting the cell cycle, Cyclin D3 also binds to cyclin-dependent kinase 6 (CDK6), forming a complex that has been shown in studies in cancer cells to phosphorylate and thereby reduce the activity of the rate-limiting glycolytic enzymes 6-phosphofructokinase". Cyclin D3 affects BOTH cell cycle and cell metabolism through CDK6, so the correct version is: "In addition to affecting the cell cycle, cyclin D3-CDK6 complex phosphorylates...."

Referee #2

(Remarks to the Author)

Thank you for the opportunity to review this paper.

This study is a follow-up to an earlier paper (ref 1) which implicated (among others) SNP rs112233623 in gene CCND3 (which encodes Cyclin D3) as associated with HbA2 levels in a sample from Sardinia. The authors present this as motivated by a set of interrelated questions about the phenotypic effects of the gene and of this variant, the transcriptomic mechanism of action, and the interplay between this variant and another known RBC-trait-associated SNP in CCND3. Given that rs112233623 has been observed at high frequency (~10%) in Sardinia but much lower elsewhere, they also ask about natural selection. The study says it addresses all these questions, including that rs112233623 causes loss of function of CCND3 by creating a GATA1 repressor. Further the claim is that they demonstrate a role for rs112233623 T allele in malaria resistance, which would be a plausible cause of the selection effect. These claims are well represented in the paper title: "A positively selected DNA variant reducing Cyclin D3 function in erythrocyte precursors protects against malaria."

I found this paper well-written and the analysis appears high quality and clearly described. It has quite clear figures and the findings are mostly well supported (some minor comments on this below), and indeed the paper addresses the above questions. The results are also substantial and potentially are important given the link to Cyclin D3, which is an area of current research due to its role in cell cycle regulation and cancer. Overall I'm satisfied that this is a substantial piece of work with a potentially important finding.

I do have one major concern and some minor ones, which I've detailed below.

1. Main concern: my only main concern is whether the study really shows that rs112233623 T is protective against malaria (as the title claims). Linked to this are some comments about the discussion of malaria protection e.g. G6PD.

To explain this concern, I note the paper does NOT carry out an epidemiological study of the effects of rs112233623 T on currently malaria-exposed populations - as the authors note this would be effectively impossible due to the lack of malaria in Sardinia now. Thus, although the authors have done as many analyses as they reasonably can to establish this protection, actual data associating the allele to disease outcomes is lacking.

This might not be thought worrying given the diverse supporting results presented, but there have been a relatively long list of findings of 'malaria protective' variants arising from in vitro laboratory studies, which have apparently failed to replicate in epidemiological studies. This indicates that translating from in vitro to population effects is not always straightforward. For example: Ma et al (Cell 2019 doi:10.1016/j.cell.2018.02.047) identified the PIEZO1 E756Del mutation based on effects on RBC phenotype in mice and humans, and showed 'protection' in in vitro parasite experiments. They also reported evidence for natural selection. Yet when tested in large population studies it shows little effect <https://doi.org/10.1016/j.cell.2018.02.047>. This is just one of many variants that have been proposed as malaria-protective yet don't seem to actually be associated with protection, or not strongly, in global populations (e.g. those tested in <https://doi.org/10.1038/ng.3107>).

(In fact, G6PD deficiency which the authors use as a direct comparator to the rs112233623 effect, is itself somewhat of this type. It is certainly well-known as an established malaria-protective locus, yet the actual evidence from population association studies has revealed complexities including possibly differing effects in males and females, and possible differences between malaria subtypes and between continents. (See e.g. doi.org/10.1038/srep45963, doi.org/10.1038/ng.3107). This is despite the fact that lab experiments indicate reduced growth e.g. <https://doi.org/10.1073/pnas.80.1.298>.)

There could be lots of reasons for these discrepancies; one important one might be that parasite populations are genetically variable i.e. different parasites behave differently. And we already know that parasite genetic variation interacts with human protective polymorphisms in real populations (doi.org/10.1038/s41586-021-04288-3). (This incidentally raises a question/concern about the Palo Alto isolate used for experiments in the current study, see below for details.)

With this in mind here are some things I think the study could do to improve their approach to this:

1. First I think it would be appropriate to discuss some of the above concerns in the main text. At the moment the text makes it sound as though G6PD is a well-known, protective mutation, giving support to their current finding. It would be beneficial to better reflect the observed nuances in this. Concretely:

1.1 I'd suggest replace 'an established protective trait against malaria' in the abstract, with more equivocal wording, 'a trait which is known to associate with malaria protection in some settings' or similar: the current wording over-simplifies the current evidence.

1.2 I'd also suggest to alter the wording in the discussion paragraph on lines 375-381. This line says 'Remarkably, defense against severe malaria...' but the authors have certainly not established defense against severe malaria. (I also think 'remarkably' is redundant?) It would be better to clarify that the mechanism of action against parasite growth appears to be the same as that conferred by G6Pd deficiency. (Another similar point about this paragraph: I think the mechanism of therapeutic action of common antimalarials is not fully understood, whereas the paragraph sounds like it is, so although I think the authors are broadly correct about ROS mechanism, this could be improved with more careful wording.)

2. The malaria in vitro expt would be considerably strengthened if it could be carried out in a genetically variable set of parasites, not just one isolate. Specific comments:

2.1. it should be stated more clearly what isolate was used for these experiments. Currently the paper says 'the genetically

stable reference *P. falciparum* (Palo Alto) strain' but the most commonly-used line for functional experiments is 3D7 which is also the genome assembly reference. Moreover there is confusion about what 'Palo Alto' actually means: [https://doi.org/10.1016/0166-6851\(91\)90176-7](https://doi.org/10.1016/0166-6851(91)90176-7). I suggest state its full and short name (symbol) and potentially its reference number at BEI Resources / ATCC, or other repository.

2.2. If this line is indeed the same as 'Uganda Palo Alto FUP/H' (which is a long-established line derived from an infection in Uganda in the 1960s), then I note that this line is known to carry mutations that alter its ability to infect individuals with sickle mutations (doi.org/10.1038/s41586-021-04288-3). It is presumed that there is an underlying effect on invasion/growth. Since as the authors mention, ROS might be part of the sickle malaria phenotype, this would make it particularly useful to include comparator parasite line(s) that do not carry these mutations.

2.3. Regardless of 2.2, the results of the in vitro experiment would be considerably more compelling if the same behaviour could be demonstrated in multiple isolates. Given G6Pd geographical variation, I'd suggest including both African and Asian-derived parasite isolates - is this possible?

2.4. And these limitations of in vitro experiments / the alternative possibilities should be discussed. (Epidemiological studies have other limitations as well, that's another possible discussion point.)

3. If point 2 could be improved to a substantial array of parasites, then I'd be somewhat more relaxed about the paper title claiming that the variant 'protects against malaria'. This is to some extent an editorial decision. I personally still think caution may be advised given the long list of variants with prior evidence of selection and/or in vitro results that have not turned out to associate with strong protection in population studies; still, it's maybe recognised that titles can state hypotheses so this is probably an editorial decision. However for clarity: my feeling is that expanding the in vitro experiment to a diverse set of parasite lines that better represent current populations should be a prerequisite for making this claim, and if not the title should be re-worded.

Minor comments:

- Line 100 onwards: I think the data analysed here are the same sample set as used for the original discovery GWAS, this should be stated.
- Section on lines 150 onwards: The transcriptomic experiment seems strong but would definitely benefit by increasing sample size. (It currently has only N=4 per genotype and which limits interpretability of the statistical analysis). Is it possible to increase sample size for this expt?
- Section on lines 150 onwards: I was confused at first because GATA1 also has an excitatory effect, but here the effect is repressive. Would it be possible to briefly mention these known opposing roles of GATA1 to clarify this, e.g. why is it expected to be repressive rather than an upregulator here?
- Line 212 and Fig 2b, I felt the statistical analysis could better match the text or be better reported, i.e. line 212 "the basal level of expression of constructs bearing the rs112233623-T allele showed no response to transactivation" should be supported with a direct test of difference between the levels in T genotype cells (i.e. report the statistical details: N, P-value and test used). Related to this is that it's hard to tell what comparisons the statistical comparison bars on Fig 2b correspond to. If I understand right, the comparison for the sentence above is actually the rightmost one, which has a single asterisk so there is some (not very strong) evidence of a difference in 'basal' level of expression? (I'm also unsure of the need for the word 'basal' here, which I didn't understand, couldn't this word simply be removed?). In any case it would be helpful to be clearer about what comparisons have been conducted, perhaps the best way is to provide full statistical results for all comparisons as a supp table (or Figure Source data) including the non-significant comparisons.
- Lines 215, 'almost complete suppression' does not seem supported by Fig 2b, which shows a similar level of expression for GATA1+FOG to the other rows in the graph (if slightly smaller) - certainly not zero expression. For example is this difference statistically significant?
- Line 288 and 764, as above please state more clearly the parasite line being used (I think it might be Uganda Palo Alto, FUP/H?).
- Line 788-789, state how were the three observer's measurements combined?
- Fig 4 legend, definition of Maturation (at 120hpi) seems at odds with Methods (defines at 4,96 and 144 hpi). (If the cycle is 48 hours then measuring at 144hpi seems appropriate.)
- I enjoyed reading the popgen material, and agree the paper presents compelling evidence of recent +ve selection of this allele/haplotype in Sardinia, and uses best-practice methods for this analysis.

Referee #3

(Remarks to the Author)

This paper describes an analysis of a largely Sardinian-specific variant, rs112233623 which has previously been reported to be associated with a variety of red blood cell traits including HbA2, erythrocyte count and various other traits. The authors make several linked claims; specifically:

- Functional work showing that this variant regulates CCND3 expression, affecting erythropoiesis via binding of SMAD3/GATA1.
- Via the effect of decreased CCND3 expression on ROS, the variant confers resistance to malaria.
- The variant experienced natural selection due to this protective effect, explaining why it is common in Sardinia but rare elsewhere.

I would say that the main contributions are to show (in vitro) that rs112233623 protects against malaria, and plausible evidence that it was under recent selection. Overall, I find this paper of moderate interest – it's interesting to see another adaptation to malaria, but it's not particularly surprising and the claims about the potential for improved treatment seem somewhat speculative.

In more detail:

I find it reasonably convincing that rs112233623 is a causal variant for the hematological phenotypes, although that is not novel.

I can't really comment much on the details of the functional work, but I hope one of the other reviewers will. I am a bit concerned that the donor samples used for some of the experiments are not matched or controlled for sex, age etc...

To me the experiments in Figure 4 seem to show reasonably convincingly that rs112233623 is likely to confer some resistance to malaria, perhaps of a similar magnitude to G6PD deficiency.

The evidence for selection is not exceptionally strong but I think probably sufficient for the claims made in the paper given a high prior that a variant protective against malaria would be selected (assuming the previous parts are convincing).

Major comments:

1) Even though rs112233623 is rare outside Sardinia, the larger sample sizes in other studies mean that you still have much more power to investigate it in other samples. For example, it's ~10 times more common in Sardinia than the rest of Europe, but but the big GWAS are more like ~100x the sample size so they have much more power; For example, even though it's only ~0.9% frequency, it has a P-value of $1e-195$ for RBC in Vuckovic et al 2020 (PMID 32888494), which is much more significant than reported in Table 1. Presumably mapping in those cohorts (e.g. UK Biobank), you would be better powered to estimate effect sizes, fine-map and detect conditional effects. As the authors point out (lines 76-78), this has in fact already been done and already identified rs112233623 and rs9349205 as independent signals, so I'm not sure the association results add much to that.

2) The effect on expression in Figure 1 is pretty weak – the P-value is marginal and there is one large outlier for the CC genotype. A larger sample would help, alternatively is there a reason not to look in the whole-blood RNA seq data from this cohort from Pala et al (2017)? According to GTEx, CCND3 is reasonably highly expressed in whole blood. The browser for the results of that paper appears to be no longer available, so I can't check but since some of the authors overlap I assume they have access to the data.

3) In the analyses based on patient-derived RBC or erythrocyte progenitors (e.g. Figure 1 and 4), are you controlling for sex, age ABO and other technical factors in the analysis? I appreciate you controlled for globin deletions and GDP6, but there are many other things associated with malaria response? In addition, it is a shame you did not look at heterozygotes – it would be relevant for the evolutionary questions whether the effects are additive or not (in many other malaria resistance alleles the protective effect is additive, while the negative consequences are recessive, explaining why they experience balancing selection).

4) Selection analysis – I think the approach is generally reasonable. The analyses seem to largely converge on a P-value somewhere in the 99th percentile. It's not particularly significant, but I think given that the authors are only testing a single variant and there's a fairly strong prior that if it's protective against malaria, it would be under selection, I think that's fine. I think the genomic null procedure makes sense. One comment is that personally I find the CLUES analysis the most convincing, so I'm surprised that they only ran this with such a small sample size. They have thousands of genomes available. This approach can certainly scale to more than 100 individuals. Similarly, since you have a large sample of present-day people you could impute rs112233623 in the ancient Sardinian data and incorporate that into the analysis (e.g. with CLUES2) – that would give you more confidence about the strength and timing of selection (is it consistent with the historical introduction of Malaria?). That would all strengthen the manuscript, that said, I don't think any of this is strictly necessary to support the claim as it is that it's under selection which I think is ok.

5) Data availability: The raw data are not publicly available (or at least it's not clear if they are). The summary statistics from the original 2015 association study are not publicly available. The summary statistics from this study are not provided. The raw data from the experiments performed in this study are not provided. The code used to analyze the data is not provided.

Consequently, the statement that “Data are provided as Source Data or Supplementary Information.” Is false. It is absolutely not acceptable to write that data are available on request. At minimum, the authors need to make publicly available all the association summary statistics, the raw data from the experiments, and the code used to analyze those data.

Minor comments.

6) In the mixed model association analysis (line 810), do you use the genome-wide GRM or the local GRM to control for stratification. It's not actually clear to me which is more appropriate. Usually one would use the genome-wide GRM, but if you think the region is under strong selection one might argue it's more appropriate to use the local one.

7) I find the LoF/GoF terminology a bit confusing. The authors mean expression decreasing or increasing variants – I tend to think of LoF variants as coding, and I feel like that's the standard in the field, but I could be wrong.

8) Line 347: This is not an example of epistatic dominance. If I understand correctly, you are claiming that the effects are additive.

9) Supplementary information “negative selection of the rs9349205-A” – I don't think this is very convincing, since there's no actual analysis here. It's also not negative or purifying selection (i.e. selection against new deleterious mutations). The variant is already segregating - it's just positive selection for the other allele. The evidence in Figure 4 that rs9349205 affects invasion etc.. is statistically very weak (e.g. it would not be significant if you corrected for the number of tests in that experiment).

10) Several small but annoying issues with statistical analysis and data presentation:

- All the boxplots should show the individual points instead (there are not so many); e.g. figure 1a and c, figure 4. Box plots are particularly misleading in Figure 4 because the number of observations is so different.
- Please add confidence intervals to all plots (e.g. Figure 3, Extended data figure 2)
- Please give a confidence interval for the selection coefficient (and give the P-value in the main text at line 264; the likelihood ratio is not easily interpretable to the non-statistical reader.
- Please give exact P-values in e.g. Figure 4
- Extended data figure 2: why are there two panels? But Sardinia is in both?

11) Typos etc...

- Figure 3; “Kenia”, “East Africa”, etc... “frequency level”
- “Tuscans in Italy” in Figure 3 but “Tuscany in Italy” in Extended data figure 2
- Line 131: “has no any primary effect”
- Line 813: “sex, age²” – assume you mean, “sex, age AND age²”

Referee #4

(Remarks to the Author)

This interesting manuscript by Marini et al addresses an important question in the malaria field, namely the relationship between human genetic variation and malaria susceptibility. As the authors note, understanding how malaria has selected for human genetic variation has the potential to lead to the discovery of novel therapeutic targets. In this work a variant in an enhancer for the cell cycle gene CCND3 is investigated. The variant has a high allele frequency in Sardinians (as compared to Europeans) suggesting it may have resulted from positive selection. Prior studies have linked a different variant in the CCND3 enhancer with hematologic traits, and the variant of question in this work has also been previously implicated to have effects on RBC counts in data from the UK Biobank.

Strengths of the paper are 1) the demonstration of an association between the rs112233623 allele and hematologic traits in Sardinians (ext data figure 1), the characterization of CCND3 expression and cell cycle parameters in erythroblasts derived from individuals with the TT vs WT background, and 3) the demonstration that the T allele leads to reduced enhancer activity in the luciferase promoter assays, as well as the involvement of the transcription factors of SMAD3 and GATA1, providing a potential mechanism for the reduced CCND3 expression in the TT allele.

Additionally, the authors analyze haplotype data and allele frequencies and come to the conclusion that there is evidence for the TT allele being under positive selection. They hypothesize that this may be due to malaria, and attempt to demonstrate this using Plasmodium invasion and growth assays.

Major points:

In Figure 1b, cultures of erythroid progenitor cells are characterized for their cell cycle stages. While a potentially interesting finding is reported where the cell cycle stages differ between the genetic backgrounds only at the polychromatic erythroblast stage and not at the earlier basophilic erythroblast stage, this conclusion is not firm as it appears that the cells were not actually staged, but rather assumed to be at a certain stage based on their day of differentiation. To be more definitive about about potential for different developmental/cell cycle phenotypes in the genetic backgrounds, should perform a more comprehensive flow cytometry-based analysis for erythrocyte stage (as in Hu et al Blood, 2013), and also provide growth curves for the erythroblasts from the two different backgrounds. If indeed there is a cell cycle defect in the TT background, this should be apparent in these standard assays. While there are some additional data presented in Ext data figure 3, only

one genetic background is shown (presumably WT) so the relevance to this issue is not clear.

One of the major conclusions of the paper is the report of a causal role of the genetic variant rs112233623-T on variation in hematological traits as well as protection against malaria. While some of the results presented in Figure 4 appear to support the conclusion about malaria, the methods as described and the data presented raise concern that the conclusion may not be adequately supported by evidence. Specifically, the authors report that the parasite assays were initiated at 5% schizont-stage parasitemia in 2% hematocrit. In this setting, the schizonts would be expected to egress from the mother cell, and re-invade at a rate $\sim 5x$, which would lead to $\sim 25\%$ ring parasitemia at 24 hours. This is an extremely high parasitemia, which would most certainly crash within that cycle (by 48 hours). The pictures in Figure 4 d-e are consistent with crashed parasites, so the interpretation that the parasites in these pictures are "delayed" is debatable. Moreover, there is a discrepancy between the pictures and the graphs (panels 4f-g). The images in Figure 4 reportedly show parasitemia at 72 hours and 120 hours, and the graphs show parasitemia at 120 hours. The pictures at 120 hours show parasitemia in the range of 25->50%, whereas the graphs report parasitemia of 1-4%. Why are these values so different? Furthermore, the authors report that they counted parasitemia several times during the assay (24 hr, 72hr, 96 hr, 120 hr, and 144hr) yet only a single timepoint is reported (120 hr), raising additional concern that there is a lack of transparency in data reporting. Given these significant issues, I am not able to evaluate the veracity of these results and any conclusions to be made from them. While the idea to try to correlate the ROS status of the cells with parasite growth phenotypes is reasonable, with the concern about the parasite growth phenotypes not being accurate, the significance of these potential correlations is not clear. At a minimum, the methods should be clarified, data from all timepoints should be reported, whether or not the smears were counted blindly should be reported, and how the data from the three observers was treated should be reported. Additionally, it does not seem feasible that all of these assays were done in a single batch, so the authors should describe how the batches were treated. If indeed these experiments were conducted using 5% initial parasitemia and then incubated for 120 hours, then to avoid crashing the experiments would need to be repeated using a lower starting parasitemia and shorter assay (presumably one round of invasion/ maturation should demonstrate any phenotype).

The error bars in Figure 1a are not defined in the legend. While it is mentioned that N=4, only one data point is visible for each genetic background, so it is not possible to fully visualize the spread of the data. It would be helpful to plot all data points for the other graphs as well (e.g. Figure 4 f-h).

In addition to the association profiles shown in Ext Data Figure 1, the conclusions regarding impact of the rs112233623-T allele on hematologic parameters would be strengthened if complete blood counts and RBC indices for blood donors from the two genetic backgrounds were provided, especially since many such donors were used for the parasite assays.

While the analysis of allele frequency differentiation and haplotype variation as evidence for positive selection is interesting, this is outside the scope of my expertise and should be assessed by a reviewer with specific expertise in this area.

Version 2:

Reviewer comments:

Referee #2

(Remarks to the Author)

Thank you for addressing these points. I particularly appreciate the careful re-wording of the title / abstract and discussion relating to what is shown re: a malaria protective effect and also in relation to G6PD; and the addition of in vitro experiments in a wider range of Pf lines. These have addressed most of my concerns.

I have a couple of minor remaining points where I would still seek clarification:

- In response 2.1, the authors say "We utilised the FUP Uganda Palo Alto strain" and that a section clarifying this has been added to Methods (presumably lines 938-945). However, as far as I could tell the revised ms and this section still don't state which version is being used. (BEI Resources / ATCC supply three such strains, which go by Uganda Palo Alto Hawaii, Cayenne and Roche respectively; the reference cited in this question indicates that different versions of FUP do have different genotypes and are probably distinct isolates.) This could be important, for example, for reproducibility. Therefore, I don't think this question has been addressed. Can the authors specify which version of FUP was used in their experiments? (If this does not correspond to a known identifier such as those from BEI, providing information on the provenance of the isolate used in the lab be given? If this information is not known, that's not necessarily a showstopper particularly given the additional data on other lines that has been provided in the revised manuscript, but I think should be clearly stated.

- Re: the question of how the blood smear measurements from different operators were combined, I don't think this has been addressed either. (The question is: given there are six blood smears in total counted by three operators independently, state how these counts were combined to form one overall count for the sample.) This may well be in the cited WHO manual but should be stated here anyway for clarity (I think another reviewer has asked a similar question.)

- I did also review some of the figure code via the figshare link provided and noted (see comments in separate box) that not all of the code is directly runnable - this should probably be fixed to make this code maximally useful.

Finally, some minor typos I noticed:

Line 315 - sentence fragment seems wrongf, "selected among not carriers of variants"

Line 411 - 'stricking' -> 'striking'

Fig 4c Throphozoites -> Trophozoites

Thanks!

(Remarks on code availability)

R scripts and source data for the main and Extended Data figures have been provided in the above figshare.

I very briefly reviewed the code for some of these figures to verify that it looked sensible and that it produces output matching that in the figures.

Not all of the scripts run completely without errors, for example:

- Figure 4b/script_2024.R attempts to load parassitemia_allinone_june2024.txt, which does not exist in the directory, so that the last plot fails

- Figure 3 - Extended data figure 2/script.R attempts to load files like frequenze_aplotipi.txt and frequenze_aplotipi.EUR.txt which similarly don't exist in the folder.

This should probably be reviewed / corrected so that all scripts are runnable, if this code repository is to be most useful. However, for other figures, in general these tests confirmed that the code produces the output shown in the paper for the figures I tested (primarily Fig 1, and Fig 4).

(I did note that, relative to the script output, the figures in the paper have had some additional annotations and other cosmetic changes added, including addition of lines/asterisks denoting P-values, rotation of axis labels, and so on.)

Referee #3

(Remarks to the Author)

I appreciate the work that the authors have done on the revision; addressing many of the points that the other reviewers and I raised in the first round.

From my perspective as a human population geneticist, I find the evidence for selection on rs112233623 specifically to be moderately compelling, but not at the level one would require in, say, a genome-wide scan. If this were just a random variant I would not be convinced that it had been under selection (a test like SDS that is more sensitive to very recent selection might strengthen that) However, if the variant were indeed protective in vivo against malaria, then my prior that it would be under selection would be quite high, and then I would find the claim of selection to be overall convincing.

On the other hand, reading the other reviewers' comments, it seems that they find the evidence for in vivo protection moderately compelling, but think that the population genetic evidence strengthens that claim. And, the paper motivates the investigation of malaria based on the evidence of selection. So it seems like there's a bit of circularity in that some reviewers think that the evidence for selection supports the protective effect, but I think the protective effect supports selection. As one of the other reviewers points out there are "many variants that have been proposed as malaria-protective yet don't seem to actually be associated with protection".

On balance I think the two lines of argument complement each other and I think I am convinced by the paper as a whole. But I can't really evaluate whether the in vitro evidence for protection against malaria is convincing that there would be a substantial in vivo effect. If it's not, I would probably feel differently.

Other comments:

1) I'm still a bit uncertain about the eQTL evidence in figure 1a. The P-value is $1.0e-2$, which doesn't seem that different than the $2.1e-2$ they report in Pala et al. which they say is "not an eQTL". In contrast, they say that rs4623235 is an eQTL in Pala et al. but then in Figure 1a, they don't control for rs4623235 genotype (or rs9349205, for that matter).

I appreciate that it is a different cell type, but I don't think those eQTL are likely to be entirely cell-type specific. If rs4623235 is really an eQTL, I would think that it would have an effect on expression here too.

Similarly, later in the response they say that "of course [sex] was controlled for as a covariate in the association analyses", but I don't think that sex is controlled in Figure 1a? Unless they are regressing out the effect of sex and performing the Wilcoxon tests on the residuals?

2) The point about the contrasting effects of rs9349205 and rs112233623 is well-made and interesting. However, I maintain this is not an example of epistasis. Epistasis would involve an interaction between the two genotypes. A difference in the marginal effect of a variant due to LD with another causal variant is not epistasis. The claim here is that the effects of rs9349205 and rs112233623 are additive. If you think there is an epistatic effect, you should test for that explicitly (i.e. phenotype \sim rs9349205 * rs112233623).

I also think that the discussion on lines 140-169 is unnecessarily long. The point can be made much more succinctly.

3) Minor comment but I don't see why they can't run things for more than 4 days on their cluster. I don't see why they need to include TSI and YRI in their relate analysis since rs112233623 is not present there. Even if you really can't run for more than 4 days, you could run 300 Sardinians instead of 100/100/100 TSI/YRI/Sardinians. That said I don't think it will make a difference to the selection claim.

4) Line 1685: "The much greater length of the haplotype with the derived allele is obvious, evidencing positive selection." I don't think this is actually obvious because rarer variants will naturally have longer haplotypes. The relevant comparison is with variants of matched frequency in panels a-c. I don't think you can draw any conclusions from eyeballing the haplotype lengths in panel d.

5) Terminology, but I don't think that variants with a small effect on mRNA expression would be called LoF/GoF either under the common usage today, or under the classical definitions (which relate to protein function or organismal phenotypes). One can easily imagine a variant that increased gene expression leading to a loss-of-function at the protein level. "pLoF" just makes it even more confusing because in the human genetics literature it usually stands for "predicted-loss-of-function" not "partial-loss-of-function". Others might disagree, but why not use "expression increasing/decreasing" or something to avoid confusion?

(Remarks on code availability)

Referee #4

(Remarks to the Author)

This work investigates a variant in an enhancer for CCND3. The variant has a high allele frequency in Sardinians (as compared to Europeans) suggesting it may have resulted from positive selection. Prior studies have linked a different variant in the CCND3 enhancer with hematologic traits, and the variant of question in this work has also been previously implicated to have effects on RBC counts in data from the UK Biobank. Some of the key results the paper are 1) the demonstration of an association between the rs112233623-T allele and hematologic traits in Sardinians, the characterization of CCND3 expression and cell cycle parameters in erythroblasts derived from individuals with the TT vs WT background, and 3) the demonstration that the T allele leads to reduced enhancer activity, as well as the involvement of the transcription factors of SMAD3 and GATA1, providing a potential mechanism for the reduced CCND3 expression in the TT allele. Additionally, the T allele is found to correlate with reduced growth of *P. falciparum*, correlated with elevated ROS levels, suggesting a mechanism of protection similar to that of G6PD.

Based on the feedback that starting with 5% schizont parasitemia would lead to parasite crashing, the authors have now stated that they made a reporting error, and that actually all of these experiments were performed at 2.5% starting schizont parasitemia, not 5%. With regards to the other aspects of Figure 4, the reviewers report that their response required "a massive and careful re-evaluation of all the results", which resulted in a replacement of this figure with a "new" figure 4. In response to a concern about potential batch effects, they have also now reported that the experiments were performed in 5 batches, and that each genotype was represented in each batch and that careful attention was paid to controlling for experimental conditions (however I could not find any report of which sample was in which batch).

1. Parasite invasion is highly variable depending on the precise culture conditions in the prior cycles, stress of the density gradient procedure to isolate schizonts, etc., and therefore the same parasitemia would not be expected from batch to batch even when genetic background and experimental conditions are maintained constant. Yet the graph in the new Figure 4b shows almost identical parasitemia within each genetic background at 48 hours (1st cycle), and the experimental variation remains minimal at subsequent timepoints, which is highly unexpected if performed in 5 different batches. This is also evident in the raw data provided for Figure 4b (excel file).

2. As it is not clear what the "massive and careful re-evaluation of all the results" entailed, I reviewed the new Figure 4 in the context of the original Figure 4. In the reviewing the "new" Figure 4 in the context of the original Figure 4, several substantial discrepancies in the data become apparent. For example, in the original Figure 4 the % invasion shown in the graph was defined as the ring parasitemia at 120 hr, and the % maturation was defined as the troph parasitemia at 120 hr. For the WT/WT/WT/WT genotype, the parasitemia for both was 3-4%, suggesting a total parasitemia of 6-8% with a ratio of 50:50 rings and trophs (original Figure 4f-g). However in the new Figure 4C, the graph shows that this genotype had 90% rings at the 120 hr timepoint, highlighting a substantial discrepancy. Moreover, while the authors notably no longer show parasitemia data for 120 hr in the Figure, in reviewing their raw data for the 120 hr timepoint, the parasitemias for the WT/WT/WT/WT are all ~17% (as opposed to the report of 6-8% at this timepoint for this genotype in the original Figure 4f-g). There is no explanation provided for these and many other discrepancies in the data for the two versions of Figure 4.

3. The authors conclude that the phenotype for GoF/GoF/PLoF/PLoF became apparent in the second cycle. However, as apparent from the raw data the difference in parasitemia was substantial from the first timepoint at 24 hours, which is also evident in the figure at the 48 hr timepoint (~4% parasitemia in the WT/WT/WT/WT in 48 hr trophozoites, and significantly reduced parasitemia in the GoF/GoF/PLoF/PLoF background). The staging data indicates that the parasites in the deficient background progressed normally from 100% rings to ~ 100% trophs in that first cycle, so this implies that there was an issue with invasion, consistent with the raw data.

The conclusions of this paper would be strengthened by a detailed explanation for the discrepancies in the reporting of the data underlying Figure 4, as well as an explanation for the apparent lack of variation in the raw data for different batches of the parasite growth assays.

(Remarks on code availability)

Referee #5

(Remarks to the Author)

I have examined the revised manuscript. I think that the authors have correctly addressed the comments of the previous reviewer and changed the manuscript accordingly.

Michele Pagano

(Remarks on code availability)

Version 3:

Reviewer comments:

Referee #2

(Remarks to the Author)

The authors have addressed the concerns I raised previously.

In re-reviewing this I considered this interesting comment by referee #3:

"[...] reading the other reviewers' comments, it seems that they find the evidence for in vivo protection moderately compelling, but think that the population genetic evidence strengthens that claim. And, the paper motivates the investigation of malaria based on the evidence of selection. So it seems like there's a bit of circularity in that some reviewers think that the evidence for selection supports the protective effect, but I think the protective effect supports selection. As one of the other reviewers points out there are "many variants that have been proposed as malaria-protective yet don't seem to actually be associated with protection". On balance I think the two lines of argument complement each other and I think I am convinced by the paper as a whole. But I can't really evaluate whether the in vitro evidence for protection against malaria is convincing that there would be a substantial in vivo effect. [...]"

In light of this I have reviewed the main pieces of evidence presented in the paper and how strong I think each is and have summarised this in comments to the editor (which I'm happy for them to share if it is felt necessary, I just don't think it needs an additional response.). As detailed there my conclusions are similar to those of referee #3 and to what I thought previously, i.e. I think the major conclusions of the paper are supported and that the paper does reflect 'Evidence for protection against malaria by an adaptive DNA variant reducing Cyclin-D3 function in erythroblasts' as stated in the title.

It is true that effects on parasite growth observed in lab-based in vitro settings do not always correspond to real-world population effects, but at the same time a population-level study is effectively impossible here so this may be the best evidence that can be assembled. (The addition of experiments in more than one strain in the previous revision was helpful in this regard.)

Suggestions:

I'm conscious this review has had multiple rounds already and am not suggesting any major changes or response. However the above did raise a couple of additional comments/questions relevant to the evidence presented, which I'm stating here:

I. Lines 310-311: Other known malaria-protective variants (HbS, HbC, O blood group, ATP2B4) are not mentioned here, why not? Is it possible these variants differ between the groups? It would be helpful to state this or acknowledge these are not measured. (HbS must surely be accounted for?)

II. Methods section on line 793 and onwards: For the experiments that use blood or cells derived from human volunteers (typo: line 794 'volunteers') it would be helpful to add more detail about the ascertainment of individuals, including

- what process was used to determine the participants that donated blood/cells? For example, state fully which genetic variants were examined (similar to the above question about lines 310-311) and how individuals were then selected based

on
these genotypes?

- And also clarify what overlap there was in the donors used in the various different experiments (RNA and protein expression, Pf growth)? (In particular, was it the same individuals?)

III. This statement from the summary paragraph (lines 35-38) does not seem to be supported by direct evidence in the paper: "Furthermore, the reduced number of erythroblast divisions seen in rs112233623-T carriers is likely to counteract the well-established delayed erythroblast differentiation promoted by *P. falciparum* to ensure its gametocytogenesis, thereby alleviating diserythropoiesis and preventing parasite sexual maturation and transmission.". I'd suggest clarifying that this is a proposed mechanism, not something the paper establishes.

(Remarks on code availability)

Referee #3

(Remarks to the Author)

I think the authors have addressed most of my concerns. As I said earlier I think the evidence for selection is somewhat modest in itself but on balance reasonable as part of the whole package.

I do think that the LoF/GoF terminology is incorrect, but up to them...

(Remarks on code availability)

Referee #4

(Remarks to the Author)

The authors have provided more details to explain their methods, findings, and conclusions for the parasite assays in Figure 4, including the batch assignments for each sample, which has strengthened the paper. I appreciate their efforts in this regard.

In reviewing the manuscript, I feel comfortable that the authors have sufficiently addressed the prior reviewer feedback.

However a statement in the Summary that the cell cycle alterations in the LoF allele is "likely to counteract the well-established delayed erythroblast differentiation induced by *P. falciparum*" is not supported by any evidence in the paper, and may come across as misleading. I strongly suggest that this sentence be removed from the Summary and instead saved as speculation in the Discussion section. The authors do not do any infections in erythroblasts and do not attempt to examine the impact of *P. falciparum* infection on erythroblast differentiation; this could be investigated if the authors felt strongly about including it, but I do not think it should be mentioned in the Summary in the absence of any data.

(Remarks on code availability)

- Referee #1: cell cycle

In this study, Maria Marini et al., show that a polymorphism in the cyclin D3 gene (CCND3) enhancer affects susceptibility of individuals to malaria. The authors demonstrate that this polymorphism decreases binding of SMAD3 while increasing the repressive interaction of GATA1 with the CCND3 enhancer. Consequently, decreased expression of cyclin D3 inhibits expansion of erythroblasts resulting in larger erythrocytes containing more hemoglobin.

Moreover, inhibition of cyclin D3 increases the levels of reactive oxygen species (ROS), which also contributes to the malaria resistance. This is a very elegant and important study, which will be of broad interest. Showing that a component of the core cell cycle machinery influences the susceptibility to malaria is very novel, very interesting and unexpected.

From the cell cycle standpoint, the results presented here are convincing and make perfect sense.

Many thanks for the positive comments and constructive suggestions.

However, I have a few suggestions for improvement:

1. In Fig. 1, the authors show that rs112233623-T variant decreases cyclin D3 mRNA expression. It would be critical to show that the levels of cyclin D3 protein are also reduced.

We agree that assessment of Cyclin-D3 protein levels may strengthen the results, and to measure these we recalled volunteers to establish new erythroblast cultures derived from individuals homozygous for either the wild-type rs112233623-C allele or the partial-loss-of-function (PLoF) rs112233623-T allele. We employed Western blot analysis to quantify CCND3 protein levels, confirming the hypothesis that the PLoF rs112233623-T allele results in a substantial reduction in CCND3 protein expression, approximately 40% lower relative to the wild-type allele.

These findings have been incorporated as Extended Data Figure 4a and are detailed in the figure legend starting at line 1299: "Western blot assay showing Cyclin-D3 expression in erythroblasts derived from individuals homozygous for the WT rs112233623-C allele (N=4) versus the PLoF rs112233623-T allele (N=5); human β -actin was used as a loading control. Band intensities in the immunoblots were quantified utilising ImageJ software using the same detection time point for both

signals (1 minute) and the differences were expressed as ratio of Cyclin-D3 to β -actin means for each group."

Additionally, in the main text on lines 177 onwards, we detail: "In erythroblasts from individuals with the plausible PLoF rs112233623-TT genotype, CCND3 expression was indeed reduced at both mRNA and protein levels compared to what as observed in erythroblasts from individuals with the rs112233623-CC genotype".

2. Can the authors demonstrate, using chromatin immunoprecipitation, that binding of SMAD3 and GATA1 to rs112233623-T enhancer is decreased and increased, respectively? So far, this important evidence is only shown using get supershifts.

To address this query, we did perform chromatin immunoprecipitation (ChIP) assays on erythroblasts derived from volunteers homozygous for either the wild-type rs112233623-C allele or the loss-of-function (PLoF) rs112233623-T allele.

These assays confirmed the differential binding of SMAD3 in the DNA region surrounding rs112233623. Specifically, we observed decreased binding of SMAD3 to DNA with the rs112233623-T allele compared to the wild-type allele. Regarding GATA1, our data indicate that its binding was relatively similar with either allele. This stability in GATA1 binding may be due to compensatory mechanisms, potentially involving an adjacent GATA1 motif located 30 bp upstream of rs112233623.

These findings have been incorporated as Extended Data Figure 5 c-d and are detailed in the corresponding figure legend starting at line 1331 "*Chromatin immunoprecipitation (ChIP-qPCR) for SMAD3 or GATA1 binding to the CCND3 enhancer region surrounding rs112233623 (CCND3) and to a SimpleChIP® Human α Satellite as negative control in erythroblasts derived from subjects homozygous for the PLoF rs112233623-T allele versus the WT rs112233623-C allele. ChIP was performed using an antibody against SMAD3 (c) or GATA1 (d). Results are represented as percentage of input and nonspecific IgG used as negative control. The mean \pm s.e.m is shown (N= 3 or 2 independent biological replicates). A two Sample t-test was used; significant differences are indicated (**P<0.01; **** P<.0001, ns, not significant).*"

Additionally, in the main text on lines 235 onwards, we detail “*The differential binding of SMAD3 to the region surrounding rs112233623 was confirmed by Chromatin ImmunoPrecipitation (ChIP) assays in erythroblasts from volunteers homozygous for either the WT rs112233623-C or the derived PLoF rs112233623-T allele (Extended Data Figure 5c). However, GATA1 binding remained relatively unchanged, presumably compensated by a nearby GATA motif (Extended Data Figure 5d)*”.

3. If I am not mistaken, Figure 2 shows that the “wild-type” G/C enhancer is activated by SMAD3 and repressed by GATA1. In contrast, the A/T enhancer shows drastically decreased basal activity, and it is no longer affected by SMAD3 or GATA1. This is different from authors’ interpretation: “By contrast, the same CCND3 luciferase constructs expressed in HUDEP-2 cells but co-transfected with GATA1 in combination with its coactivator FOG1 showed almost complete suppression of luciferase activity of the CCND3 enhancer bearing the rs112233623-T derived allele. The activity of the CCND3 enhancer was also reduced, albeit to a lesser extent, for luciferase constructs carrying the wild-type rs112233623-C allele (Figure 2b)”.

We thank this reviewer for this comment regarding the enhancer activity profiles shown in Figure 2, which has prompted us to revise the text to more accurately reflect the observed experimental results.

The revised text, from line 245, now reads: “*By contrast, the same CCND3 luciferase constructs expressed in HUDEP-2 cells but co-transfected with GATA1 in combination with its coactivator FOG129 showed an even greater reduction in luciferase activity of the CCND3 enhancer bearing the rs112233623-T derived allele*”.

4. CDK4 was shown to directly phosphorylate and inhibit SMAD3 (PMID: 15241418; PMID: 20736297). While erythroid cells likely express CDK6, not CDK4, as the major cyclin D-partner, it may be worth to discuss the findings of Marini et al in the context of these past publications.

Motivated by this interesting question, we re-analysed public RNA-seq datasets from *in vitro* cultured erythroblasts (PMID: 24637361; PMID: 31189107) together with our real-time qPCR analyses performed in HUDEP-2 cells and *in vitro* cultured erythroblasts. These analyses consistently show co-expression of both CDK4 and CDK6 in erythroblasts at the same developmental stages as SMAD3. This is also consistent with the findings of Uras *et al.* (PMID: 28255017), who showed that

while proliferating erythroblasts express both CDK4 and CDK6, CDK4 is degraded during erythroid maturation, leaving CDK6 as the predominant kinase in mature erythrocytes.

However, while it may be interesting to discuss these observations, which help to understand the different roles of these kinases in erythroid cell cycling and maturation, and their potential interactive dynamics with SMAD3, we also feel that this may be somewhat beyond the main scope and emphasis of this paper -- and certainly difficult to reconcile with the editorial limit on number of words, so that we have not added further to the discussion.

5. Minor point. The authors wrote: "in addition to affecting the cell cycle, Cyclin D3 also binds to cyclin-dependent kinase 6 (CDK6), forming a complex that has been shown in studies in cancer cells to phosphorylate and thereby reduce the activity of the rate-limiting glycolytic enzymes 6-phosphofructokinase". Cyclin D3 affects BOTH cell cycle and cell metabolism through CDK6, so the correct version is: "In addition to affecting the cell cycle, cyclin D3- CDK6 complex phosphorylates...."

We have revised the text accordingly so that from lines 83 onwards it reads: "*In addition to affecting the cell cycle, the Cyclin-D3/CDK6 complex was found to phosphorylate and thereby reduce the activity of the rate-limiting glycolytic pathway enzymes 6-phosphofructokinase (PFK1) and pyruvate kinase-M2 (PKM2) in cancer cells¹⁶*"

Referee #2 human infectious disease genetics, genetic epidemiology

Thank you for the opportunity to review this paper.

This study is a follow-up to an earlier paper (ref 1) which implicated (among others) SNP rs112233623 in gene CCND3 (which encodes Cyclin D3) as associated with HbA2 levels in a sample from Sardinia. The authors present this as motivated by a set of interrelated questions about the phenotypic effects of the gene and of this variant, the transcriptomic mechanism of action, and the interplay between this variant and another known RBC-trait associated SNP in CCND3. Given that rs112233623 has been observed at high frequency (~10%) in Sardinia but much lower

elsewhere, they also ask about natural selection. The study says it addresses all these questions, including that rs112233623 causes loss of function of CCND3) by creating a GATA1 repressor. Further the claim is that they demonstrate a role for rs112233623 T allele in malaria resistance, which would be a plausible cause of the selection effect. These claims are well represented in the paper title: "A positively selected DNA variant reducing Cyclin D3 function in erythrocyte precursors protects against malaria."

I found this paper well-written and the analysis appears high quality and clearly described. It has quite clear figures and the findings are mostly well supported (some minor comments on this below), and indeed the paper addresses the above questions. The results are also substantial and potentially are important given the link to Cyclin D3, which is an area of current research due to its role in cell cycle regulation and cancer. Overall I'm satisfied that this is a substantial piece of work with a potentially important finding.

We thank the referee for the comments and all helpful suggestions.

I do have one major concern and some minor ones, which I've detailed below.

1. Main concern: my only main concern is whether the study really shows that rs112233623 T is protective against malaria (as the title claims). Linked to this are some comments about the discussion of malaria protection e.g. G6PD.

To explain this concern, I note the paper does NOT carry out an epidemiological study of the effects of rs112233623 T on currently malaria-exposed populations - as the authors note this would be effectively impossible due to the lack of malaria in Sardinia now. Thus, although the authors have done as many analyses as they reasonably can to establish this protection, actual data associating the allele to disease outcomes is lacking.

This might not be thought worrying given the diverse supporting results presented, but there have been a relatively long list of findings of 'malaria protective' variants arising from in vitro laboratory studies, which have apparently failed to replicate in epidemiological studies. This indicates that translating from in vitro to population effects is not always straightforward. For example: Ma et al (Cell 2019 doi:10.1016/j.cell.2018.02.047) identified the PIEZO1 E756Del mutation based on effects on RBC phenotype in mice and humans, and showed 'protection' in in vitro parasite experiments. They also reported evidence for natural selection. Yet when tested in

large population studies it shows little effect <https://doi.org/10.1016%2Fj.cell.2018.02.047>. This is just one of many variants that have been proposed as malaria-protective yet don't seem to actually be associated with protection, or not strongly, in global populations (e.g. those tested in <https://doi.org/10.1038/ng.3107>).

One of the causes may be the high frequency of “malaria protective” variants evidenced in malaria endemic areas with potential cumulative and/or synergistic protective actions exerted by known and unknown mutations.

(In fact, G6PD deficiency which the authors use as a direct comparator to the rs112233623 effect, is itself somewhat of this type. It is certainly well-known as an established malaria protective locus, yet the actual evidence from population association studies has revealed complexities including possibly differing effects in males and females, and possible differences between malaria subtypes and between continents. (See e.g. <https://doi.org/10.1038/srep45963>, doi.org/10.1038/ng.3107). This is despite the fact that lab experiments indicate reduced growth e.g. <https://doi.org/10.1073/pnas.80.1.298>.)

There could be lots of reasons for these discrepancies; one important one might be that parasite populations are genetically variable i.e. different parasites behave differently. And we already know that parasite genetic variation interacts with human protective polymorphisms in real populations (doi.org/10.1038/s41586-021-04288-3). (This incidentally raises a question/concern about the Palo Alto isolate used for experiments in the current study, see below for details.)

As we discussed in our manuscript, and as this reviewer points out, epidemiological studies in areas where malaria is still endemic to confirm our laboratory findings of antimalarial effects for rs112233623 T at the population level are ruled out for this allelic variant because the allele is absent there. And we fully agree that even for other variants that are present in these areas, there are several factors that may explain why genetic epidemiological studies in some cases fail to confirm effects derived from *in vitro* studies. As pointed out by the reviewer, these discrepancies may be due to the genetic diversity of malaria parasites in different areas or the potential interactions between host polymorphisms prevalent in malaria-endemic regions, such as HbS, HbC, alpha and beta thalassaemias, G6PD deficiency, ovalocytosis and elliptocytosis. In addition, the widespread use of modern antimalarial therapies and blood transfusions, which have greatly improved clinical outcomes, have introduced confounding variables that complicate the design and interpretation of robust clinical trials.

Furthermore, the lack of concordance between laboratory and clinical findings in some cases may be due to *in vitro* findings that are actually false positives unrelated to malaria resistance. But given the absolute impossibility of conducting epidemiological studies to confirm our laboratory findings, we believe that the risk of false positives is minimised in our study by 1) the unusually high degree of robustness and consistency of the *in vitro* effects on parasitaemia, demonstrated even more clearly in this revised version of the manuscript, 2) a proposed dual mechanism of action against the deleterious effects of malaria, in particular severe anaemia, based on the observed metabolic and cell cycle effects of rs112233623 T, that adds to the likelihood of effects on pathogenesis; and 3) consistent evidence from all the tests of positive selection used, providing an indirect but clear-cut link to a clinical benefit of the variant in the past history of Sardinia. The benefit is very unlikely to be due to other causes, given the concordant laboratory data and the epidemiological history of this island population, which endured the highest prevalence of malaria in Europe.

At the same time, we tried both experimentally and analytically to rule out possible confounding variables that we could control for. For example, from the outset we stratified subjects with different genotypes to account not only for variability in the specific erythroid enhancer of Cyclin-D3 that we were considering, but also to exclude other variants at other loci that have been implicated in malaria resistance and that are present in the Sardinian population. Similarly, we were very careful to avoid batch effects when recruiting individuals on the basis of their genotype to experimentally test the *P. falciparum* life cycle on their erythrocytes *in vitro*, and to ensure that all assays were performed under strictly controlled, standardised and homogeneous conditions, using exactly the same experimental protocols, including identical reagent batches, instrument settings and environmental parameters. We now appreciate the reviewer's suggestion and have followed it to exclude variability in effects associated with different *P. falciparum* isolates.

Overall, we believe that our study may provide the most extensive laboratory and evolutionary evidence ever presented for the effects of variants newly associated with malaria resistance, and the possible importance of the results is increased by the interesting clinical and therapeutic implications of what we hope will ultimately be a link to the clinic with the development of new therapies based on our observations.

With this in mind here are some things I think the study could do to improve their approach to this:

1. First I think it would be appropriate to discuss some of the above concerns in the main text. At the moment the text makes it sound as though G6PD is a well-known, protective mutation, giving

support to their current finding. It would be beneficial to better reflect the observed nuances in this.

Concretely:

1.1 I'd suggest replace 'an established protective trait against malaria' in the abstract, with more equivocal wording, 'a trait which is known to associate with malaria protection in some settings' or similar: the current wording over-simplifies the current evidence.

1.1 We agree with all of the reviewer's suggestions, including the one about caution above, and have revised the text accordingly. For example, in the abstract the sentence now reads as suggested:

“..we indeed demonstrate impaired parasite growth, with impairment correlating with ROS levels. This mimics what we see in erythrocytes from individuals deficient in the PPP enzyme G6PD, a trait known to be associated with malaria protection in some settings, and highlights a common ROS-based mechanism of malaria resistance.”

1.2 I'd also suggest to alter the wording in the discussion paragraph on lines 375-381. This line says 'Remarkably, defense against severe malaria...' but the authors have certainly not established defense against severe malaria. (I also think 'remarkably' is redundant?) It would be better to clarify that the mechanism of action against parasite growth appears to be the same as that conferred by G6Pd deficiency (Another similar point about this paragraph: I think the mechanism of therapeutic action of common antimalarials is not fully understood, whereas the paragraph sounds like it is, so although I think the authors are broadly correct about ROS mechanism, this could be improved with more careful wording.)

1.2. Again we agree and we rephrased these sentences in the discussion as follows (from lines 415 onwards): *“The strong inhibitory effect of CCND3 PLoF on parasitaemia in vitro, together with evidence for positive selection in a population with a long history of endemic malaria, supports a protective effect against severe malaria, which is inherently associated with high levels of parasitaemia. It is also suggestive that this predicted defence against severe malaria by increased ROS production taps into the same pathway as G6PD deficiency and is one of the main proposed mechanisms of action of the most commonly used antimalarial drugs, including quinolines, atovaquone, and artemisinin and its derivatives^{44,45}. Accordingly, all have been shown to increase oxidative stress to levels that compromise malarial parasites, leading to the damage of cellular*

*components including lipids, proteins, and DNA, and ultimately cell death via apoptosis or necrosis*⁴⁶.”

2. The malaria in vitro expt would be considerably strengthened if it could be carried out in a genetically variable set of parasites, not just one isolate.

2. We appreciate the reviewer's suggestion to expand our *in vitro* experiments by incorporating genetically diverse *P. falciparum* isolates. Although the primary goal of our work was to investigate the mechanistic effects of the PLoF rs112233623 mutation in a controlled experimental setting using the well-characterised Palo Alto strain of *P. falciparum* (FUP), we fully acknowledge the importance of assessing genetic variability in *P. falciparum* parasites. The Palo Alto strain was specifically chosen as a standardised model for our research due to its extensive validation in our previous studies, which demonstrated similar development in normal red blood cells and those affected by sickle cell trait, beta-thalassaemia, alpha-thalassaemia, and HbH disease (PMID:15280204).

With considerable logistical effort, we therefore recalled volunteers for each genotype under consideration and performed the infection experiments on their red cells with the Palo Alto strain and two other *P. falciparum* isolates, 3D7 (African Strain) and DD2 (Southeast Asian Strain), in parallel. The results across all three strains (Palo Alto, 3D7, and DD2) were consistent, demonstrating that the observed reduction in parasitaemia in rs112233623-T carriers was not strain-dependent. This strongly reinforces the biological plausibility of the protective effect and addresses concerns about potential isolate-specific biases. These expanded findings have been incorporated into the revised manuscript from lines 341 further strengthening the robustness of our conclusions: “*Consistent results on parasitaemia levels were also obtained by repeating the same experiments in RBCs from a smaller group of recalled individuals carrying the same genotypes of interest as in the previous experiments, but infected in parallel with the Uganda Palo Alto strain of Plasmodium falciparum (FUP) and two other different P.falciparum isolates (3D7 and DD2). This experiment, shown in Extended Data Figure 10, suggests that the effects due to genetic differences in CCND3, and in particular the inhibitory effect of PLoF rs112233623-T on P.falciparum growth, are unlikely to be influenced by genetic differences in P.falciparum.*”

2.1. it should be stated more clearly what isolate was used for these experiments. Currently the paper says 'the genetically stable reference P. falciparum (Palo Alto) strain' but the most

commonly-used -line for functional experiments is 3D7 which is also the genome assembly reference. Moreover there is confusion about what 'Palo Alto' actually means: [https://doi.org/10.1016/0166-6851\(91\)90176-7](https://doi.org/10.1016/0166-6851(91)90176-7). I suggest state its full and short name (symbol) and potentially its reference number at BEI Resources / ATCC, or other repository.

2.1. Thank you for your comments. We utilised the FUP Uganda Palo Alto strain. This is explicitly stated in the Methods section at lines 938: "*In vitro studies were conducted using the genetically stable reference FUP strain (mycoplasma-free) according to protocols previously reported*⁷⁰ "

Additionally, we acknowledge the potential confusion surrounding the term 'Palo Alto', as highlighted in the reviewer's referenced study (PMID: 1944415). To clarify this, we have included a brief statement in Methods section (*P.falciparum* culture paragraph) acknowledging the historical ambiguity in nomenclature and explaining why the FUP strain was specifically chosen.

2.2. If this line is indeed the same as 'Uganda Palo Alto FUP/H' (which is a long-established line derived from an infection in Uganda in the 1960s), then I note that this line is known to carry mutations that alter its ability to infect individuals with sickle mutations (doi.org/10.1038/s41586-021-04288-3).

2.2. See above. As an aside, sickle cell mutations are basically absent in native Sardinians and in all the volunteers of the SardiNIA study experimentally studied here.

It is presumed that there is an underlying effect on invasion/growth. Since as the authors mention, ROS might be part of the sickle malaria phenotype, this would make it particularly useful to include comparator parasite line(s) that do not carry these mutations.

See above. We addressed this point as well by repeating experiments in RBCs from Palo Alto (FUP) as well as in 3D7 (African) and DD2 (Southeast Asian) isolates.

2.3. Regardless of 2.2, the results of the in vitro experiment would be considerably more compelling if the same behaviour could be demonstrated in multiple isolates. Given G6Pd geographically variation, I'd suggest including both African and Asian-derived parasite isolates - is this possible?

2.3. To address this concern, we performed additional *in vitro* infection experiments using two genetically distinct *P. falciparum* isolates, in addition to the originally used Uganda Palo Alto (FUP) strain: 3D7 (African strain) and DD2 (Southeast Asian strain)

.2.4. And these limitations of in vitro experiments / the alternative possibilities should be discussed. (Epidemiological studies have other limitations as well, that's another possible discussion point.)

2.4. See points discussed above, and in general we have made the entire narrative across the whole study more cautious.

3. If point 2 could be improved to a substantial array of parasites, then I'd be somewhat more relaxed about the paper title claiming that the variant 'protects against malaria'. This is to some extent an editorial decision. I personally still think caution may be advised given the long list of variants with prior evidence of selection and/or in vitro results that have not turned out to associate with strong protection in population studies; still, it's maybe recognised that titles can state hypotheses so this is probably an editorial decision. However for clarity: my feeling is that expanding the in vitro experiment to a diverse set of parasite lines that better represent current populations should be a prerequisite for making this claim, and if not the title should be re-worded.

3. Again, we recognise the importance of caution and have revised the title accordingly.

Minor comments:

- Line 100 onwards: I think the data analysed here are the same sample set as used for the original discovery GWAS, this should be stated

- We have updated the main text to make this clearer.

From lines 104 onwards, we report: " Extending our previous work on the genetic architecture of haemoglobin levels in 6,305 individuals¹¹ we examined genetic associations in the CCND3 gene region for a larger set of haematological traits in 6,824 individuals from the SardiNIA general population cohort¹² (Methods). "

- Section on lines 150 onwards: The transcriptomic experiment seems strong but would definitely benefit by increasing sample size. (It currently has only N=4 per genotype and which limits interpretability of the statistical analysis). Is it possible to increase sample size for this expt?

- Following the suggestion, we have successfully recruited additional volunteers and established new erythroid cultures from individuals homozygous for either the wild-type rs112233623-C allele or the

PLoF rs112233623-T allele. This has allowed us to increase the number of samples per genotype from 4 to 11, thereby enhancing the robustness of our findings.

- Section on lines 150 onwards: I was confused at first because GATA1 also has an excitory effect, but here the effect is repressive. Would it be possible to briefly mention these known opposing roles of GATA1 to clarify this, e.g. why is it expected to be repressive rather than an upregulator here?

- We appreciate the reviewer's comment highlighting the complex regulatory roles of GATA-1, which is known to function as both an activator and repressor in various gene expression contexts. To illuminate the repressive role of GATA-1 described in our manuscript, we included a literature reference in the text (PMID: 19941827), which outlines through ChIP-seq analysis distinct groups of genes that are activated or repressed by GATA-1 in erythroid cells. Their results show that many genes associated with cell cycle processes fall into the negatively regulated group, illustrating the ability of GATA-1 to act as a repressor in certain regulatory scenarios, particularly in erythropoiesis. We believe this clarification will improve our understanding of why GATA-1 shows a repressive effect in our study, contrasting its activating role in other contexts.

- Line 212 and Fig 2b, I felt the statistical analysis could better match the text or be better reported, i.e. line 212 "the basal level of expression of constructs bearing the rs112233623-T allele showed no response to transactivation" should be supported with a direct test of difference between the levels in T genotype cells (i.e. report the statistical details: N, P-value and test used). Related to this is that it's hard to tell what comparisons the statistical comparison bars on Fig 2b correspond to. If I understand right, the comparison for the sentence above is actually the rightmost one, which has a single asterisk so there is some (not very strong) evidence of a difference in 'basal' level of expression? (I'm also unsure of the need for the word 'basal' here, which I didn't understand, couldn't this word simply be removed?). In any case it would be helpful to be clearer about what comparisons have been conducted, perhaps the best way is to provide full statistical results for all comparisons as a supp table (or Figure Source data) including the non-significant comparisons.

- We have revised the text accordingly, the term “basal” was removed to avoid confusion, and the sentence in line 243 was updated as follows: “*Conversely, the level of expression of constructs bearing the rs112233623-T allele showed no response to transactivation (Figure 2a).*”

We have revised the manuscript to ensure that the statistical details are more explicitly linked with the textual descriptions. The legend for Figure 2 now reads (from lines 702 onwards): “*The mean \pm*

*s.e.m is shown (n = 7 (vectors); n=3 (vectors + SMAD3), n=3 (vectors+SMAD3+p300), n=4 (vectors + GATA1+FOG) biologically independent experiments). A two-sided Student's t-test was used, with level of significance indicated by asterisks (*P<.05; **P<.01; ***P<.001; ns, not significant). The statistical results for all comparisons are provided in Supplementary Table 2.”*

In addition, the significance bars in Figure 2 have been repositioned and labelled to clearly match the corresponding text descriptions. We have now included a table (Supplementary Table 2), with the complete statistical results for all comparisons, including those that are not significant, as suggested.

- Lines 215, 'almost complete suppression' does not seem supported by Fig 2b, which shows a similar level of expression for GATA1+FOG to the other rows in the graph (if slightly smaller) - certainly not zero expression. For example is this difference statistically significant?

- We agree and have modified the text of lines 245 to read “*By contrast, the same CCND3 luciferase constructs expressed in HUDEP-2 cells but co-transfected with GATA1 in combination with its coactivator FOG1²⁹ showed an even greater reduction in luciferase activity of the CCND3 enhancer bearing the rs112233623-T derived allele.*” As above, we have included a table (Supplementary Table 2), with the complete statistical results for all comparisons.

- Line 288 and 764, as above please state more clearly the parasite line being used (I think it might be Uganda Palo Alto, FUP/H?).

- We thank the reviewer for this comment. To clarify, the parasite line used in our experiments is the Uganda Palo Alto strain (FUP). We have updated the manuscript from lines 321 onwards to more clearly state the full name of the parasite strain used in our study. “*To assess genotypic effects on malaria resistance, RBCs purified from recalled SardiNIA donors were infected with the Uganda Palo Alto strain of Plasmodium falciparum (FUP) and maintained in culture for three blood-stage cycles as described in Methods.*”

- Line 788-789, state how were the three observer's measurements combined?

- We appreciate the reviewer's inquiry regarding the combination of measurements. As stated in the Methods section (from lines 966 onwards) "*Two thin smears per condition were blindly counted three independent times by three operators according to the latest guidelines of WHO (Malaria Microscopy Quality Assurance Manual).*"

- *Fig 4 legend, definition of Maturation (at 120hpi) seems at odds with Methods (defines at 4, 96 and 144 hpi). (If the cycle is 48 hours then measuring at 144hpi seems appropriate.)*

-We apologise for the oversight regarding Figure 4. The previous version contained an error in the figure legend, which has now been corrected. In response to this issue, we have not only revised the legend but also updated Figure 4 by incorporating additional assays to further enhance the representation of the obtained data. We appreciate the reviewer's feedback, which has allowed us to improve the accuracy and clarity of our manuscript.

- *I enjoyed reading the popgen material, and agree the paper presents compelling evidence of recent +ve selection of this allele/haplotype in Sardinia, and uses best-practice methods for this analysis.*

Referee #3

This paper describes an analysis of a largely Sardinian-specific variant, rs112233623 which has previously been reported to be associated with a variety of red blood cell traits including HbA2, erythrocyte count and various other traits. The authors make several linked claims; specifically:

- *Functional work showing that this variant regulates CCND3 expression, affecting erythropoiesis via binding of SMAD3/GATA1.*

- *Via the effect of decreased CCND3 expression on ROS, the variant confers resistance to malaria.*

- *The variant experienced natural selection due to this protective effect, explaining why it is common in Sardinia but rare elsewhere.*

I would say that the main contributions are to show (in vitro) that rs112233623 protects against malaria, and plausible evidence that it was under recent selection. Overall, I find this paper of moderate interest – it's interesting to see another adaptation to malaria, but it's not particularly surprising and the claims about the potential for improved treatment seem somewhat speculative.

We thank this referee for taking the time to review our paper and for providing useful comments and suggestions. However, we respectfully disagree that the work is of moderate interest and we reprise features below that promote an upgrade of his/her assessment.

Indeed, a critical asset of the work is the coherence and interrelationship of various lines of evidence, which taken together provide what we suggest is a consequential and interesting scientific story based on years of complex experimentation and analyses. In fact, to our knowledge, there are not many examples of studies that have gone from an agnostic, hypothesis-generating GWAS through a series of analyses and functional tests to a full and detailed understanding of the molecular mechanisms underlying the observed association, identifying the molecules that act along the chain of events leading to phenotypic, evolutionary, and medical consequences, including resistance to a major human pathogen such as malaria.

Moreover, we emphasise that functional follow-up and establishing the mechanistic and possible medical implications of the initial genetic association is precisely the focus of the paper. And while the reviewer says it's interesting to see another adaptation to malaria but not particularly surprising, we suggest that this level of evidence, indicating a possible fundamental convergence of at least two mechanisms of antimalarial protection by a single variant, is unique and not just 'another' example of malaria adaptation. And we do believe that what is surprising, and unusual is the robustness and convergence of the *in vitro* evidence for malaria adaptation as a mechanism. Of course, the establishment of a new therapeutic regimen mimicking the "experiment of nature" highlighted by our findings would require considerable development and testing of candidate therapeutics, and is therefore - as the reviewer suggests - "somewhat speculative"; but the work and the nature of the target raise the possibility that we are in the realm of things that could happen here, and would certainly be welcomed for a disease that is still the second leading cause of child mortality worldwide.

Finally, we believe that there are also interesting results within the domain of statistical genetics, as we discuss in more detail below in response to the specific comments of the referee. In particular, the example of a reversal of the effects of one allelic variant due to another allelic variant 160 bp apart for which we provide convergent genetic and functional evidence and elucidate the underlying

mechanisms. Indeed, this type of phenomenon is obviously difficult to detect, with most known examples coming from analyses in mouse models, where specific variants can be more easily assessed in fixed backgrounds. The more general implication for human geneticists supported by our findings, is that this phenomenon may explain apparently inconsistent association results in different populations.

Below are our detailed responses to the Reviewer's specific comments.

I find it reasonably convincing that rs112233623 is a causal variant for the hematological phenotypes, although that is not novel.

I can't really comment much on the details of the functional work, but I hope one of the other reviewers will. I am a bit concerned that the donor samples used for some of the experiments are not matched or controlled for sex, age etc...

To me the experiments in Figure 4 seem to show reasonably convincingly that rs112233623 is likely to confer some resistance to malaria, perhaps of a similar magnitude to G6PD deficiency. The evidence for selection is not exceptionally strong but I think probably sufficient for the claims made in the paper given a high prior that a variant protective against malaria would be selected (assuming the previous parts are convincing).

Major comments.

1) Even though rs112233623 is rare outside Sardinia, the larger sample sizes in other studies mean that you still have much more power to investigate it in other samples. For example, it's ~10 times more common in Sardinia than the rest of Europe, but the big GWAS are more like ~100x the sample size so they have much more power; For example, even though it's only ~0.9% frequency, it has a P-value of $1e-195$ for RBC in Vuckovic et al 2020 (PMID 32888494), which is much more significant than reported in Table 1. Presumably mapping in those cohorts (e.g. UK Biobank), you would be better powered to estimate effect sizes, finemap and detect conditional effects. As the authors point out (lines 76-78), this has in fact already been done and already identified rs112233623 and rs9349205 as independent signals, so I'm not sure the association results add much to that.

We agree with the reviewer that many of the association results presented here were previously reported by others and by us. However, we believe that, given the large difference in frequency between Sardinians and Europeans, taking advantage of the genetic architecture present in the

island is the key to ease the understanding of the selection analyses, the connection to malaria and the availability of a large number of PLoF homozygotes to perform functional experiments as well as to understand the complex genetic and biological relationships between the PLoF variant and a nearby GoF variant. And there are some traits, such as levels of different haemoglobin subtypes, with therapeutic implications -- see Casgevy, the approved therapy for beta-thalassaemia and sickle cell disease that targets BCL11A based on genetic evidence initially discovered in our cohort (PMID: 18245381) -- which have not been included in large biobank efforts and for which this study provides some new interesting results. In particular after conditioning on the PLoF, we observed a null association between the GoF of *CCND3* and HbA2 unlike other haematological traits (i.e. RBC, MCV, MCH). This in turn may suggest that PLoF highlights a specific window during terminal erythropoiesis in which cell cycle shortening increases HbA2 (and to a much lesser extent HbF), whereas cell cycle prolonging GoF has no effect on these haemoglobins.

We emphasise that also the detection of signals of positive selection that led to the hypothesis of a role for malaria as a plausible selective pressure would not have been possible in other European populations, especially those from northern countries, because of the absence of malaria, or in Africa and Asia because of the absence of the variant of interest.

And given the strong emphasis on functional follow-up, also experiments to test the role of malaria as a possible selective pressure in Sardinia are truly impractical in any other population, even in ultra-large biobanking efforts. In fact, most of the largest biobank collections are not even bioresources where individuals can be recalled on the basis of their genetic profile and resampled to perform functional tests on the collected biological samples. And even if they were, in order to call and draw blood from individuals in a general Northern European population to obtain, say, 20 individuals homozygous for rs112233623, without considering the need to further match for a large number other variables and avoid confounders of different ethnic backgrounds, it would be necessary to start with 200,000 genetically profiled individuals, typically spread over large areas; confirm that they are willing to give blood; and then be sure that the samples arrive at a central laboratory in controlled conditions without batch effects. Such a course of action would be impractical, and makes the experiments reported here, which also yielded unambiguous results, unique.

To be clear, we have also been among the first cohorts to participate enthusiastically in genetic studies based on very large sample sizes and we agree that very large cohorts can generally provide better estimates but for the specific variants studied here, our cohort offers an easier way to a complete comprehension of biological mechanisms.

2a) The effect on expression in Figure 1 is pretty weak – the P-value is marginal and there is one large outlier for the CC genotype. A larger sample would help, alternatively is there a reason not to look in the whole-blood RNA seq data from this cohort from Pala et al (2017)? According to GTEx, CCND3 is reasonably highly expressed in whole blood. The browser for the results of that paper appears to be no longer available, so I can't check but since some of the authors overlap I assume they have access to the data.

We thank the reviewer for this suggestion.

Regarding *CCND3* expression in whole blood, while GTEx reports that *CCND3* is expressed in this tissue, our study specifically focuses on erythroid precursors, as they represent the most relevant cellular context for assessing the functional impact of rs112233623-T on *CCND3* expression and its downstream effects. Whole blood consists of a heterogeneous mix of cell types, like B cells, with varying *CCND3* expression levels, making it less suitable for detecting cell-type-specific regulatory effects. In contrast, erythroid progenitors that are normally absent in the whole blood, provide a more controlled system for investigating *CCND3*'s role in erythropoiesis. Moreover, GTEx does not report eQTLs for *CCND3* in whole blood on its online portal, making it impossible to verify such an association.

Regarding the RNA-seq data from Pala et al. (2017), the significant eQTL results remain publicly available on the download page (<https://eqtlstownload.irgb.cnr.it>). To improve accessibility, we have also made link to an Excel version of the significant results and a browser available. This ensures continued access to the data while the website managed by our colleagues at Stanford is undergoing maintenance. Additionally, we recently added the full summary statistics, allowing users to examine associations for all tested genetic variants, not just the significant ones.

However, it is important to note that this dataset was generated from leukocytes, not whole blood, meaning that erythroid cells are not characterised and cannot be used as a reference for studying regulatory mechanisms specific to the erythroid lineage. Despite this, *CCND3* (ENSG00000112576.8) was tested for eQTL mapping in Pala et al. (2017), and an eQTL was identified: rs4623235 (chr6:41924853, $P=5.9 \times 10^{-7}$). However, rs112233623 (chr6:41924998) was not an eQTL for *CCND3* in this dataset ($P=2.1 \times 10^{-2}$, and $r^2 = 0.01$ with the eQTL rs4623235).

We hope this clarifies our rationale for focusing on erythroblasts rather than whole blood or leukocytes and addresses your concerns. And despite the difficulties in assessing erythroblasts to increase the sample size, we attempted, with considerable logistical effort, to increase the sample size

for erythroid cultures by recruiting additional individuals homozygous for either the wild-type rs112233623-C allele or the PLoF rs112233623-T allele. This allowed us to increase the number of samples per genotype from 4 to 11, thereby increasing the robustness of our results. We also preferred to compare expression levels between genotypes using a non-parametric test (Wilcoxon-Mann-Whitney test) rather than parametric statistical tests, to limit the effect of outliers on the means.

3) In the analyses based on patient-derived RBC or erythrocyte progenitors (e.g. Figure 1 and 4), are you controlling for sex, age ABO and other technical factors in the analysis? I appreciate you controlled for globin deletions and GDP6, but there are many other things associated with malaria response? In addition, it is a shame you did not look at heterozygotes – it would be relevant for the evolutionary questions whether the effects are additive or not (in many other malaria resistance alleles the protective effect is additive, while the negative consequences are recessive, explaining why they experience balancing selection).

As the reviewer appreciated, our work was based on an "a priori" selection strategy for individuals to be recalled and resampled for functional testing based on extreme genotypes at *CCND3* (see our response below), and fixed multiple negative genotypes for variation at the other major loci [HBB (rs11549407-A), HBA (rs141494605-C), and G6PD (rs5030868-A)] with allelic frequencies of 0.051, 0.15 and 0.083, respectively] previously implicated in malaria resistance and common in Sardinia.

As a significant proportion of the Sardinians are carriers of one or more of the 3 variants described above, the number of individuals recalled for functional testing on *CCND3* genotypes was reduced to a total of 43 individuals (among those still alive and willing to be resampled from the original cohort of ~8,000 participants), further subgroup analyses were somewhat limited.

With regard to age and sex, the effect of age is weak, whereas the effect of sex was significant for the haematological traits assessed (and of course was controlled for as a covariate in the association analyses) and controlled for in the functional tests (M:F=5:9 for GoF and 5:12 for PLoF) as now reported in Methods section, lines 806-807.

As for ABO blood groups, they were not defined in our cohort. However, given that a large meta-analysis (PMID: 30029997) confirmed some increase in the risk of severe *P. falciparum* infection among individuals with non-O blood groups compared with O, but reported no significant difference in the level of *P. falciparum* parasitaemia among the same groups, and also considering their random

distribution across the different genotype groups used in our experiments to assess parasitaemia, we are highly confident that ABO blood groups did not have an effect in our experiments.

It would indeed be interesting to look at heterozygotes and we think the point is well made. We considered doing this from the outset, but it would have further complicated both the logistical part of the study - in terms of recalling volunteers based on their genotype and matching them for the various other traits involved in malaria resistance - and the design and experimental conditions of the functional part, as the larger and more heterogeneous group of individuals to be assessed would have increased the risk of inaccuracies, batch effects and other sources of bias. We therefore decided to focus on samples from individuals with extreme genotypes, accepting that we cannot answer all the open questions in this paper and planning to perform some analyses such as those proposed here and others in a follow-up study to complement and refine those performed in this study.

4) Selection analysis – I think the approach is generally reasonable. The analyses seem to largely converge on a P-value somewhere in the 99th percentile. It's not particularly significant, but I think given that the authors are only testing a single variant and there's a fairly strong prior that if it's protective against malaria, it would be under selection, I think that's fine. I think the genomic null procedure makes sense. One comment is that personally I find the CLUES analysis the most convincing, so I'm surprised that they only ran this with such a small sample size. They have thousands of genomes available. This approach can certainly scale to more than 100 individuals. Similarly, since you have a large sample of present-day people you could impute rs112233623 in the ancient Sardinian data and incorporate that into the analysis (e.g. with CLUES2) – that would give you more confidence about the strength and timing of selection (is it consistent with the historical introduction of Malaria?). That would all strengthen the manuscript, that said, I don't think any of this is strictly necessary to support the claim as it is that it's under selection which I think is ok.

We appreciate the reviewer's overall assessment of the selection evidence. To address the questions raised we ran CLUES on the samples of size 100 in part to have a comparable size to the YRI and TSI samples from the 1000 Genomes. Also, the ARG inference using RELATE that CLUES depends on does become computational costly with samples sizes in the 1000s (we attempted it but it did not complete in time with our compute cluster's standard 4-day limit). Regarding imputation of the aDNA, we fully agree that assessing evidence for positive selection from an ancient DNA time series imputing rs112233623 (or even redesigning the capture assay to include rs112233623 and many other variants not present in the currently used assay) would certainly be attractive as an additional complementary line of evidence. However, this is currently precluded by the fact that the existing

sample sizes across archaeological time for Sardinia are small (n~100 over all archaeological periods combined) and unfortunately provide little power to distinguish selection from neutrality in shifts in allele frequency. Moreover, the available samples poorly cover the last 3000 years; a period at the beginning of which *P. falciparum* was plausibly introduced to the island.

5) Data availability: The raw data are not publicly available (or at least it's not clear if they are). The summary statistics from the original 2015 association study are not publicly available. The summary statistics from this study are not provided. The raw data from the experiments performed in this study are not provided. The code used to analyze the data is not provided. Consequently, the statement that "Data are provided as Source Data or Supplementary Information." Is false. At minimum, the authors need to make publicly available all the association summary statistics, the raw data from the experiments, and the code used to analyze those data.

We apologise for this misunderstanding. We meant to indicate that the raw data will be made available when the paper is published, although it was our fault to not provide preliminarily the data to the reviewers and include them in the data availability section, as we are now doing.

We acknowledge the reviewer's comment as a valid and important point regarding data availability.

Data availability from Danjou et al. (2015)

At the time of publication, the raw data from Danjou et al. (PMID: 26366553) were made publicly available in compliance with journal policies at the (European Genome-phenome Archive (EGA) with code EGAD00010001722. While providing raw data was considered sufficient at the time, we recognise that making summary statistics publicly available is now a widely accepted and beneficial practice. To address this, we have made the summary statistics from the Danjou et al. study accessible through our Pheweb Sardinia repository: <http://pheweb.irgb.cnr.it>. This repository not only provides direct access to summary statistics but also includes an interactive browser to visualise the data. Furthermore, we are in the process of preparing these data for submission to the GWAS Catalog to enhance accessibility and visibility.

2. Data availability for the current study

For the raw data from this study, we have set up a dedicated repository to ensure full transparency and reproducibility. The data can be accessed at: <https://figshare.com/s/d89f8f9f90d3a76eccb9?file=48025780>.

This repository includes: 1) raw data and/or videos for each figure in the manuscript; 2) summary statistics of the genetic associations at the *CCND3* locus; 3) the code and formatted input files used in the analyses.

We are committed to ensuring full data accessibility and transparency. Our participation in collaborative studies and meta-analyses has consistently led to public data sharing. We believe that these steps will further improve the accessibility and usability of our data for the broader research community.

Minor comments.

6) In the mixed model association analysis (line 810), do you use the genome-wide GRM or the local GRM to control for stratification. It's not actually clear to me which is more appropriate. Usually one would use the genome-wide GRM, but if you think the region is under strong selection one might argue it's more appropriate to use the local one.

This is a good point. We usually use the genome-wide GRM (obtained in our specific case considering about 100k variants distributed along the genome, with $MAF > 0.01$ and $r^2 < 0.1$, excluding the HLA region due to its peculiar haplotype structure). In general, we prefer not to use a local GRM to avoid a loss of statistical power due to potential proximal contamination in including the candidate variant in the model as a fixed effect (in the association) and as a random effect (in the GRM), as also reported in several papers (PMID: 22669648; PMID: 24473328; PMID: 34017140).

Based on the reviewer's comment, we tested this hypothesis in our data and repeated the association analyses using a local GRM matrix. We considered a region of ± 1 Mb and a region of ± 5 Mb around rs112233623 and included in the estimation 3,091 and 21,018 variants, respectively, selected under the same frequency and LD conditions as in the genome-wide GRM estimation. The results presented in the table below show a general coherence of effects among the models but, under the same conditions of sample size, normalisation and covariates, an increase in the standard error when using the local GRM due to the higher noise introduced into the model by the local kinship estimation.

Trait	N samples	Geno count (CC/CT/TT)	MAF	GRM genome-wide			local GRM (+5Mb)			local GRM (+1Mb)		
				Effect (beta)	SE	P value	Effect (beta)	SE	P value	Effect (beta)	SE	P value
HbA2 (g/dl)	6762	5480/1219/63	0.099	0.306	0.032	4.03E-22	0.292	0.047	3.87E-10	0.264	0.053	6.87E-07
MCV	6824	5526/1234/64	0.100	0.356	0.032	3.77E-29	0.349	0.053	6.19E-11	0.338	0.064	1.15E-07
MCH	6824	5526/1234/64	0.100	0.319	0.031	2.48E-24	0.321	0.053	1.79E-09	0.321	0.067	1.57E-06
RBC	6819	5521/1234/64	0.100	-0.267	0.032	1.18E-16	-0.318	0.047	9.53E-12	-0.324	0.053	1.16E-09
HbF (g/dl)	6710	5436/1211/63	0.099	0.091	0.032	4.32E-03	0.091	0.040	0.02151	0.068	0.041	0.1009

The table reports the comparison of association results reported in Table 1 with those obtained using different local GRM matrices.

7) I find the LoF/GoF terminology a bit confusing. The authors mean expression decreasing or increasing variants – I tend to think of LoF variants as coding, and I feel like that's the standard in the field, but I could be wrong.

We are using the terminology in the usual sense associated with classical genetics where GoF and LoF are defined with respect to the gene's activity – regardless of whether the mutation is coding or non-coding molecularly. To help slightly we have modified the terms and now use 'partial loss of function' (PLoF) to reflect the expression of the protein is reduced but not completely abrogated in homozygous individuals.

8) Line 347: This is not an example of epistatic dominance. If I understand correctly, you are claiming that the effects are additive.

We are grateful to this reviewer for this comment, which has allowed us to better define the findings referred to in this point. This is indeed an example of a type of masking/overriding effect where the marginal additive effect of one variant, rs9349205-A, differs by genetic background due to different linkage disequilibrium with a second causal variant, rs112233623-T, which acts in the opposite direction. In that statistical sense – where the marginal effect varies by background – we believe this is epistasis. Mechanistically, we also see a plausible interaction as the effect of rs9349205-A on regulation via affecting chromatin conformation is plausibly irrelevant in the presence of the rs112233623-T, given rs112233623-T removes the SMAD3 binding site. That said, and although we were able to functionally and directly demonstrate that rs112233623-T reverses the effect of rs9349205-A on *CCND3* expression when they are cloned in the same DNA fragment in erythroid cell line co-transfection experiments, we acknowledge that there are difficulties in performing a well-powered statistical test for epistatic interactions at the population level because the LD patterns in the

Sardinian cohort and the particular configuration of genotypes available globally limit this type of analysis.

We now report and discuss the results more directly without making specific claims about a specific type of epistasis, such that readers may draw their own conclusions on the matter.

9) Supplementary information “negative selection of the rs9349205-A” – I don’t think this is very convincing, since there’s no actual analysis here. It’s also not negative or purifying selection (i.e. selection against new deleterious mutations). The variant is already segregating - it’s just positive selection for the other allele. The evidence in Figure 4 that rs9349205 affects invasion etc.. is statistically very weak (e.g. it would not be significant if you corrected for the number of tests in that experiment).

We agree and have removed the claims of negative selection for rs9349205-A and significantly reduced the emphasis on it.

10) Several small but annoying issues with statistical analysis and data presentation:

- All the boxplots should show the individual points instead (there are not so many); e.g. figure 1a and c, figure 4. Box plots are particularly misleading in Figure 4 because the number of observations is so different.

As requested, individual points will now be shown in all the boxplots.

- Please add confidence intervals to all plots (e.g. Figure 3, Extended data figure 2)

As suggested, we added the confidence interval to frequencies in Figure 3, while we preferred to keep the haplotype frequencies figure (Extended data figure 2) unchanged, to avoid it becoming too confusing to read.

- Please give a confidence interval for the selection coefficient (and give the P-value in the main text at line 264; the likelihood ratio is not easily interpretable to the non-statistical reader.

Unfortunately, the tool we used to calculate the selection coefficient (CLUES) only provides a point estimate. As suggested, we have removed the likelihood ratio value from the main text (but this is reported in the supplementary material). Also, in this type of analysis, the P-value is expressed as a genomic percentile.

- Please give exact P-values in e.g. Figure 4

We have now included a table (Supplementary Table 2), with the complete statistical results for all comparisons, including those that are not significant, as suggested.

- Extended data figure 2: why are there two panels? But Sardinia is in both?

It has now been better specified in the text legend Extended Data Figure 2: “*The haplotype frequencies of rs9349205-rs112233623 haplotypes in 1000 genomes superpopulations (a) and in European populations (b)*”

11) Typos etc...

- Figure 3; “Kenia”, “Eest Africa”, etc... “frequency level”

- “Tuscans in Italy” in Figure 3 but “Tuscany in Italy” in Extended data figure 2

- Line 131: “has no any primary effect”

- Line 813: “sex, age²” – assume you mean, “sex, age AND age²”

We thank the Referee and accordingly we have corrected the text.

Referee #4

This interesting manuscript by Marini et al addresses an important question in the malaria field, namely the relationship between human genetic variation and malaria susceptibility. As the authors note, understanding how malaria has selected for human genetic variation has the potential to lead to the discovery of novel therapeutic targets. In this work a variant in an enhancer for the cell cycle gene CCND3 is investigated. The variant has a high allele frequency in Sardinians (as compared to Europeans) suggesting it may have resulted from positive selection. Prior studies have linked a different variant in the CCND3 enhancer with hematologic traits, and the variant of

question in this work has also been previously implicated to have effects on RBC counts in data from the UK Biobank.

Strengths of the paper are 1) the demonstration of an association between the rs112233623 allele and hematologic traits in Sardinians (ext data figure 1), the characterization of CCND3 expression and cell cycle parameters in erythroblasts derived from individuals with the TT vs WT background, and 3) the demonstration that the T allele leads to reduced enhancer activity in the luciferase promoter assays, as well as the involvement of the transcription factors of SMAD3 and GATA1, providing a potential mechanism for the reduced CCND3 expression in the TT allele.

Additionally, the authors analyze haplotype data and allele frequencies and come to the conclusion that there is evidence for the TT allele being under positive selection. They hypothesize that this may be due to malaria, and attempt to demonstrate this using Plasmodium invasion and growth assays.

In response to the insightful queries raised by this reviewer, we have thoroughly re-examined and meticulously recounted all smears and parasitaemia assays to ensure accuracy and relevance. Furthermore, we have reviewed and reprocessed critical data to make it readily and widely available

Major points:

In Figure 1b, cultures of erythroid progenitor cells are characterized for their cell cycle stages. While a potentially interesting finding is reported where the cell cycle stages differ between the genetic backgrounds only at the polychromatic erythroblast stage and not at the earlier basophilic erythroblast stage, this conclusion is not firm as it appears that the cells were not actually staged, but rather assumed to be at a certain stage based on their day of differentiation. To be more definitive about about potential for different developmental/cell cycle phenotypes in the genetic backgrounds, should perform a more comprehensive flow cytometry-based analysis for erythrocyte stage (as in Hu et al Blood, 2013), and also provide growth curves for the erythroblasts from the two different backgrounds. If indeed there is a cell cycle defect in the TT background, this should be apparent in these standard assays.

While there are some additional data presented in Ext data figure 3, only one genetic background is shown (presumably WT) so the relevance to this issue is not clear.

Based on the reviewer's recommendation, we have strengthened our methodology by establishing new erythroid cultures from erythroblasts derived from individuals homozygous for either the wild-type rs112233623-C allele or the PLoF rs112233623-T allele. To analyse their maturation stages at

specific time points (day 7 and day 9 of erythroid culture), we implemented a more comprehensive flow cytometry-based approach for erythrocyte staging. This analysis utilises a combination of GPA, Band3, and $\alpha 4$ -integrin markers, allowing for a more precise characterization of erythroblast populations. Additionally, we compared these results with cytopsin data from the same erythroid cultures for further validation.

These updated findings are now presented in Figure 1 and Extended Data Figures 3 and 4.

To enhance clarity, we have updated the methodology in the Methods section (from lines 816 onwards): "*For the single-cell population, CD235a-PC7 versus FSC-A was used to gate on CD235a-PC7 positive cells. Erythroblast maturation was evaluated by flow cytometry through the simultaneous detection of surface markers $\alpha 4$ -Integrin, Glycophorin A (GPA), and Band3, using anti-CD49d-APC, anti-CD235a-PC7 and anti-Band3-PE monoclonal antibodies, respectively.*"

Furthermore, to assess potential differences in cell cycle progression, we included a growth curve analysis comparing erythroblasts from both genotypes. This is now presented in Extended Data Figure 4b and described in the manuscript at line 189: "*These results were confirmed by the differential growth curves observed for the two genotypes (Extended Data Figure 4b)*".

One of the major conclusions of the paper is the report of a causal role of the genetic variant rs112233623-T on variation in hematological traits as well as protection against malaria. While some of the results presented in Figure 4 appear to support the conclusion about malaria, the methods as described and the data presented raise concern that the conclusion may not be adequately supported by evidence. Specifically, the authors report that the parasite assays were initiated at 5% schizont-stage parasitemia in 2% hematocrit. In this setting, the schizonts would be expected to egress from the mother cell, and re-invade at a rate $\sim 5x$, which would lead to $\sim 25\%$ ring parasitemia at 24 hours. This is an extremely high parasitemia, which would most certainly crash within that cycle (by 48 hours).

We thank again the reviewer for the careful and critical examination of our work giving us the possibility to amend/clarify relevant points in the revised manuscript. On this specific point, the 5% schizont-stage parasitaemia was a refuse and indeed all the experiments were performed with an initial 2.5% schizont-stage parasitaemia in 2% haematocrit, as can be verified by all data and images

we are sharing (see below). We acknowledge the error in reporting this information and have now corrected it in the manuscript. (from lines 960 onwards: “*To start plasmodium cultures, density isolated schizonts were added to erythrocytes at 2% haematocrit in growth medium to bring the density of parasite infected erythrocytes at 2.5%.*”).

The pictures in Figure 4 d-e are consistent with crashed parasites, so the interpretation that the parasites in these pictures are “delayed” is debatable. Moreover, there is a discrepancy between the pictures and the graphs (panels 4f-g). The images in Figure 4 reportedly show parasitemia at 72 hours and 120 hours, and the graphs show parasitemia at 120 hours. The pictures at 120 hours show parasitemia in the range of 25->50%, whereas the graphs report parasitemia of 1-4%. Why are these values so different?

In light of this reviewer's comments, and after a massive and careful re-evaluation of all the results, the old Figure 4 has been removed and replaced with a new Figure 4 along with an Extended Data Figure 9 that together addresses all the concerns, showing the results for all time points and making them clear. In addition, the results will be available as source data in the appropriate files, as well as images (photomicrographs of blood smears) and associated videos at <https://figshare.com/s/d89f8f9f90d3a76eccb9>).

To focus on the questions related to the old Figure 4, the apparent inconsistency between the parasitaemia inferred from the images in Figure 4 panels b-e) (showing high parasitaemia at 120 hours) and the numerical parasitaemia values (1-4%) reported in Figure 4 panels f-g) was due to differences in the presentation of the data, with the images intended to highlight morphological changes in the parasites magnifying representative fields, rather than being representative of the actual quantification of parasitaemia; which was admittedly misleading.

Furthermore, the authors report that they counted parasitemia several times during the assay (24 hr, 72hr, 96 hr, 120 hr, and 144hr) yet only a single timepoint is reported (120 hr), raising additional concern that there is a lack of transparency in data reporting. Given these significant issues, I am not able to evaluate the veracity of these results and any conclusions to be made from them. While the idea to try to correlate the ROS status of the cells with parasite growth phenotypes is reasonable, with the concern about the parasite growth phenotypes not being accurate, the significance of these potential correlations is not clear.

As anticipated above, in response to this concern, we have now included all time points in the revised Figure 4 and in the Extended Data Figure 9 to ensure data transparency and reproducibility of the observations - which is indeed a primary goal also for us - we made available the related raw data in the appropriate files, images and associated videos.

At a minimum, the methods should be clarified, data from all timepoints should be reported, whether or not the smears were counted blindly should be reported, and how the data from the three observers was treated should be reported. Additionally, it does not seem feasible that all of these assays were done in a single batch, so the authors should describe how the batches were treated. If indeed these experiments were conducted using 5% initial parasitemia and then incubated for 120 hours, then to avoid crashing the experiments would need to be repeated using a lower starting parasitemia and shorter assay (presumably one round of invasion/ maturation should demonstrate any phenotype).

Again we fully agree and in addition to data on all timepoints mentioned above we have now reported all the requested information as Source Data.

Likewise, as we report above in response to reviewer 2 we confirm that all parasitaemia measurements were performed in a blinded manner by three independent operators following the WHO Malaria Microscopy Quality Assurance Manual guidelines, and we have added from lines 966 onwards the following statement in the Methods section: *“Two thin smears per condition were blindly counted three independent times by three operators according to the latest guidelines of WHO (Malaria Microscopy Quality Assurance Manual).”*

The specific query on whether assays were done in a single or multiple batches and how these were treated is also relevant and we have now better clarified this from lines 985 onwards of the revised manuscript. *“The experiments were conducted in 5 batches of homogeneous size, ensuring that individuals recruited for each experimental batch had a similar proportion of the different genotypes assessed to avoid 'batch effects'. To this end, we also ensured that all assays were performed under strictly controlled, standardised and homogeneous conditions to maintain consistency and comparability between batches, using exactly the same experimental protocols, including identical reagent batches, instrument settings and environmental parameters. This approach ensured the validity, accuracy and reproducibility of the results and minimised the variability associated with sample processing over time.”*

Regarding the experimental conditions, as noted earlier, all experimental infections were conducted at 2.5% initial parasitaemia, and to prevent excessive parasitaemia, we diluted cultures by 50% every 48 hours by adding RBCs. This step is crucial to maintaining optimal parasite growth and preventing nutrient depletion. We now emphasise this point in the revised Methods section (from lines 960 onwards): "*To start plasmodium cultures, density isolated schizonts were added to erythrocytes at 2% haematocrit in growth medium to bring the density of parasite infected erythrocytes at 2.5%. RBC infection is expected to be completed after 10–14 h. Every 48 hours from the start of Plasmodium cultures, an equivalent volume of RBCs was added to the cultures diluting parasites by 50%. Diluting parasites has the advantage of avoiding an increase in parasitaemia that may delay parasite growth due to nutrients depletion.*"

The error bars in Figure 1a are not defined in the legend. While it is mentioned that N=4, only one data point is visible for each genetic background, so it is not possible to fully visualize the spread of the data. It would be helpful to plot all data points for the other graphs as well (e.g. Figure 4 f-h).

Now it has been better specified in legend Figure 1a: "*The box-and-whiskers plot indicates the CCND3 median expression (the line across the box); the box ranges from the first quartile to the third quartile of the distribution; the "whiskers" extend from each box to the most extreme data points.*" All data points have now been added to the plots as suggested.

In addition to the association profiles shown in Ext Data Figure 1, the conclusions regarding impact of the rs112233623-T allele on hematologic parameters would be strengthened if complete blood counts and RBC indices for blood donors from the two genetic backgrounds were provided, especially since many such donors were used for the parasite assays.

We have now added the Supplementary Table 2, where we have reported the summary statistics (range, 1st and 3rd quartile, mean, median and standard error) for rs112233623-CC and rs112233623-TT carriers in the entire sample set for all the traits listed in Table 1.

While the analysis of allele frequency differentiation and haplotype variation as evidence for

positive selection is interesting, this is outside the scope of my expertise and should be assessed by a reviewer with specific expertise in this area.

We hope that the revised manuscript with the new data, figures and the effort to make the underlying data and images fully available to everyone will give a satisfactory response and allay the legitimate concerns of the reviewers that we thank for their valuable contribution to improve the manuscript.

Referees' comments and responses:

Referee #2 (Remarks to the Author):

Thank you for addressing these points. I particularly appreciate the careful re-wording of the title / abstract and discussion relating to what is shown re: a malaria protective effect and also in relation to G6PD; and the addition of in vitro experiments in a wider range of Pf lines. These have addressed most of my concerns.

I have a couple of minor remaining points where I would still seek clarification:

- In response 2.1, the authors say "We utilised the FUP Uganda Palo Alto strain" and that a section clarifying this has been added to Methods (presumably lines 938-945). However, as far as I could tell the revised ms and this section still don't state which version is being used. (BEI Resources / ATCC supply three such strains, which go by Uganda Palo Alto Hawaii, Cayenne and Roche respectively; the reference cited in this question indicates that different versions of FUP do have different genotypes and are probably distinct isolates.) This could be important, for example, for reproducibility. Therefore, I don't think this question has been addressed. Can the authors specify which version of FUP was used in their experiments? (If this does not correspond to a known identifier such as those from BEI, providing information on the provenance of the isolate used in the lab be given? If this information is not known, that's not necessarily a showstopper particularly given the additional data on other lines that has been provided in the revised manuscript, but I think should be clearly stated.

The FUP (*Falciparum Uganda Palo Alto*) strain employed in our experiments does not correspond to one of the commercially available BEI Resources/ATCC versions (e.g., Uganda Palo Alto Hawaii, Cayenne, or Roche). Instead, the isolate was originally donated to our laboratory by Prof. Evelin Schwarzer (University of Torino, Italy), with whom we have a long-standing scientific

collaboration. This particular FUP strain has been maintained in continuous *in vitro* culture in our laboratory since 2011. It has been utilized in numerous reports, some of them cited in the ms. We have now explicitly clarified this point in the Methods section (Line 934) of the revised manuscript.

- Re: the question of how the blood smear measurements from different operators were combined, I don't think this has been addressed either. (The question is: given there are six blood smears in total counted by three operators independently, state how these counts were combined to form one overall count for the sample.) This may well be in the cited WHO manual but should be stated here anyway for clarity (I think another reviewer has asked a similar question.)

As noticed, we did not explicitly describe how the resulting counts were combined. In the revised Methods section (Lines 980–983) we have added the following sentence to address this point: “*For each condition, two thin blood smears were independently counted by three trained microscopists in a blinded fashion according to the latest guidelines of WHO (Malaria Microscopy Quality Assurance Manual). The resulting six counts were averaged to obtain a single arithmetic mean per condition*”.

- I did also review some of the figure code via the figshare link provided and noted (see comments in separate box) that not all of the code is directly runnable - this should probably be fixed to make this code maximally useful.

We are grateful to the reviewer for carefully checking the scripts and highlighting the issues encountered during code execution. We have now updated the code and the input data to ensure that they run correctly and produce the figures with additional elements as in the manuscript.

Finally, some minor typos I noticed:

Line 315 - sentence fragment seems wrongf, "selected among not carriers of variants"

Line 411 - 'stricking' -> 'striking OK

Fig 4c Throphozoites -> Trophozoites OK

Thanks!

Referee #2 (Remarks on code availability):

R scripts and source data for the main and Extended Data figures have been provided in the above figshare.

I very briefly reviewed the code for some of these figures to verify that it looked sensible and that it produces output matching that in the figures.

Not all of the scripts run completely without errors, for example:

- Figure 4b/script_2024.R attempts to load parassitemia_allinone_june2024.txt, which does not exist in the directory, so that the last plot fails

- Figure 3 - Extended data figure 2/script.R attempts to load files like frequenze_aplotipi.txt and frequenze_aplotipi.EUR.txt which similarly don't exist in the folder.

This should probably be reviewed / corrected so that all scripts are runnable, if this code repository is to be most useful. However, for other figures, in general these tests confirmed that the code produces the output shown in the paper for the figures I tested (primarily Fig 1, and Fig 4). (I did note that, relative to the script output, the figures in the paper have had some additional annotations and other cosmetic changes added, including addition of lines/asterisks denoting P-values, rotation of axis labels, and so on.)

Referee #3 (Remarks to the Author):

I appreciate the work that the authors have done on the revision; addressing many of the points that the other reviewers and I raised in the first round.

From my perspective as a human population geneticist, I find the evidence for selection on rs112233623 specifically to be moderately compelling, but not at the level one would require in, say, a genome-wide scan. If this were just a random variant I would not be convinced that it had been under selection (a test like SDS that is more sensitive to very recent selection might strengthen that). However, if the variant were indeed protective in vivo against malaria, then my prior that it would be under selection would be quite high, and then I would find the claim of selection to be overall convincing.

On the other hand, reading the other reviewers' comments, it seems that they find the evidence for in vivo protection moderately compelling, but think that the population genetic evidence strengthens that claim. And, the paper motivates the investigation of malaria based on the evidence of selection. So it seems like there's a bit of circularity in that some reviewers think that the evidence for selection supports the protective effect, but I think the protective effect supports selection. As one of the other reviewers points out there are "many variants that have been proposed as malaria-protective yet don't seem to actually be associated with protection".

On balance I think the two lines of argument complement each other and I think I am convinced by the paper as a whole. But I can't really evaluate whether the in vitro evidence for protection against malaria is convincing that there would be a substantial in vivo effect. If it's not, I would probably feel differently.

Other comments:

1) I'm still a bit uncertain about the eQTL evidence in figure 1a. The P-value is 1.0e-2, which doesn't seem that different than the 2.1e-2 they report in Pala et al. which they say is "not an eQTL". In contrast, they say that rs4623235 is an eQTL in Pala et al. but then in Figure 1a, they don't control for rs4623235 genotype (or rs9349205, for that matter).

I appreciate that it is a different cell type, but I don't think those eQTL are likely to be entirely cell-type specific. If rs4623235 is really an eQTL, I would think that it would have an effect on expression here too.

We appreciate the reviewer's comment and the opportunity to clarify this point.

We would first like to note that the nominal p-value reported in Figure 1a ($p = 1.0e-2$) was obtained from a relatively small sample size ($n = 22$), directly comparing expression values between the two genotypes CC and TT, without controlling for other variables. The results in Pala et al. were inferred from a much larger cohort (606 individuals), with expression levels adjusted for latent factors identified by PEER10 (v1.3). This implies that p-values from the two studies should not be directly compared, because of the difference in the statistical models applied and in standard errors on which p-values (and statistical power) depend. That substantially affects the interpretation of nominal significance. Moreover, the analysis in Figure 1a consists of a unique statistical test, while eQTLs from Pala et al. were derived from *in-cis* association analyses within 1 Mb of the TSS of the gene, thus requiring multiple testing correction to the nominal p-value to declare significance.

A formal analysis supports the conclusion that there is no evidence for rs112233623 as an eQTL in Pala et al. In that study, the sole identified *cis*-eQTL for *CCND3* was rs4623235, the top associated variant in the region (and meeting the 5% FDR threshold). To assess the presence of additional independent *cis*-eQTLs, we performed a stepwise forward regression: rs4623235 was

regressed out from the expression trait, and the association analysis was repeated. No other variants reached significance at an FDR of 5%. Specifically, the p-value of rs112233623 after conditioning on rs4623235 was $p = 0.8112$, providing no evidence --neither nominal nor suggestive -- for an association with *CCND3* expression in the Pala et al. dataset.

In Figure 1a of Marini et al., analyses were not controlled both for rs4623235 (previously confirmed as an eQTL in bulk leukocytes) and rs9349205, because all individuals (except one, Donor11, see attached Table) in the erythroid cell dataset are isogenic at this locus. Thus, rs4623235 and rs9349205 cannot confound the observed association between rs112233623 and *CCND3* expression.

Sample	Genotype	Expression	Sex	rs4623235
Donor1	CC	1.2591	2	AA
Donor2	CC	0.4894	2	AA
Donor3	CC	0.2080	2	AA
Donor4	CC	0.5444	2	AA
Donor5	CC	0.2269	1	AA
Donor6	CC	0.1249	1	AA
Donor7	CC	0.2123	1	AA
Donor8	CC	0.3456	1	AA
Donor9	CC	0.2346	1	AA
Donor10	CC	0.2161	2	AA
Donor11	CC	0.2433	2	GG
Donor12	TT	0.6766	2	AA
Donor13	TT	0.1640	2	AA
Donor14	TT	0.2435	2	AA
Donor15	TT	0.1860	2	AA
Donor16	TT	0.1362	1	AA
Donor17	TT	0.0977	2	AA
Donor18	TT	0.2075	2	AA
Donor19	TT	0.1913	2	AA
Donor20	TT	0.1547	1	AA
Donor21	TT	0.1449	1	AA
Donor22	TT	0.1617	1	AA

Similarly, later in the response they say that “of course [sex] was controlled for as a covariate in the association analyses”, but I don’t think that sex is controlled in Figure 1a? Unless they are regressing out the effect of sex and performing the Wilcoxon tests on the residuals?

Regarding the impact of sex on the analyses, we minimized its potential confounding effect by balancing the number of male and female samples (M:F = 5:6 in CC and 4:7 in TT carriers, respectively; see table above). Following the reviewer’s comment and acknowledging that the small sample size could be underpowered with further variable stratification, we applied linear regression to the trait, normalizing it with the inverse normal transformation.

While there is some apparent visual signal (figure below), we find sex does not significantly affect *CCND3* expression levels (p -value = 0.11), which supports the analysis approach to not include it as a covariate in the association model.

*2) The point about the contrasting effects of rs9349205 and rs112233623 is well-made and interesting. However, I maintain this is not an example of epistasis. Epistasis would involve an interaction between the two genotypes. A difference in the marginal effect of a variant due to LD with another causal variant is not epistasis. The claim here is that the effects of rs9349205 and rs112233623 are additive. If you think there is an epistatic effect, you should test for that explicitly (i.e. phenotype ~ rs9349205 * rs112233623).*

I also think that the discussion on lines 140-169 is unnecessarily long. The point can be made much more succinctly.

We understand the reviewer's point, and have now shortened the section on lines 140–160.

3) Minor comment but I don't see why they can't run things for more than 4 days on their cluster. I don't see why they need to include TSI and YRI in their relate analysis since rs112233623 is not present there. Even if you really can't run for more than 4 days, you could run 300 Sardinians instead of 100/100/100 TSI/YRI/Sardinians. That said I don't think it will make a difference to the selection claim.

Following the suggestion we carried out the analyses with 300 random samples from the SardiNIA cohort (of which 23 overlapped with the 100 samples previously tested) and we confirmed the evidence of selection already seen in the 100 samples. In the Supplementary material, we now report the description of these analyses and we state: “*The point estimate of the selection coefficient for the partial-LoF variant in this analysis was $s=0.01133$ ($2\log LR=7.1712$, $p=0.0074$). In this version of the analysis, the $2\log LR$ statistic for the partial-LoF variant is in the 95.4th percentile of all matched variants.*”

4) Line 1685: “The much greater length of the haplotype with the derived allele is obvious, evidencing positive selection.” I don’t think this is actually obvious because rarer variants will naturally have longer haplotypes. The relevant comparison is with variants of matched frequency in panels a-c. I don’t think you can draw any conclusions from eyeballing the haplotype lengths in panel d.

We agree and have removed this piece of interpretative text from the caption for panel d.

5) Terminology, but I don’t think that variants with a small effect on mRNA expression would be called LoF/GoF either under the common usage today, or under the classical definitions (which relate to protein function or organismal phenotypes). One can easily imagine a variant that increased gene expression leading to a loss-of-function at the protein level. “pLoF” just makes it even more confusing because in the human genetics literature it usually stands for “predicted-loss-of-function” not “partial-loss-of-function”. Others might disagree, but why not use “expression increasing/decreasing” or something to avoid confusion?

We appreciate the reviewer's comment and the observation regarding the potential confusion associated with the term “pLoF”. While we continue to use LoF/GoF language, we have revised the manuscript to clarify terminology.

In detail, in the text we now refer to the rs112233623-T allele as a “partial-LoF” variant, explicitly indicating that this designation refers to a partial loss of function at the gene expression level. We find this preferable over introducing “expression increasing/decreasing” due to the brevity and the alignment with observed coherent effects at the protein level. We believe that this change improves clarity and aligns the terminology more closely with current standards and reviewer

recommendations.

Referee #4 (Remarks to the Author):

This work investigates a variant in an enhancer for CCND3. The variant has a high allele frequency in Sardinians (as compared to Europeans) suggesting it may have resulted from positive selection. Prior studies have linked a different variant in the CCND3 enhancer with hematologic traits, and the variant of question in this work has also been previously implicated to have effects on RBC counts in data from the UK Biobank. Some of the key results the paper are 1) the demonstration of an association between the rs112233623-T allele and hematologic traits in Sardinians, the characterization of CCND3 expression and cell cycle parameters in erythroblasts derived from individuals with the TT vs WT background, and 3) the demonstration that the T allele leads to reduced enhancer activity, as well as the involvement of the transcription factors of SMAD3 and GATA1, providing a potential mechanism for the reduced CCND3 expression in the TT allele.

Additionally, the T allele is found to correlate with reduced growth of P. falciparum, correlated with elevated ROS levels, suggesting a mechanism of protection similar to that of G6PD.

Based on the feedback that starting with 5% schizont parasitemia would lead to parasite crashing, the authors have now stated that they made a reporting error, and that actually all of these experiments were performed at 2.5% starting schizont parasitemia, not 5%. With regards to the other aspects of Figure 4, the reviewers report that their response required “a massive and careful re-evaluation of all the results”, which resulted in a replacement of this figure with a “new” figure 4. In response to a concern about potential batch effects, they have also now reported that the experiments were performed in 5 batches, and that each genotype was

represented in each batch and that careful attention was paid to controlling for experimental conditions (however I could not find any report of which sample was in which batch).

1. Parasite invasion is highly variable depending on the precise culture conditions in the prior cycles, stress of the density gradient procedure to isolate schizonts, etc., and therefore the same parasitemia would not be expected from batch to batch even when genetic background and experimental conditions are maintained constant. Yet the graph in the new Figure 4b shows almost identical parasitemia within each genetic background at 48 hours (1st cycle), and the experimental variation remains minimal at subsequent timepoints, which is highly unexpected if performed in 5 different batches. This is also evident in the raw data provided for Figure 4b (excel file).

We thank the Reviewer for the careful reading and this observation, which has prompted us to discuss this topic in more detail, presenting new analyses and supplementary figures.

It is true that the visual inspection of the graph in the new Figure 4b shows low variability in parasitaemia within each genetic background. This may be partially due to a series of commonly used precautions to reduce experimental variability and enhance reproducibility, including: maintaining the age range of parasites at the initiation of cultures (time zero) as narrow as possible; the cultures used for the experiments initiated by adding *Percoll*-purified schizonts to red blood cells; schizonts purification performed on cultures that had been synchronized every three cycles; the use of isolated schizonts immediately after purification, along with stringent temperature control throughout the process to maximize parasite synchronicity over the course of the three replication cycles; and the similar storage time of RBCs used for all cultures.

Nevertheless, we believe that the main factor ensuring the relative low variability in the various batches is the fairly strong impact of the genetic factors. More specifically, we considered individuals who not only were homozygous for the GoF or GoF/PLoF genotypes, or who had the wild-type mating status at the two considered CCND3 SNPs, but importantly, who were all negative for beta- and alpha-thalassaemia traits, as well as for the G6PD Mediterranean deficiency mutation, (except for the group specifically selected to be hemizygous for the latter and negative for all other variants). The experiments were indeed set up to minimise variation and capitalise on the unique characteristics of the study population, which required considerable effort.

That said, we note that some variability was nevertheless observed among the data from the four batches considered in Figure 4b. This has now been presented in Supplementary Figure 3. This figure provides a visual representation of variability in parasitaemia within batches, as well as a statistical quantification of this variability using the non-parametric Kruskal–Wallis test to compare medians across time points for each genotype. Furthermore, when we assessed parasitaemia including the fifth batch (using the parasitaemia data from the more recently collected samples to assess --in response to a specific request of Reviewer 2--, possible effects on parasitaemia due to different parasite strains, in this case the *P. falciparum* Palo Alto strain data) we saw more variability (Supplemental Figure 4).

To fully clarify this point, we have now provided as Source Data for Figure 4b and Extended Figure 10 a Table listing the batch assignment for each donor used in the infection tests.

Finally, it is also important to note that all blood samples were obtained from volunteer participants in the SardiNIA cohort. Because of this, we faced the additional practical constraint that the pool of eligible donors was limited and that not all recalled individuals ultimately came for the scheduled blood draw for functional testing. Despite our efforts to maintain balance and consistency across experiments, this limited availability occasionally affected the exact number of individuals carrying a specific genotype within batches.

In the Methods section, we have revised the sentence to improve clarity (lines 1001-1005) as follows:

“The experiments were conducted in 5 batches to ensuring as much as possible that individuals recruited for each experimental batch had a similar proportion of the different genotypes, especially GoF and partial-LoF, in order to control for potential batch effects (Supplementary Information). A Table listing the batch assignment for each donor used in the infection experiments is provided in the Source Data.”

To provide a more precise description of the experimental design, we have further specified in the Supplementary Information the following details: *“To provide a more precise description of the experimental design, we specify that in the first two batches of the malaria infection experiments, we focused exclusively on the two extreme genotypes — partial-LoF and GoF — in order to preliminarily assess whether P. falciparum infection could represent the selective pressure underlying the strong positive selection observed for the partial-LoF genotype in Sardinia. Following the consistent results obtained in both batch 1 and batch 2, we expanded the experimental design in the subsequent batches to include also the Wild Type (WT) and the positively selected G6PD-deficient genotypes, which served as internal controls.*

Notably, samples used in batch 5 — which included all four genotypes — were specifically collected to assess whether the observed genotype-specific differences in resistance to malaria infection were consistent not only for the P. falciparum Palo Alto strain, but also for two additional genetically distinct P. falciparum isolates. Data from this experiment were used to generate Extended Data Figure 10.”

2a. As it is not clear what the “massive and careful re-evaluation of all the results” entailed, I reviewed the new Figure 4 in the context of the original Figure 4. In the reviewing the “new” Figure 4 in the context of the original Figure 4, several substantial discrepancies in the data become apparent. For example, in the original Figure 4 the % invasion shown in the graph was

defined as the ring parasitemia at 120 hr, and the % maturation was defined as the troph parasitemia at 120 hr. For the WT/WT/WT/WT genotype, the parasitemia for both was 3-4%, suggesting a total parasitemia of 6-8% with a ratio of 50:50 rings and trophs (original Figure 4f-g). However in the new Figure 4C, the graph shows that this genotype had 90% rings at the 120 hr timepoint, highlighting a substantial discrepancy. Moreover, while the authors notably no longer show parasitemia data for 120 hr in the Figure, in reviewing their raw data for the 120 hr timepoint, the parasitemias for the WT/WT/WT/WT are all ~17% (as opposed to the report of 6-8% at this timepoint for this genotype in the original Figure 4f-g).

The discrepancy is due to a typing error in the legend of the previous figure 4. Actually, old Figures 4f and 4g referred to two distinct time points (120 hours and 144 hours from the start of the cultures, 120 hours correspond to the ring stage while at 144 hours the same parasites had matured to trophozoites). We appreciate the Reviewer's concern and have corrected the typo; as an aside, we had already addressed this issue in our previous round of Responses, to address an earlier comment by Reviewer 2.

Furthermore, we would like to clarify that the parasitaemia data at 120 hours were not included in the new Figure 4 solely due to space constraints, which made it unfeasible to present all time points and the corresponding plots. However, the complete dataset, including parasitaemia percentages at all analysed timepoints (24, 48, 72, 96, 120, and 144 hours), is fully available as Source Data.

Should the Reviewer consider it important, we would be happy to include an additional Supplementary Figure displaying parasitaemia values for all timepoints.

2b. There is no explanation provided for these and many other discrepancies in the data for the two versions of Figure 4.

We would like to thank the reviewer for his/her thorough reading of the manuscript and submitted material. We agree that some detailed explanations about the issues noted are indeed necessary, and this has given us the opportunity to clarify some relevant aspects that were not reported or explained properly in the manuscript or in our previous point-by-point exchanges.

Specifically, the discrepancies between the old and new versions of Figure 4 boils down to differences in experimental conditions underlying the results shown in the two versions. The key point is that all malaria infection experiments were initiated in parallel using two haematocrit settings (2% and 1%). These settings were used for all replicates, with the old version of Figure 4 presenting results from experiments conducted at an initial haematocrit level of 2%, and the new version presenting findings from experiments conducted at an initial haematocrit level of 1%. The Pantaleo laboratory, where our *in vitro* malaria experiments were performed, uses this dual initial haematocrit setting, especially when reobtaining blood samples is challenging, as it offers the option to use the more reliable data from one of the two settings.

In response to the reviewer's initial observations/requests, our malaria group supported by Professor Franco Turrini, thus reanalysed the entire dataset at each time point (24, 48, 72, 96, 120 and 144 hours post-infection, respectively) for both duplicate cultures (2% and 1% haematocrits). This process enabled us to refine the treatment of some methodological aspects that had not been explicitly dealt with in the previous versions of the manuscript. In this revised version of the manuscript, we opted to present parasite growth in terms of total parasitaemia, excluding dead parasites (see Figure 4b). At the same time, we calculated the percentage of each parasite stage at each time point, including dead parasites, to highlight changes in GoF/GoF/PLoF/PLoF cultures (see Figure 4C). These results were similar to those obtained in G6PD Med hemizygous cultures. Crucially, it was realised that the new version of Figure 4 and the malaria experiments reported in the revised manuscript should be based more robustly on experiments conducted with an initial

haematocrit level of 1% and a parasitaemia level of 2.5%, given the lower impact of this setting on cultures of infected erythrocytes, especially the long-term ones, due to reduced glucose depletion and the resulting reduced negative effect on parasitaemia. Glucose consumption in these experiments is expected to be around 50% lower when compared to those performed with an initial haematocrit level of 2%. Under these conditions, as shown in Figure 4b, parasitaemia reached 4.06% after 48 hours, 9.40% after 96 hours, and 17.10% after 144 hours in WT controls, ensuring a steady parasite growth less affected by initial haematocrit levels and by the periodic addition of erythrocytes to the cultures. It reflects an approximate fourfold increase in the number of parasitised red cells taking into consideration the subsequent parasitaemia dilution resulting from the addition of uninfected packed RBCs prior to each reinvasion step. Thus, the focus on the 1% condition in the revised manuscript is aimed to improve the robustness and biological relevance of the conclusions in relation to genotype-specific effects on parasitaemia, while minimising the impact of variables related to the experimental setting.

We apologise for our oversight in omitting the fact that the new Figure 4 refers to experiments conducted at an initial haematocrit level of 1%, and for not including in the revised manuscript details about the design of the dual-haematocrit conditions and the rationale for choosing this protocol. This corresponding section of the Methods was accidentally omitted during the last internal revision process, and we only became aware of this upon reading the reviewers' comments. We are glad to have had the opportunity to correct this error and provide more detailed clarification about the study design and rationale for the chosen setting, as reported in detail from line 957 to line 979 in the Methods section of the revised manuscript.

“All experiments involving in vitro malaria infections were performed in parallel with two initial haematocrit settings: 2% and 1%. This parallel experimental design provided the option to select between the two settings for subsequent short- and long-term parasite culture analysis, circumventing

the complex and unpredictable process of repeating experiments from a small pool of donors with the relevant multilocus genotype combination, who are difficult to resample.

Samples with an initial haematocrit of 2% used 240 μ L of red blood cells (RBCs) with 2.5% schizont parasitaemia. We added 240 μ L of packed erythrocytes every 48 hours, coupled with replacement of the growth medium. This led to a gradual increase in haematocrit up to approximately 6% by the third cycle.

For 1% initial haematocrit samples, we used 2.5% schizont parasitaemia to infect 120 μ L of packed erythrocytes and the same amount of RBCs was added to the cultures every 48 hours, alongside the replacement of the growth medium. In this case the final haematocrit level reached approximately 3% by the third cycle.

These two settings were used for all replicates at each time point (24, 48, 72, 96, 120 and 144 hours post-infection, respectively). As analysis of long-term cultures was performed to allow a wider range of observations, experiments were selected that began with an initial haematocrit level of 1%.

This is because the haematocrit level significantly affects the glucose level: as the haematocrit increases, so does glucose consumption, resulting in glucose depletion; in turn, the glucose level significantly impacts parasitaemia, as the parasites rely heavily on glycolysis for energy and survival. Especially in the late cycles of long-term cultures, when the haematocrit level increases proportionally with the progressive addition of erythrocytes, we expect that parasitaemia will be less (negatively) affected by glucose consumption by both uninfected and infected erythrocytes at the selected lower haematocrit setting.”

3. The authors conclude that the phenotype for GoF/GoF/PLoF/PLoF became apparent in the second cycle. However, as apparent from the raw data the difference in parasitemia was substantial from the first timepoint at 24 hours, which is also evident in the figure at the 48 hr

timepoint (~4% parasitemia in the WT/WT/WT/WT in 48 hr trophozoites, and significantly reduced parasitemia in the GoF/GoF/PLoF/PLoF background). The staging data indicates that the parasites in the deficient background progressed normally from 100% rings to ~ 100% trophs in that first cycle, so this implies that there was an issue with invasion, consistent with the raw data.

We thank the Reviewer for this observation. We have revised the sentence accordingly, as follows (lines 320-324): “*Notably, parasite growth inhibition in GoF/GoF/PLoF/PLoF RBCs, already evident during the first cycle and suggestive of reduced invasion efficiency, became even more evident from the second cycle onward, where parasites were retained in the schizont stage, leading to delayed merozoite egress and reinvasion of uninfected RBCs (Figure 4c).*”

Referee #5 (Remarks to the Author):

I have examined the revised manuscript. I think that the authors have correctly addressed the comments of the previous reviewer and changed the manuscript accordingly.

Michele Pagano

We would like to thank the referees once again for their suggestions throughout the review process, including the latest ones. Our responses are provided below, point by point.

Referee#2

Suggestions:

I. Lines 310-311: Other known malaria-protective variants (HbS, HbC, O blood group, ATP2B4) are not mentioned here, why not? Is it possible these variants differ between the groups? It would be helpful to state this or acknowledge these are not measured. (HbS must surely be accounted for?)

We have now clarified this aspect in the Supplementary Information “*Additional malaria-related variants were checked in the collected samples, including rs334-T and rs33930165-C in HBB gene, associated with HbS and HbC diseases respectively. These variants were absent in our SardiNIA sequenced cohort. Other potentially relevant variants, such as rs8176719-T in ABO gene, rs3118662-A in REXO4 gene and rs10900585-T in ATP2B4 gene do not detectably influence parasitaemia in our experiments (Supplementary Figure 3).*”. We have also provided an additional Supplementary Figure 3 summarising these analyses.

II. Methods section on line 793 and onwards: For the experiments that use blood or cells derived from human volunteers (typo: line 794 'volonteers') it would be helpful to add more detail about the ascertainment of individuals, including what process was used to determine the participants that donated blood/cells? For example, state fully which genetic variants were examined (similar to the above question about lines 310-311) and how individuals were then selected based on these genotypes?

-We have clarified this aspect in the Methods section (lines 688-699). We now specify that: “*The samples collected for the experiments reported in this study were specifically selected to ensure representation of homozygous genotypes at the loci of interest, rs112233623 and rs9349205 close to the CCND3 gene. For the malaria experiments, individuals who were carriers of the most frequent allelic variants in Sardinia that have previously been associated with protection against malaria were excluded from the collection. These variants include the beta⁰39 thalassaemia mutation (rs11549407-A) in the HBB gene (frequency = 0.051), the 3.7 kb alpha thalassaemia deletion [NG_000006.1:g.34164_37967del3804 (frequency = 0.14)], and the Mediterranean deficiency variant (rs5030868-A) in the G6PD gene (frequency = 0.083), with the exception of a small group of samples that were selected to be hemizygous for the G6PD Mediterranean deficiency variant. We also tried to ensure a balance in terms of sex in the various genotype groups being compared (Supplementary Information).*”

- And also clarify what overlap there was in the donors used in the various different experiments (RNA and protein expression, Pf growth)? (In particular, was it the same individuals?)

The overlap in the donors used in the various different experiments across different datasets is now reported from lines 704 to 706 of the 'Methods' section of the manuscript “*Further details about sample collection are provided in the Supplementary Information, whereas the overlap of donors across the different functional and P. falciparum growth experiments is detailed in Supplementary Table 5.*”

III. This statement from the summary paragraph (lines 35-38) does not seem to be supported by direct evidence in the paper: "Furthermore, the reduced number of erythroblast divisions seen in rs112233623-T carriers is likely to counteract the well-established delayed erythroblast differentiation promoted by *P. falciparum* to ensure its gametocytogenesis, thereby alleviating diserythropoiesis and preventing parasite sexual maturation and transmission.". I'd suggest clarifying that this is a proposed mechanism, not something the paper establishes.

-We have removed the statement from the summary paragraph, as you rightly pointed out, it is not supported by any direct evidence in the paper

Referee #3 (Remarks to the Author):

I do think that the LoF/GoF terminology is incorrect, but up to them...

-We agree that the LoF/GoF terminology was indeed inappropriate and potentially misleading. We have therefore revised the terminology throughout the manuscript as you suggested. Thank you for highlighting this issue once more.

Referee #4 (Remarks to the Author):

The authors have provided more details to explain their methods, findings, and conclusions for the parasite assays in Figure 4, including the batch assignments for each sample, which has strengthened the paper. I appreciate their efforts in this regard.

In reviewing the manuscript, I feel comfortable that the authors have sufficiently addressed the prior reviewer feedback.

However a statement in the Summary that the cell cycle alterations in the LoF allele is "likely to counteract the well-established delayed erythroblast differentiation induced by *P. falciparum*" is not supported by any evidence in the paper, and may come across as misleading. I strongly suggest that this sentence be removed from the Summary and instead saved as speculation in the Discussion section. The authors do not do any infections in erythroblasts and do not attempt to examine the impact of *P. falciparum* infection on erythroblast differentiation; this could be investigated if the authors felt strongly about including it, but I do not think it should be mentioned in the Summary in the absence of any data.

- We have removed the statement from the summary paragraph, as it is indeed not supported by any direct evidence in this paper. This speculative point has been retained as a proposed mechanism in the 'Discussion' section. Thank you once again for all your helpful feedback.